# Structured Dropout Variational Inference for Bayesian Neural Networks

**Son Nguyen**[†,1]   **Duong Nguyen**[3]   **Khai Nguyen**[1]   **Khoat Than**[3,1]   **Hung Bui**[*,1]   **Nhat Ho**[*,2]

[1] VinAI Research, Viet Nam
[2] University of Texas, Austin
[3] Hanoi University of Science and Technology

## Abstract

Approximate inference in Bayesian deep networks exhibits a dilemma of how to yield high fidelity posterior approximations while maintaining computational efficiency and scalability. We tackle this challenge by introducing a novel variational structured approximation inspired by the Bayesian interpretation of Dropout regularization. Concretely, we focus on the inflexibility of the factorized structure in Dropout posterior and then propose an improved method called *Variational Structured Dropout* (VSD). VSD employs an orthogonal transformation to learn a structured representation on the variational Gaussian noise with plausible complexity, and consequently induces statistical dependencies in the approximate posterior. Theoretically, VSD successfully addresses the pathologies of previous Variational Dropout methods and thus offers a standard Bayesian justification. We further show that VSD induces an adaptive regularization term with several desirable properties which contribute to better generalization. Finally, we conduct extensive experiments on standard benchmarks to demonstrate the effectiveness of VSD over state-of-the-art variational methods on predictive accuracy, uncertainty estimation, and out-of-distribution detection.

## 1   Introduction

Bayesian Neural Networks (BNNs) [49, 63] offer a probabilistic interpretation for deep learning models by imposing a prior distribution on the weight parameters and aiming to infer a posterior distribution instead of only point estimates. By marginalizing over this posterior for prediction, BNNs perform a procedure of ensemble learning. These principles improve the model generalization, robustness and allow for uncertainty quantification. However, exactly computing the posterior of non-linear BNNs is infeasible and approximate inference has been devised. The core challenge is how to construct an expressive approximation for the true posterior while maintaining computational efficiency and scalability, especially for modern deep learning architectures.

Variational inference is a popular deterministic approximation approach to deal with this challenge. The first practical methods were proposed in [22, 8, 39], in which the approximate posteriors are assumed to be fully factorized distributions, also called mean-field variational inference. In general, the mean-field approximation family encourages several advantages in inference including computational tractability and effective optimization with the stochastic gradient-based methods. However, it ignores the strong statistical dependencies among random weights of neural nets, leading to the inability to capture the complicated structure of the true posterior and to estimate the true model uncertainty.

---

*These two authors contributed equally. †Correspondence to Son Nguyen: `<v.sonnv27@vinai.io>`

35th Conference on Neural Information Processing Systems (NeurIPS 2021).

To overcome this limitation, many extensive studies proposed to provide posterior approximations with richer expressiveness. For instance, [47] treats the weight matrix as a whole via a matrix variate Gaussian [24] and approximates the posterior based on this parametrization. Several later works have exploited this distribution to investigate different structured representations for the variational Gaussian posterior, such as Kronecker-factored [89, 71, 72], k-tied distribution [77], non-centered or rank-1 parameterization [21, 15]. Another original idea to represent the true covariance matrix of Gaussian posterior is by employing the low-rank approximation [67, 35, 80]. For robust approximation with multimodality, [48] adopted hierarchical variational model framework [69] for inferring an implicit marginal distribution in high dimensional Bayesian setting. Despite significant improvements in both predictive accuracy and uncertainty calibration, some of these methods incur a large computational complexity and are difficult to integrate into deep convolutional networks.

**Motivations.** In this paper, we approach the structured posterior approximation in Bayesian neural nets from a different perspective which has been inspired by the Bayesian interpretation of Dropout training [74, 51]. More specifically, the methods proposed in [39, 20] reinterpret Dropout regularization as approximate inference in Bayesian deep models and base on this connection to learn a variational Dropout posterior over the weight parameters. From the literature, inference approaches based on Bayesian Dropout have shown competitive performances in terms of predictive accuracy on various tasks, even compared to the structured Bayesian methods aforementioned, but with much cheaper computational complexity. Moreover, with the solid and intriguing theories on effective regularization [81, 26, 83], generalization bound [52, 59], convergence rate and robust optimization [55, 54, 7], Dropout principle offers several potentials to further improve approximate inference in Bayesian deep networks. However, since these Bayesian Dropout methods also employed simple structures of the mean-field family, their approximations often fail to obtain satisfactory uncertainty estimates [17]. In addition, Variational Dropout methods based on multiplicative Gaussian noise also suffer from theoretical pathologies, including improper prior leading to ill-posed true posterior, and singularity of the approximate posterior making the variational objective undefined [31].

**Contributions.** With the above insights, we propose a novel structured variational inference framework, which rationally acquires complementary benefits of the flexible Bayesian inference and Dropout inductive bias. Our method adopts an orthogonal approximation called Householder transformation to learn a structured representation for multiplicative Gaussian noise in Variational Dropout method [39, 57]. As a consequence of the Bayesian interpretation, we go beyond the mean-field family and obtain a variational Dropout posterior with structured covariance. Furthermore, to make our framework more expressive, we deploy a hierarchical Dropout procedure, which is equivalent to inferring a joint posterior in a hierarchical Bayesian neural nets. We name the proposed method as *Variational Structured Dropout* (VSD) and summarize its advantages as follows:

**1.** Our structured approximation is implemented on low dimensional space of variational noise with considerable computational efficiency. VSD can be employed for deep CNNs in a direct way while maintaining the backpropagation in parallel and optimizing efficiently with gradient-based methods.

**2.** Especially, VSD has a standard Bayesian justification, in which our method can overcome the critiques from the non-Bayesian perspective of previous Variational Dropout methods. Our inference framework uses a proper prior, non-singular approximate posterior and derives a tractable variational lower bound without further simplified approximation.

**3.** Compared with previous Bayesian Dropout methods which are relatively inflexible by some *strict conditions*, VSD is more efficient on both the criteria of expressive approximation and flexible hierarchical modeling. Therefore, VSD is promising to be a general-purpose approach for Bayesian inference and in particular for BNNs.

**4.** To reinforce the complementary advantages unified in our proposal, we also investigate the inductive biases induced by the adaptive regularization of structured Dropout noise. We further provide an interpretation that VSD implicitly facilitates the networks to converge to a local minima with smaller spectral norms and stable rank. This properties suggests better generalization and we present empirical results to support this implication.

**5.** Finally, we carry out extensive experiments with standard datasets and different network architectures to validate the effectiveness of our method on many criteria, including scalability, predictive accuracy, uncertainty calibration, and out-of-distribution detection, in comparison to popular variational inference methods.

**Notation.** For a matrix $\mathbf{A}$, $\|\mathbf{A}\|_F$ and $\mathbf{A}^\top$ denotes the Frobenius norm and the transpose matrix, $\mathbf{A}_{i:}$ and $\mathbf{A}_{:j}$ denote the $i$-th row and the $j$-th column. For an interger $i$, $\mathbf{e}_i$ is the $i$-th standard basis, $\mathbf{1}_i \in \mathbb{R}^i$ is the vector of all ones. The diagonal matrix with diagonal entries as the elements of a vector $\mathbf{x}$ is denoted by diag($\mathbf{x}$). The inner product between two matrices $\mathbf{A}$ and $\mathbf{B}$ is denoted by $\langle \mathbf{A}, \mathbf{B} \rangle$.

## 2   Background

**Variational inference for Bayesian neural networks:** Given a dataset $\mathcal{D}$ consisting of input-output pairs $(\mathbf{X}, \mathbf{Y}) := \{(\mathbf{x}_n, \mathbf{y}_n)\}_{n=1}^N$. In BNNs, we impose a prior distribution over random weights $p(\mathbf{W})$ whose form is in a tractable parametric family and aim to infer an intractable true posterior $p(\mathbf{W}|\mathcal{D})$. Variational inference (VI) [30, 33] can do this by specifying a variational distribution $q_\phi(\mathbf{W})$ with free parameter $\phi$ and then minimizing the Kullback-Leibler (KL) divergence $\mathbb{D}_{KL}(q_\phi(\mathbf{W})\|p(\mathbf{W}|\mathcal{D}))$. This optimization is equivalent to maximizing the Evidence Lower Bound (ELBO) with respect to variational parameters $\phi$ as follows:

$$\mathcal{L}(\phi) = \mathbb{E}_{q_\phi(\mathbf{W})} \log p(\mathcal{D}|\mathbf{W}) - \mathbb{D}_{KL}(q_\phi(\mathbf{W})\|p(\mathbf{W})). \tag{1}$$

By leveraging the reparameterization trick [40] combined with the Monte Carlo integration, we can derive an unbiased differentiable estimation for the variational objective above. Then, this estimation can be effectively optimized using stochastic gradient methods with the variance reduction technique such as the *local* reparameterization trick [39].

**Variational Bayesian inference with Dropout regularization:** Given a deterministic neural net with the weight parameter $\Theta$ of size $K \times Q$, training this model with stochastic regularization techniques such as Dropout [29, 74] can be interpreted as approximate inference in Bayesian probabilistic models. This is because injecting a stochastic noise into the input layer is equivalent to multiplying the rows of subsequent deterministic weight by the same random variable, namely with each datapoint $(\mathbf{x}_n, \mathbf{y}_n)$ and a noise vector $\xi$, we have: $\mathbf{y}_n = (\mathbf{x}_n \odot \xi)\Theta = \mathbf{x}_n \text{diag}(\xi)\Theta$. This induces a BNN with random weight matrix defined by $\mathbf{W} := \text{diag}(\xi)\Theta$. Applying VI to this Bayesian model, with some specific choices for prior and approximate posterior, the variational lower bound (1) can resemble the form of Dropout objective in the original deterministic network. This principle is referred to as the KL condition [18]. Gal et al. [20], Kingma et al. [39] used this principle to propose Bayesian Dropout inference methods such as MC Dropout (MCD) and Variational Dropout (VD).

Dropout inference is practical approximate framework especially in high dimensional setting. However, the scope of Bayesian inference in these methods is restricted in terms of flexibility of both prior and approximate posterior. Concretely, the Dropout posteriors $q_\phi(\mathbf{W})$ in MCD and VD both have simple structures of mean-field approximation which often underestimate the variance of true posterior, possibly leading to a poor uncertainty representation [17]. Moreover, in theory, VD employed an improper log-uniform prior which can result in an ill-posed true posterior and generally push the parameters towards the penalized maximum likelihood solution [31]. In addition, VD also suffers from the singularity issue of approximate posterior that makes the KL divergence term undefined. Our work gains an efficient remedy to these pathologies.

## 3   Variational Structured Dropout

We focus on Bayesian Dropout methods using multiplicative Gaussian noise with correlated parameterization [39]. This procedure induces a random weight $\mathbf{W} = \text{diag}(\xi)\Theta$, where the Dropout noise $\xi$ is a multivariate Gaussian with diagonal covariance $q_\alpha(\xi) = \mathcal{N}(\mathbf{1}_K, \text{diag}(\alpha))$, and $\alpha$ is the droprate vector. The corresponding Dropout posterior then is given by $q_\phi(\mathbf{W}) = \text{Law}(\text{diag}(\xi)\Theta)$. This distribution on each column exhibits a factorized structure with the form of $q(\mathbf{W}_{:j}) = \mathcal{N}(\Theta_{:j}, \text{diag}(\alpha \odot \Theta_{:j}^2))$, whilst allows a correlation on each row because each $\mathbf{W}_{i:}$ is shared by the same scalar noise $\xi_i$ respectively. The parameters $\phi := (\alpha, \Theta)$ are optimized via maximizing a variational lower bound as follows: $\mathcal{L}(\phi) := \mathbb{E}_{q_\phi(\mathbf{W})} \log p(\mathcal{D}|\mathbf{W}) - \mathbb{D}_{KL}(q_\phi(\mathbf{W})\|p(\mathbf{W})) = \mathbb{E}_{q_\alpha(\xi)} \log p(\mathcal{D}|\xi, \Theta) - \mathbb{D}_{KL}(q_\phi(\mathbf{W})\|p(\mathbf{W}))$, where the later equation is derived from the change of variables formula.

### 3.1   The orthogonal approximation for variational structured noise

Intuitively, a richer representation for the noise distribution can enrich the expressiveness of Dropout posterior via the Bayesian interpretation. We implement this intuition with an assumption that the

Dropout noise could be sampled from a Gaussian distribution with a full covariance matrix instead of a diagonal structure, namely, $q_{\mathbf{\Sigma}}(\xi) = \mathcal{N}(\mathbf{1}_K, \mathbf{\Sigma})$ with $\mathbf{\Sigma}$ is a positive definite matrix of size $K \times K$. To make this covariance matrix learnable, we first represent $\mathbf{\Sigma}$ in the form of the spectral decomposition: $\mathbf{\Sigma} = \mathbf{P}\mathbf{\Lambda}\mathbf{P}^T$, where $\mathbf{P}$ is an orthogonal matrix with its eigenvectors in columns, $\mathbf{\Lambda}$ is a diagonal matrix where diagonal elements are the eigenvalues. By the basis-kernel representation theorem [6, 76], we parameterize the orthogonal matrix $\mathbf{P}$ as a product of Householder matrices in the following form: $\mathbf{P} = \mathbf{H}_{T^*}\mathbf{H}_{T^*-1}...\mathbf{H}_1$, where $\mathbf{H}_t = \mathbf{I} - 2\mathbf{v}_t\mathbf{v}_t^T/\|\mathbf{v}_t\|_2^2$, $\mathbf{v}_t$ is the Householder vector of size $K$, and $T^*$ is the degree of $\mathbf{P}$. This parameterization relaxes the orthogonal constraint of matrix $\mathbf{P}$, and we can then directly optimize the covariance matrix $\mathbf{\Sigma}$ via gradient-based methods.

Notably, this transformation can be interpreted as a sequence of invertible mappings. More explicitly, we extract a zero-mean Gaussian noise $\eta^{(0)}$ from the original noise $\xi^{(0)} \sim \mathcal{N}(\mathbf{1}_K, \text{diag}(\alpha))$ in the form of $\xi^{(0)} = 1 + \eta^{(0)}$, and by successively transforming $\eta^{(0)}$ through a chain of $T$ Householder reflections, we obtain the induced noise and the corresponding density at each step $t$ as follows:

$$\xi^{(t)} := 1 + \mathbf{H}_t\mathbf{H}_{t-1}...\mathbf{H}_1\eta^{(0)} = 1 + \mathbf{U}\eta^{(0)}, \quad \text{and} \quad q_t(\xi) := \mathcal{N}(\mathbf{1}_K, \mathbf{U}\text{diag}(\alpha)\mathbf{U}^T).$$

By injecting the structured noise $\xi^{(t)}$ into the deterministic weight $\Theta$, we obtain a random weight $\mathbf{W}^{(t)} := \text{diag}(\xi^{(t)})\Theta$ and an induced Dropout posterior $q_t(.) = \text{Law}(\text{diag}(\xi^{(t)})\Theta)$ with *fully correlated* representation, in which the marginal column distribution is given by: $q_t(\mathbf{W}_{:j}) = \mathcal{N}(\Theta_{:j}, \text{diag}(\Theta_{:j})\mathbf{U}\text{diag}(\alpha)\mathbf{U}^T\text{diag}(\Theta_{:j}))$. Detailed discussion about the expressiveness of this correlated structure is presented in Appendix A.1. With the above derivations, we use variational inference with approximate posterior $q_t(\mathbf{W})$ and optimize the variational lower bound as follows:

$$\mathcal{L}(\phi) := \mathbb{E}_{q_t(\mathbf{W})} \log p(\mathcal{D}|\mathbf{W}) - \mathbb{D}_{KL}(q_t(\mathbf{W})\|p(\mathbf{W}))$$
$$= \mathbb{E}_{q_\alpha(\xi)} \log p(\mathcal{D}|\Theta, \xi^{(t)}) - \mathbb{D}_{KL}(q_t(\mathbf{W})\|p(\mathbf{W})). \tag{2}$$

**Overcoming the singularity issue of approximate posterior.** In Variational Dropout method with correlated parameterization, there is a mismatch in support between the approximate posterior and the prior, thus making the KL term $\mathbb{D}_{KL}(q(\mathbf{W})\|p(\mathbf{W}))$ undefined [31]. Specifically, the form $\mathbf{W} = \text{diag}(\xi)\Theta$ is equivalent to multiplying each row $\mathbf{W}_{i:}$ by the same scalar noise $\xi_i$, namely $\mathbf{W}_{i:} = \xi_i\Theta_{i:}$ with $q(\xi_i) = \mathcal{N}(1, \alpha_i)$. This means that the approximate distribution always assigns all its mass on subspaces defined by the directions aligned with the rows of $\Theta$. These subspaces have Lebesgue measure zero causing the singularities in approximate posterior, and the KL term will be undefined whenever the prior $p(\mathbf{W})$ puts zero mass to these subspaces. However, in VSD the scalar noises are not treated separately due to the structured correlation, namely VSD would injects each $\xi_i$ into the whole matrix $\Theta$ instead of some individual directions. Indeed, we have: $\mathbf{W}^{(VD)} := \text{diag}(\xi^{(0)})\Theta = \Theta + \text{diag}(\eta^{(0)})\Theta = \Theta + \sum_{i=1}^K \eta_i^{(0)}(\text{diag}(\mathbf{e}_i)\Theta) = \Theta + \sum_{i=1}^K \eta_i^{(0)}\Theta_{(i)}$, and $\mathbf{W}^{(VSD)} := \text{diag}(\xi^{(t)})\Theta = \Theta + \text{diag}(\mathbf{U}\eta^{(0)})\Theta = \Theta + \sum_{i=1}^K \eta_i^{(0)}(\text{diag}(\mathbf{U}_{i:})\Theta)$, where $\Theta_{(i)}$ is the matrix $\Theta$ with only the $i$-th row retained. While VD causes *singular* components represented by $\{\Theta_{(i)}\}_{i=1}^K$, VSD maintains a trainable orthogonal matrix $\mathbf{U}$ which prevents the approximate posterior from having degenerate supports with measure zero, thereby avoiding the singularity issue. In the following section, we will present an appropriate choice of the prior distribution $p(\mathbf{W})$ such that the KL term is well-defined, and then derive a tractable objective function satisfying the KL condition in Bayesian Dropout frameworks.

### 3.2 Derivation of tractable variational objective

We consider employing an isotropic Gaussian as the prior distribution, namely $p(\mathbf{W}) = \prod_{j=1}^Q p(\mathbf{W}_{:j})$ with $p(\mathbf{W}_{:j}) = \mathcal{N}(0, \text{diag}(\beta_{:j}^{-1}))$ and $\beta$ is a hyper-parameter matrix of the same size with $\mathbf{W}$. With the previous analysis, our approximate posterior $q_t(\mathbf{W})$ would be absolutely continuous w.r.t the prior $p(\mathbf{W})$, and thus the KL term $\mathbb{D}_{KL}(q_t(\mathbf{W})\|p(\mathbf{W}))$ is defined. Furthermore, the Gaussian prior helps our proposal avoid the pathologies of improper true posterior and ill-posed inference in VD.

Note that, the prior $p(\mathbf{W})$ is a fully factorized Gaussian which usually facilitates simple analysis and efficient computation. This factorized structure is chosen also because we have no reason for the correlation between non-identical weights at first. Moreover, several arguments indicate that a simple prior over parameter $p(\mathbf{W})$, when interacts with neural nets architecture $f(\mathcal{D}; \mathbf{W})$, induces a sophisticated prior over function $p(f(\mathcal{D}; \mathbf{W}))$, with desirable properties and useful inductive

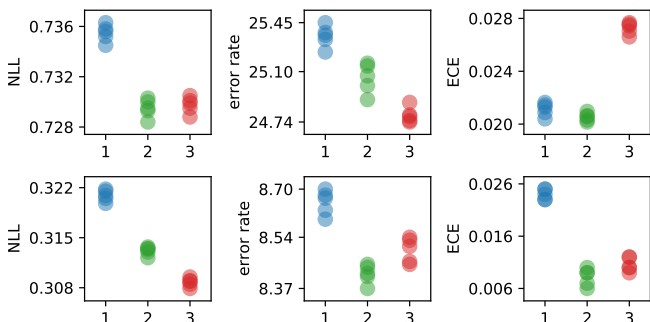

Figure 1: The performance of VSD when using the number of transformations $T \in \{1, 2, 3\}$. Evaluation over 5 runs on CIFAR10 (above) and SVHN (below) with LeNet architecture.

biases [86]. However, when we are interested in structured approximations in parameter space, the factorized prior may raise some contradictions. By relative entropy decomposition, we have:

$$\mathbb{D}_{KL}(q_t(\mathbf{W})||p(\mathbf{W})) = \sum_{j=1}^{Q} \mathbb{D}_{KL}(q_t(\mathbf{W}_{:j})||p(\mathbf{W}_{:j})) + \mathbf{I}(\mathbf{W}_{:1}, \mathbf{W}_{:2}, ..., \mathbf{W}_{:Q}), \qquad (3)$$

where $\mathbf{I}(.)$ is the mutual information measured by the distribution $q_t(.)$, and this term is validly defined in VSD. Maximizing the variational lower bound tends to encourage smaller KL term, and hence constrains the components in RHS of equation (3). Intuitively, a relatively small mutual information can break the strong correlations between the columns of $\mathbf{W}$. Several studies have focused on this limitation and suggested using richer priors such as matrix variate Gaussian [75, 91], doubly semi-implicit distribution [58]. Our solution for this scenario is derived naturally from equation (3), in which we leverage the mutual information as an additional regularization term. Concretely, we maximize an alternative variational objective as follows:

$$\mathcal{L}_{MI}(\phi) := \mathbb{E}_{q_\alpha(\xi)} \log p(\mathcal{D}|\Theta, \xi^{(t)}) - \mathbb{D}_{KL}(q_t(\mathbf{W})||p(\mathbf{W})) + \mathbf{I}(\mathbf{W}_{:1}, \mathbf{W}_{:2}, ..., \mathbf{W}_{:Q})$$

$$= \mathbb{E}_{q_\alpha(\xi)} \log p(\mathcal{D}|\Theta, \xi^{(t)}) - \mathbb{D}_{KL}(q_t^\star(\mathbf{W})||p(\mathbf{W})), \qquad (4)$$

where $q_t^\star(\mathbf{W}) := \prod_{j=1}^{Q} q_t(\mathbf{W}_{:j})$ is the product of marginal column distributions. From the information-theoretic perspective, augmenting the mutual information is a standard principle for structure learning in Bayesian networks [41]. Particularly in our derivation, maximizing the alternative objective $\mathcal{L}_{MI}(\phi)$ would help to sustain the dependence structure between columns of the network weights, and thus fixes appropriately the *broken* ELBO as mentioned above. Interestingly, this technique is utilized reasonably in our method. This is because our dependence structure allows to specify explicitly the marginal distribution on each column of $\mathbf{W}$, leading to a tractable objective in equation (4) whose KL divergence between two multivariate Gaussian can be calculated analytically in closed-form. We note that a similar application to other structured approximations, such as low-rank, Kronecker-factored or matrix variate Gaussian, can be non-trivial. We also remark that our alternative variational objective might be not necessarily a valid lower bound of the original model evidence, but would be the lower bound of new model evidence defined on an alternative prior $\hat{p}(\mathbf{W})$, which satisfies $\mathbb{D}_{KL}(q_t(\mathbf{W})||\hat{p}(\mathbf{W})) = \mathbb{D}_{KL}(q_t^\star(\mathbf{W})||p(\mathbf{W}))$. Indeed, the correlated prior determined by $\hat{p}(\mathbf{W}) \propto p(\mathbf{W}) * q_t(\mathbf{W})/q_t^\star(\mathbf{W})$ meets this constraint, and thus we could reinterpret the use of mutual information as adopting this prior at each iteration of the training procedure. To clarify, our idea was partly motivated by the similar technique that has been extensively adopted in deep latent variable models, in which a mutual information maximization is also added to the variational lower bound to mitigate the degenerate issue of amortized inference in these models [1, 92].

**The KL condition in VSD.** With the new variational objective in equation (4), to offer VSD complementary advantages of structured Dropout and Bayesian inference, we need to ensure $\mathbb{D}_{KL}(q_t^\star(\mathbf{W})||p(\mathbf{W}))$ satisfies the KL condition. We solve this prerequisite by specifying the precision parameter $\beta$ of the prior $p(\mathbf{W})$ via the Empirical Bayes (EB) approach. As a result, we obtain an optimal value for this hyperparameter in an analytical form $\beta^*$. The optimal $\beta^*$ is then substituted back into the prior, and thereby we get the optimal KL term with a form independent of the deterministic weight $\Theta$ as follows:

$$\mathbb{D}_{KL}^{EB}(q_t^\star(\mathbf{W})||p(\mathbf{W})) = \frac{Q}{2} \sum_{i=1}^{K} \log \frac{1 + \sum_{j=1}^{K} \alpha_j U_{ij}^2}{\alpha_i}. \qquad (5)$$

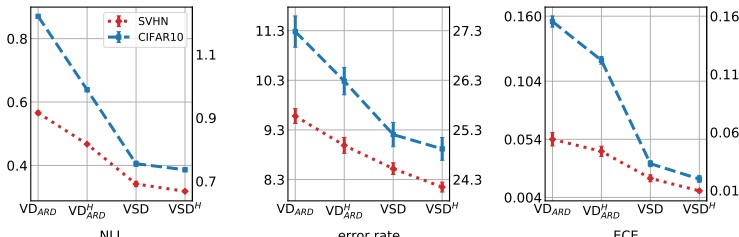

Figure 2: The performance of ARD-VD and VSD when using the prior hierarchy (coressponding to the labels $\text{VD}_{ARD}^H$ and $\text{VSD}^H$). The left and right $y-$axis are represented for SVHN and CIFAR10 dataset, respectively.

A formal proof of equation (5) is in Appendix B.

**The number of transformations steps $T$ in VSD**. An appropriate choice of $T$ is essential in our method. For deep learning architectures, the degree $T^*$ of the orthogonal matrix $P$ might be relatively large, so it is quite challenging to adjust the empirical value of $T$ in a principled way to meet the basis-kernel representation theorem. We can employ an efficient parameterization introduced in [90], in which we only need the Householder vectors $\{\mathbf{v}_t, t \geq 1\}$ with sizes much smaller than the order of the matrix $P$. This will facilitate tuning $T$ with a larger range in an applicable computation time. In our method, we instead only adopt a small $T \in \{1, 2, 3\}$ as a form of *"low-degree"* approxiamtion, and to make the reflections more expressive, we use a fully connected layer between successive Householder vectors, i.e. $\mathbf{v}_t = \mathbf{FC}(\mathbf{v}_{t-1})$. The low dimension of variational noise in VSD leads to a good adaptation, but with a little trade-off in computational complexity when increasing the number of transformation steps $T$. We show the performance of VSD when using this neural parameterization with $T \in \{1, 2, 3\}$ in Figure 1, where a larger $T$ could potentially improve the results on predictive measures. However, to maintain computational efficiency, we recommend using $T = 1$ or $2$ in large-scale experiments. Indeed, this neural parameterization has been successfully implemented in the context of learning the latent space in deep latent variable models [79, 5].

### 3.3 Joint inference with hierarchical prior

We further promote our proposal by introducing a prior hierarchy in VSD framework and then obtain a joint approximation for the Dropout posterior. This will facilitate the flexibility of Bayesian inference in our method in terms of the expressiveness of both prior distribution and approximate posterior. We design a two-level hierarchical prior given by: $p(\mathbf{W}, \mathbf{z}) = p(\mathbf{W}|\mathbf{z}, \beta)p(\mathbf{z})$, where $p(\mathbf{W}|\mathbf{z}, \beta) = \prod_{j=1}^{Q} p(\mathbf{W}_{:j}|\mathbf{z}, \beta_{:j})$ and $p(\mathbf{W}_{:j}|\mathbf{z}, \beta_{:j}) = \mathcal{N}(0, \text{diag}(\mathbf{z} \odot \beta_{:j}^{-1}))$; the hyperprior $p(\mathbf{z})$ is a distribution with positive support such as Gamma or half-Cauchy distribution; the latent $\mathbf{z}$ has the size of the number of rows and is shared across columns of the weight matrix $\mathbf{W}$; the hyper-parameter matrix $\beta$ is treated as a scaling factor. This prior family has a centered parameterization and induces several compelling properties such as facilitating feature sparsity [46, 12], model selection [21] or improving robustness, uncertainty calibration [15].

We implement variational inference with a joint approximate posterior, also referred to as the joint Dropout posterior, which is parameterized as follows:

$$q_\phi(\mathbf{W}, \mathbf{z}) = q_\psi(\mathbf{z})q_\phi(\mathbf{W}|\mathbf{z}) \quad \text{with} \quad q_\phi(\mathbf{W}_{:j}|\mathbf{z}) = \mathcal{N}(\mathbf{z} \odot \Theta_{:j}, \text{diag}(\mathbf{z}^2 \odot \alpha \odot \Theta_{:j}^2)),$$

where $q_\phi(\mathbf{W}|\mathbf{z})$ is the conditional Dropout posterior, $q_\psi(\mathbf{z})$ is chosen depending on the family of prior $p(z)$ so that the reparametrization trick can be utilized. Sampling the random weight $\mathbf{W}$ from the joint variational posterior $q_\phi(\mathbf{W}, \mathbf{z})$ includes two steps: $\mathbf{z}^* \sim q_\psi(\mathbf{z})$ and $\mathbf{W}^* \sim q_\phi(\mathbf{W}|\mathbf{z}^*)$, in which the second one can be reparameterized as: $\mathbf{W}^* = \text{diag}(\mathbf{z}^*)\text{diag}(\xi)\Theta = \text{diag}(\mathbf{z}^* \odot \xi)\Theta$, with the noise $\xi \sim \mathcal{N}(\mathbf{1}_K, \text{diag}(\alpha))$. This new representation adapts to the vanilla Dropout procedure but allows our method to regularize each unit layer with different levels of stochasticity. We derive some insights about the role of hierarchical prior in our framework in Appendix A.2. We apply the Householder transformation to the variational noise $\xi$ and obtain a new joint approximate posterior:

$$q_t(\mathbf{W}, \mathbf{z}) = q_\psi(\mathbf{z})q_t(\mathbf{W}|\mathbf{z}) \quad \text{with} \quad q_t(\mathbf{W}_{:j}|\mathbf{z}) = \mathcal{N}(\mathbf{z} \odot \Theta_{:j}, \mathbf{V}_j\mathbf{U}\text{diag}(\alpha)(\mathbf{V}_j\mathbf{U})^T),$$

where $\mathbf{V}_j = \text{diag}(\mathbf{z} \odot \Theta_{:j})$. Similarly, we define $q_t^\star(\mathbf{W}|\mathbf{z}) = \prod_{j=1}^{Q} q_t(\mathbf{W}_{:j}|\mathbf{z})$ and then optimize an alternative variational objective given by:

$$\mathcal{L}(\phi, \psi) := \mathbb{E}_{q_\psi(\mathbf{z})q_t(\mathbf{W}|\mathbf{z})} \log p(\mathcal{D}|\mathbf{W}) - \mathbb{E}_{q_\psi(\mathbf{z})}(\mathbb{D}_{KL}(q_t^\star(\mathbf{W}|\mathbf{z}, \phi)||p(\mathbf{W}|\mathbf{z}, \beta)) - \mathbb{D}_{KL}(q_\psi(\mathbf{z})||p(\mathbf{z})).$$

Table 1: Computational complexity per layer of MAP and different variational methods.

| Method | Time | Memory |
|---|---|---|
| MAP | $\mathcal{O}(KL|\mathcal{B}|)$ | $\mathcal{O}(L|\mathcal{B}|)$ |
| BBB | $\mathcal{O}(sKL|\mathcal{B}|)$ | $\mathcal{O}(sKL + L|\mathcal{B}|)$ |
| BBB-LTR | $\mathcal{O}(2KL|\mathcal{B}|)$ | $\mathcal{O}(2L|\mathcal{B}|)$ |
| VMG | $\mathcal{O}(m^3 + 2KL|\mathcal{B}|)$ | $\mathcal{O}(KL|\mathcal{B}|)$ |
| SLANG | $\mathcal{O}(r^2KL + rsKL|\mathcal{B}|)$ | $\mathcal{O}(rKL + sKL|\mathcal{B}|)$ |
| ELRG | $\mathcal{O}(r^3 + (r+2)KL|\mathcal{B}|)$ | $\mathcal{O}((r+2)L|\mathcal{B}|)$ |
| **VSD** | $\mathcal{O}(K^2 + KL|\mathcal{B}|)$ | $\mathcal{O}(K^2 + K|\mathcal{B}|)$ |
| **VSD-low rank** | $\mathcal{O}(rK + KL|\mathcal{B}|)$ | $\mathcal{O}(K^2 + K|\mathcal{B}|)$ |

Table 2: Computation time of variational methods compared to standard MAP (1x).

| Methods | Time/epochs (s) | | |
|---|---|---|---|
| | LeNet5 | AlexNet | ResNet18 |
| BBB-LTR | 1.53x | 1.75x | 3.28x |
| MNF | 2.86x | 3.40x | 4.88x |
| VD | 1.18x | 1.15x | 1.32x |
| VSD $T=1$ | 1.25x | 1.32x | 1.86x |
| VSD $T=2$ | 1.35x | 1.49x | 2.90x |

We have chosen the (inverse) Gamma and log-Normal distribution for $p(z)$ and $q(z)$ respectively. These distributions have positive supports, can be reparametrized and the KL-divergence between them also has a closed-form due to the conjugacy. A full derivation of $\mathcal{L}(\phi, \psi)$ including the KL condition is given in Appendix C. The parameterization of the prior hierarchy in our method is flexible without any simplifying assumptions about hyperprior $p(\mathbf{z})$. We can directly apply it for ARD-Variational Dropout framework (ARD-VD) [36] (a derivation is in Appendix F). As the experimental results are shown in Figure 2, the hierarchical prior significantly improves the performance of both ARD-VD and VSD on predictive metrics. Therefore, we aim to introduce VSD with hierarchical prior as an unified framework and a general-purpose approach for Bayesian inference, particularly for BNNs.

### 3.4 Scalability of Variational Structured Dropout

Approximating a structured posterior directly on the random weights of deep convolutional models is challenging. Besides expensive computation, it is difficult to employ the local reparameterization trick [39], leading to the high variance issue in training. We apply VSD to convolutional layer by learning a structured noise with the size of the number of kernels and imposing it to convolutional weights: $\xi \sim \mathcal{N}(\mathbf{1}_K, \mathbf{U}\text{diag}(\alpha)\mathbf{U}^T)$ and $\mathbf{W}_{ijk} = \xi_k \Theta_{ijk}$, with $i, j, k$ are the indexes representing height, width, and kernel respectively. This simple solution greatly reduces computational complexity while being able to captures the dependencies among kernels of the convolutional layer.

We present the complexity of MAP and VI methods in terms of computational cost and memory storage in Table 1, with the detailed analysis given in Appendix D. VSD adopts the advantage of Dropout training and maintains an efficiency on both criteria. We also give more results in Table 2 about the empirical computation time of VSD and some other methods. Based on these tables, VSD shows more effective running time even than the mean-field BBNs. Although there is a trade-off when using a larger number of $T$, VSD does not incur much extra computation time compared to VD.

### 3.5 On explicit regularization of Variational Structured Dropout

There are several compelling theories to explain the tremendous success of Dropout technique, in which regularization-based is one of the most active approaches [81, 26, 55, 53, 83, 10]. We would follow this direction to investigate inductive biases induced by the structured Dropout in VSD, and from which to consolidate our claim of complementary advantages unified in the proposed method. To characterize the regularization of VSD, we consider a deep linear neural net with $L$ layers parameterized by $\{\Theta^{(i)}\}_{i=1}^{L}$, and define some notations as: $\mathbf{x}$ is an input data, $\mathcal{B}$ is the data batch; $\mathbf{h}_i$ is the $i$-th hidden layer; $\mathbf{J}_i(\mathbf{x})$ denotes the Jacobian of network output w.r.t $\mathbf{h}_i(\mathbf{x})$; $\mathbf{H}_i(\mathbf{x})$ and $\mathbf{H}_{\text{out}}(\mathbf{x})$ denotes the Hessian of the loss w.r.t $\mathbf{h}_i(\mathbf{x})$ and the network output, respectively. Then we have $\mathbf{J}_i = (\prod_{l=i}^{L} \Theta^{(l)})^T \triangleq \Theta^{[i:L]}$ the transposition of linear multiplication of weight matrices from $i$-th layer to the last one. From a detailed derivation presented in Appendix E, VSD induces an explicit regularization given by:

$$R_{VSD} = \mathbb{E}_{(\mathbf{x} \sim \mathcal{B})} \sum_{i=1}^{L} \left\langle \mathbf{H}_i, \text{diag}(\mathbf{h}_i)\mathbf{U}\text{diag}(\alpha)\mathbf{U}^T\text{diag}(\mathbf{h}_i) \right\rangle.$$

Note that, $\mathbf{H}_i$ can be approximated by $\mathbf{J}_i^T\mathbf{H}_{\text{out}}\mathbf{J}_i$ after ignoring the non-PSD term which is less important empirically [73]. This regularizer offers some intriguing but popular interpretations related

Table 3: Results for VSD and baselines on vectorized MNIST, CIFAR10 and SVHN. Results are averaged over 5 random seeds. For all metrics, lower is better.

| Method | MNIST | | | | | | CIFAR10 | | | SVHN | | |
| | FC 400x2 | | | FC 750x3 | | | CNN 32x64x128 | | | | | |
| | NLL | err. rate | ECE | NLL | err. rate | ECE | NLL | err. rate | ECE | NLL | err. rate | ECE |
|---|---|---|---|---|---|---|---|---|---|---|---|---|
| MAP | 0.098 | 1.32 | 0.011 | 0.109 | 1.27 | 0.011 | 2.847 | 34.04 | 0.272 | 0.855 | 12.26 | 0.086 |
| BBB | 0.109 | 1.59 | 0.011 | 0.140 | 1.50 | 0.013 | 1.202 | 30.11 | 0.098 | 0.545 | 10.57 | 0.017 |
| MCD | 0.049 | 1.26 | 0.007 | 0.057 | 1.22 | 0.007 | 0.794 | 26.91 | 0.024 | 0.365 | 9.23 | 0.013 |
| VD | 0.051 | 1.21 | 0.007 | 0.061 | 1.17 | 0.008 | 1.176 | 27.45 | 0.156 | 0.534 | 9.47 | 0.055 |
| ELRG | 0.053 | 1.54 | - | - | - | - | 0.871 | 29.43 | - | - | - | - |
| **VSD** | **0.042** | **1.08** | **0.006** | **0.048** | **1.09** | **0.006** | **0.730** | **24.92** | **0.020** | **0.299** | **8.39** | **0.008** |
| D.E | 0.057 | 1.29 | 0.009 | 0.063 | 1.21 | 0.009 | 1.815 | 26.44 | 0.042 | 0.783 | 9.31 | 0.070 |
| SWAG | 0.044 | 1.27 | 0.008 | **0.043** | 1.25 | 0.007 | 0.799 | 26.94 | **0.012** | 0.312 | 8.42 | 0.021 |

to the curvature of loss landscape [83, 10] (see a detailed explanation in Appendix E). We now show novel properties of VSD induced by the orthogonal matrix $\mathbf{U}$.

*VSD imposes a Tikhonov-like regularization and reshapes the gradient of network weights:* We rewrite our regularization corresponding to layer $i$-th by: $R_{VSD}^{(i)} = \mathbb{E}_{(\mathbf{x} \sim \mathcal{B})} \|\mathbf{H}_i^{1/2} \mathrm{diag}(\mathbf{h}_i) \mathbf{U} \mathrm{diag}(\alpha^{1/2})\|_F^2$. This form can be interpreted as the Tikhonov-*like* regularization imposed on the square root of Hessian matrix $\mathbf{H}_i$, in which the Tikhonov matrix $\Gamma := \mathrm{diag}(\mathbf{h}_i) \mathbf{U} \mathrm{diag}(\alpha^{1/2})$ is automatically learned during training. This principle can improve the conditioning of the estimation problem. Furthermore, when considering the case of regression problem, we have:

$$R_{VSD}^{(i)} = \mathbb{E}_{(\mathbf{x} \sim \mathcal{B})} \left[ \Theta^{[i:L]} \mathrm{diag}(\mathbf{h}_i) \mathbf{U} \mathrm{diag}(\alpha) \mathbf{U}^T \mathrm{diag}(\mathbf{h}_i) \Theta^{[i:L].T} \right]. \tag{6}$$

This is a data-dependent regularization with adaptive structure determined by the matrix $\Gamma\Gamma^T = \mathrm{diag}(\mathbf{h}_i) \mathbf{U} \mathrm{diag}(\alpha) \mathbf{U}^T \mathrm{diag}(\mathbf{h}_i)$. From the algorithmic perspective, this regularizer allows VSD to reshape the gradient of network weights according to the geometry of the data based on both scale and direction information [14, 25]. Meanwhile $\Gamma\Gamma^T = \mathrm{diag}(\mathbf{h}_i) \mathrm{diag}(\alpha) \mathrm{diag}(\mathbf{h}_i)$ only plays as a scaling factor in VD.

*VSD penalizes implicitly the spectral norm of weight matrices:* Let $\Omega_i := \mathrm{diag}(\mathbf{h}_i) \mathbf{J}_i^T \mathbf{H}_{out} \mathbf{J}_i \mathrm{diag}(\mathbf{h}_i)$, then our regularizer can be rewritten as: $R_{VSD}^{(i)} = \mathbb{E}_{(\mathbf{x} \sim \mathcal{B})} \sum_{k=1}^{K} \alpha_k^2 \mathbf{U}_{:k}^T \Omega_i \mathbf{U}_{:k}$. Since the trainable matrix $\mathbf{U}$ satisfies $\mathbf{U}_{:k}^T \mathbf{U}_{:k} = 1$ for any $k$, a penalty on $\mathbf{U}_{:k}^T \Omega_i \mathbf{U}_{:k}$ implies that VSD likely prefers a solution with smaller spectral norms of the matrix $\mathbf{H}_{out}^{1/2} \mathbf{J}_i \mathrm{diag}(\mathbf{h}_i)$ and thus of the network weights. This implication points us to the well-studied theories about generalization bound based on the spectral norm [4, 65]. Concretely, Neyshabur et al. [65] suggests that smaller spectral norm and stable rank can lead to better generalization. This expectation can be observed empirically in VSD through Table 9 in Appendix E. A more solid investigation about the generalization of VSD is of interest.

## 4 Experiments

In this section, we provide experimental evaluations to show the effectiveness of our proposed methods compared with the existing methods in terms of both predictability and scalability. We focus mainly on variational inference methods of the following two directions: the first one is direct approximations of the posterior on the random weights of Bayesian nets, including Bayes by Backprop (BBB) [8], Variational Matrix Gaussian (VMG) [47], low-rank approximations (SLANG, ELRG) [56, 80]; and the other one is the Bayesian Dropout methods with MC Dropout (MCD) [19, 20], Variational Dropout (VD) [39], and our method-Variational Structured Dropout (VSD). In addition, we evaluate the performance of point estimate framework MAP and two standard non-variational baselines Deep Ensemble (D.E) [44] and SWAG [50]. Details about data descriptions, network architectures, hyper-parameter tuning are presented in Appendix I.

### 4.1 Image classification

We now compare the predictive performance of the aforementioned methods for classification tasks on three standard image datasets: MNIST [45], CIFAR10 [43], and SVHN [64]. We evaluate the

Table 4: Image classification using AlexNet architecture. Results are averaged over 5 random seeds.

| AlexNet | CIFAR10 | | | CIFAR100 | | | SVHN | | | STL10 | | |
|---|---|---|---|---|---|---|---|---|---|---|---|---|
| | NLL | ACC | ECE | NLL | ACC | ECE | NLL | ACC | ECE | NLL | ACC | ECE |
| MAP | 1.038 | 69.58 | 0.121 | 4.705 | 40.23 | 0.393 | 0.418 | 87.56 | 0.033 | 2.532 | 65.70 | 0.267 |
| BBB | 0.994 | 65.38 | 0.062 | 2.659 | 32.41 | **0.049** | 0.476 | 87.30 | 0.094 | 1.707 | 65.46 | 0.222 |
| MCD | 0.717 | 75.22 | 0.023 | 2.503 | 42.91 | 0.151 | 0.401 | 88.03 | 0.023 | 1.059 | 63.65 | **0.052** |
| VD | 0.702 | 77.28 | **0.028** | 2.582 | 43.10 | 0.106 | 0.327 | 90.76 | 0.010 | 2.130 | 65.48 | 0.195 |
| ELRG | 0.723 | 76.87 | 0.065 | 2.368 | 42.90 | 0.099 | 0.312 | 90.66 | **0.006** | 1.088 | 59.99 | **0.018** |
| VSD | **0.656** | **78.21** | 0.046 | **2.241** | **46.85** | 0.112 | **0.290** | **91.62** | 0.008 | **1.019** | 67.98 | 0.079 |
| D.E | 0.872 | 77.56 | 0.115 | 3.402 | 46.42 | 0.314 | 0.319 | 90.30 | **0.008** | 2.229 | **68.51** | 0.241 |
| SWAG | **0.651** | **78.14** | 0.059 | **1.958** | **49.81** | **0.028** | 0.331 | 90.04 | 0.031 | 1.522 | **68.41** | 0.161 |

Table 5: Image classification using ResNet18 architecture. Results are averaged over 5 random seeds.

| ResNet18 | CIFAR10 | | | CIFAR100 | | | SVHN | | | STL10 | | |
|---|---|---|---|---|---|---|---|---|---|---|---|---|
| | NLL | ACC | ECE | NLL | ACC | ECE | NLL | ACC | ECE | NLL | ACC | ECE |
| MAP | 0.644 | 86.34 | 0.093 | 2.410 | 55.38 | 0.243 | 0.232 | 95.32 | 0.028 | 1.401 | 71.26 | 0.199 |
| BBB | 0.697 | 76.63 | 0.071 | 2.239 | 41.07 | 0.100 | 0.218 | 94.53 | 0.047 | 1.290 | 71.55 | 0.179 |
| MCD | 0.534 | 87.47 | 0.084 | 2.121 | 59.28 | 0.227 | 0.207 | 95.78 | 0.026 | 1.333 | 72.28 | 0.188 |
| VD | 0.451 | 87.68 | **0.024** | 2.888 | 56.80 | 0.284 | 0.164 | 96.11 | 0.017 | 1.084 | 73.29 | 0.084 |
| ELRG | **0.382** | 87.24 | **0.018** | 1.634 | 58.14 | **0.096** | 0.145 | 96.03 | **0.003** | 0.811 | 73.66 | **0.080** |
| VSD | 0.464 | 87.44 | 0.061 | **1.504** | 60.15 | 0.116 | **0.140** | 96.41 | 0.003 | **0.769** | **74.50** | 0.083 |
| D.E | 0.488 | **88.91** | 0.069 | 1.913 | **60.16** | 0.203 | 0.171 | **96.36** | 0.020 | 1.197 | 73.16 | 0.177 |
| SWAG | **0.330** | **88.77** | 0.026 | **1.417** | **62.45** | **0.028** | **0.130** | **96.72** | 0.016 | 0.843 | 73.15 | **0.069** |

predictive probabilities using negative log-likelihood (NLL), error rate, and expected calibration error (ECE) [61, 23]. Details on experimental settings are available in Appendix I.4.

The synthesis results of this experiment are in Table 3. On MNIST, VMG achieves err. rates of 1.17% and 1.27% with FC 400x2 and FC 750x3 respectively, while SLANG reports 1.72% err. rate with FC 400x2. In general, VSD outperforms consistently other variational methods in most settings. For D.E and SWAG, VSD exhibits competitive results on all three metrics. Especially, the figures on NLL and ECE indicate well-calibrated probabilities in our model. This is also a common but noteworthy behavior in structured approximations. On the other hand, for the remaining methods such as MAP and BBB, the error rates are worse by a large margin compared to VSD (respectively about 9% and 5% on CIFAR10, 4% and 2% on SVHN). On CIFAR10 and SVHN, these two methods and VD all show poor results on both NLL and ECE measures, implying that it will be difficult for them to reason properly about the model uncertainty especially in the out-of-distribution context. For MC Dropout, we observe a pretty good performance with the second-best result in variational methods that is similar to those reported of other works [56, 68, 80]. These results of the Bayesian Dropout methods are competitive with structured methods such as VMG, SLANG, ELRG. This further reinforces our motivation about the potential of Dropout methods for improving the predictive performance.

## 4.2 Scaling up Bayesian deep convolutional networks

We conduct additional experiments to integrate VSD into large-scale convolutional networks. We reproduce the experiments proposed in [80] (ELRG), in which we trained AlexNet and ResNet18 on 4 datasets CIFAR10, SVHN, CIFAR100 [43], and STIL10 [11] to evaluate the predictive performance of our proposal compared to other methods.

The final results are given in Table 4 and Table 5, in which the top two results will be highlighted in bold. For AlexNet, the performance of VSD is more consistent and higher than other variational methods. Modern deeper architectures facilitate deterministic estimates like MAP to better learn discriminative information extracted from training data but also makes its predictions more confident when picking excessively on unique optima. Therefore, although MAP has comparable predictive accuracy on some settings, it comes at a trade-off with the worst results on both NLL and ECE. Meanwhile, ELRG with a low-rank structure on the variational posterior gains desirable properties on uncertainty metrics. Its performance on NLL and ECE are competitive to that of VSD, however, our method obtains significant improvements on accuracy metric. For the remaining methods, BBB performs poorly on CIFAR10 and CIFAR100 in predictive accuracy, but it still has better performance than MAP in terms of NLL and ECE.

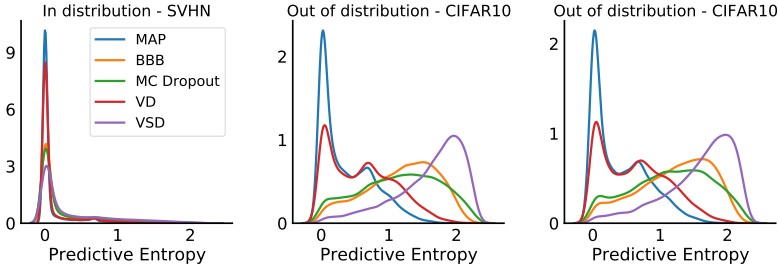

Figure 3: Histograms of predictive entropy for LeNet architecture trained on SVHN dataset.

For ResNet18 architecture, while MAP and BBB still exhibit the same behavior as mentioned above, VSD continues to achieve the convincing results, namely, it has the best performance on CIFAR100 and SVHN over all three metrics when compared to variational-based methods. On CIFAR10 and STL10, VSD also gets competitive statistics on NLL and accuracy. Overall, compared with MCD, VD, and ELRG, our method maintains good performance with better stability.

### 4.3 Predictive entropy performance

We now evaluate the predictive uncertainty of each model on out-of-distribution settings that have been implemented in previous works [44, 48, 68, 80]. We evaluate the entropy of predictive distribution $p(y^*|x^*, \mathcal{D})$ and use the density of this entropy to assess the quality of uncertainty estimates. Basically, an accurate and well-calibrated model is expected to represent entropy values being concentrated mostly around 0 (i.e. high confidence) when the test data comes from the same underlying distribution as the training data, and in the opposite case, the predictive entropies should be evenly distributed (i.e. higher uncertainty). In fact, the deep learning models do not achieve simultaneously on both expectations at the most ideal, but instead, accurate and well-calibrated ones tend to exhibit a moderate level of confidence on in-distribution data, and then provide a reasonable representation for uncertainty estimates on out-of-distribution data.

For LeNet, we train the model on SVHN dataset and then consider out-of-distribution data from CIFAR10 and CIFAR100. The results are shown in Figure 3. All methods work well on in-distribution data SVHN with the entropy value being distributed mostly around zero. However, the entropy densities of MAP and VD are concentrated excessively. This indicates that these methods would tend to make overconfident predictions on out-of-distribution data. This claim is consolidated by the qualitative results on CIFAR10 and CIFAR100 datasets. In contrast, MCD, BBB, and VSD are well-calibrated with a moderate level of confidence for in-distribution data. On CIFAR10 and CIFAR100 datasets, VSD gains better results with entropy values being distributed over a larger support, meaning that the predictions of VSD are closer to uniform on unseen classes.

We run a similar experiment, in which we train AlexNet on CIFAR10 and use SVHN, CIFAR100 as out-of-distribution data. The results are shown in the top row of Figure 4 in Appendix G.1. While MAP and VD still exhibit the same overconfident phenomenon as on LeNet, we observe the underconfident predictions of BBB and MC Dropout even on in-distribution data, which possibly leads to a high uncertainty on out-of-distribution data. We hypothesize that this is because the models trained with these methods are likely underfit with a low accuracy on the in-distribution training data. In contrast, VSD estimates reasonably the predictive entropy in both settings. The remaining scenario with ResNet18 trained on CIFAR100 is shown in the bottom row of Figure 4 with the same behaviors.

## 5 Conclusions

We proposed a novel approximate inference framework for Bayesian deep nets, named Variational Structured Dropout (VSD). In VSD, we learn a structured approximate posterior via the Dropout principle. VSD is able to yield a flexible inference while maintaining computational efficiency and scalability for deep convolutional models. The extensive experiments have evidenced the advantages of VSD such as well-calibrated prediction, better generalization, good uncertainty estimation. Given a consistent performance of VSD as presented throughout the paper, an extension of that method to other problems, such as Bayesian active learning or reinforcement learning, is of interest.

## Acknowledgements

We would like to thank the anonymous reviewers for their insightful comments and valuable suggestions. We also thank Ngo Trung Nghia (VinAI Research) for helpful discussions throughout this work. Khoat Than was funded by Gia Lam Urban Development and Investment Company Limited, Vingroup, and supported by Vingroup Innovation Foundation (VINIF) under project code VINIF.2019.DA18.

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
