# Supplement to "Structured Dropout Variational Inference for Bayesian Neural Networks"

In this supplementary material, we collect proofs and remaining materials that were deferred from the main paper. In Appendix A, we analyze the expressiveness of Variational Structured Dropout (VSD) through the approximate posterior structure and the parameterization of prior hierarchy. In Appendix B, we provide proof for the KL condition in VSD. In Appendix C, we derive in details the variational objective of VSD with hierarchical prior. Details of computational complexity of different Bayesian methods are in Appendix D. In Appendix E, we present the explicit regularization of VSD and then provide several theoretical insights induced by our regularizer. In Appendix F, we provide a new and complementary Bayesian justification for Variational Dropout methods. The next Appendices are for additional experiments of VSD. Finally, we give further discussions and investigations of VSD compared to non-variational methods.

## A    The expressiveness of Variational Structured Dropout

### A.1    The fully correlated structure of approximate posterior

A essential question is how expressive the Dropout posterior in VSD is. In VSD framework, we inject a structured noise $\xi^{(t)}$ into the deterministic weight $\Theta$, then obtain a random weight $\mathbf{W}^{(t)} = \text{diag}(\xi^{(t)})\Theta$ and an induced posterior $q_t(\mathbf{W})$. Because each scalar noise $\xi_i^{(t)}$ is shared across the row $\mathbf{W}_{i:}$ respectively, it results in a correlation on each row of $\mathbf{W}$. Meanwhile, the marginal distribution on each column $\mathbf{W}_{j:}$ is a Gaussian distribution with full covariance matrix: $q_t(\mathbf{W}_{:j}) = \mathcal{N}(\Theta_{:j}, \text{diag}(\Theta_{:j})\mathbf{U}\text{diag}(\alpha)\mathbf{U}^T\text{diag}(\Theta_{:j}))$. Therefore, the Dropout posterior in VSD has a fully correlated structure.

The correlation structure of VSD has a natural interpretation which bridges our approach to other structured approximations. Indeed, a full correlation over the whole random matrix can be parameterized separately into the correlations among the rows and columns of that matrix, which implicitly affects the correlations among the input and output hidden units. Interestingly, this connection can be exhibited explicitly in our method. When performing the forward pass, Dropout procedure will generally introduce the correlations between elements at pre-activation layer, namely output hidden units. In addition, the structured noise in our method can be considered as an auxiliary variable, which has the ability to captures the correlations among neurons of input hidden units, or otherwise, it will encourage neurons to borrow statistical strength from one another through variational learning. On the other hand, we know that due to the statistical noise, the correlations among hidden units appear naturally, and this therefore rationalizes the original proposal in our paper.

### A.2    The role of hierarchical prior

In Bayesian inference, the prior distribution plays an important role in representing the capacity of Bayesian neural networks. By appropriately employing the informative priors, we can significantly improve the predictive performance [84, 15]. This distribution also allows incorporating external domain knowledge or specific properties such as feature sparsity, into the Bayesian deep models [12]. We derive here some detailed discussions about the role of hierarchical prior particularly in our method.

**Gaussian scale mixture prior and mixture approximate posterior**. The well-known property of expanding a model hierarchically is that it induces new dependencies between the data, either through shrinkage or an explicitly correlated prior [16]. The hierarchical representation in our method is a *center parameterization*, and by integrating out the latent variable $\mathbf{z}$, we obtain a marginal prior distribution as follows:

$$p(\mathbf{W}_{:j}|\beta_{:j}, \tau) = \int \mathcal{N}(\mathbf{W}_{:j}|0, \text{diag}(\mathbf{z} \odot \beta_{:j}^{-1})p(\mathbf{z}|\tau)d\mathbf{z}, \tag{7}$$

where $p(\mathbf{z}|\tau)$ is treated as the mixing distribution. The equation (7) can also be written in an equivalent *expanded* or *non-centered parameterization* as: $\mathbf{W}_{:j} = \gamma \odot \mathbf{z}$ with $\gamma \sim \mathcal{N}(0, \text{diag}(\beta_{:j}^{-1}))$ and $\mathbf{z} \sim p(\mathbf{z}|\tau)$. This prior family have been widely used in BNN literature [46, 21, 66, 15]. By

approximating the above integral with Monte Carlo sampling $\mathbf{z}^{(i)} \sim p(\mathbf{z}|\eta)$, we can resemble an informative prior known as Gaussian scale mixtures (GSM) [8, 12] with the following term:

$$p(\mathbf{W}_{:j}|\beta_{:j}, \tau) \approx \frac{1}{M} \sum_{i=1}^{M} \mathcal{N}(\mathbf{W}_{:j}|0, \text{diag}(\mathbf{z}^{(i)} \odot \beta_{:j}^{-1})). \tag{8}$$

Therefore, our hierarchical framework can provide an appealing connection between multiplicative Gaussian noise with GSM priors. This interpretation is close to the recent work in [62], in which the authors show that multiplicative noise is equivalent to the structured shrinkage prior and interestingly, MC Dropout objective is a lower bound on the scale mixture model's marginal MAP objective.

Instead of investigating extensively the expressiveness of this prior through different parameterizations, in the scope of this work, we focus on the advantages of the joint inference that can make a mixture approximation in the variational objective function:

$$\mathbb{E}_{q_\psi(\mathbf{z})q_t(\mathbf{W}|\mathbf{z})} \log p(\mathcal{D}|\mathbf{W}) \approx \frac{1}{M} \sum_{i=1}^{M} \mathbb{E}_{q_t(\mathbf{W}|\mathbf{z}^{(i)})} \log p(\mathcal{D}|\mathbf{W}).$$

This approximation is equivalent to leveraging a mixture of structured covariance posterior, which has a reasonable potential to recover the multimodality of the true Bayesian posterior. Moreover, this mixture is also practical in the high dimensional setting of Bayesian deep models, because it does not require the additional computational cost from multiplying the components. We also suggest that hierarchical parameterization with joint inference is a promising approach to improve the predictive performance of Bayesian deep models, especially compared with non-Bayesian methods such as Deep Ensemble, which is evidenced by a recent work in [15].

**Hierarchical prior imposes a global stochastic noise and enforces a stronger regularization.** At each iteration of the training process, while we draw $\mathbf{W}$ from the conditional posterior $q_t(\mathbf{W}|\mathbf{z})$ separately for each data as the Dropout procedure, the latent $\mathbf{z} \sim q_\psi(\mathbf{z})$ is shared across the entire data batch. This means for each data input $\mathbf{x}_n$, we introduce a joint-structured noise $\widehat{\xi}_n^{(t)}$ with the following form:

$$\mathbf{z} \sim q_\psi(\mathbf{z}), \quad \xi_n^{(t)} \sim \mathcal{N}(\mathbf{1}_K, \mathbf{U}\text{diag}(\alpha)\mathbf{U}^T), \tag{9}$$
$$\widehat{\xi}_n^{(t)} = \mathbf{z} \odot \xi_n^{(t)}, \quad \mathbf{W} = \text{diag}(\widehat{\xi}_n^{(t)})\Theta.$$

The above reinterpretation demonstrates that by the Monte Carlo estimation, the joint variational inference with hierarchical prior in our method adapts to the vanilla Dropout procedure. The new representation of Dropout noise allows our method to regularize each unit layer with different levels of stochasticity. The latent $\mathbf{z}$ under this representation can be considered as a global variational noise and by learning its variational distribution $q_\psi(\mathbf{z})$, we can capture correlation characteristics of input samples in each data batch. Furthermore, in our prior hierarchy, whilst the hyperparameter $\beta$ has the same size with the network weight $\mathbf{W}$ that might induce a relatively poor regularization, the latent $\mathbf{z}$ is designed with the size of the number of rows and is shared across columns of the matrix $\mathbf{W}$. This row-partitioning will discourage allowing too many degrees of freedom in the parameterization. Basically, such a technique applied to variational Bayesian inference will enforce a stronger regularization in the objective and then prevent the model from the overfitting issue.

## B  The KL condition in Variational Structured Dropout

The alternative variational objective of Variational Structured Dropout (VSD) is given as follows:

$$\mathcal{L}(\alpha, \Theta, .) = \mathbb{E}_{q_\phi(\mathbf{W})} \log p(\mathcal{D}|\mathbf{W}^{(t)}) - \mathbb{D}_{KL}(q_t^\star(\mathbf{W})||p(\mathbf{W}))$$
$$= \mathbb{E}_{q_\alpha(\xi)} \log p(\mathcal{D}|\Theta, \xi^{(t)}) - \sum_{j=1}^{Q} \mathbb{D}_{KL}(q_t(\mathbf{W}_{:j})||p(\mathbf{W}_{:j})). \tag{10}$$

By applying Monte Carlo sampling combined with the reparameterization trick to approximate the expected log-likelihood, we perform a procedure being equivalent to injecting a structured noise $\xi^{(t)}$ into the model parameters $\Theta$. However, to make this Monte Carlo estimation follows the Dropout

training procedure, we separately draw one realization $\xi_n^{(t)}$ for each data point $(\mathbf{x}_n, \mathbf{y}_n)$. This can be interpreted as arising from the local reparameterization trick [39], in which the global parameter uncertainty is translated backward into local unit uncertainty at the pre-linear layer instead of the post-linear one. Finally, to ensure the KL condition, we will specify the KL term in the form independent of $\Theta$. Then, the above lower bound recovers the Dropout objective function.

We have the Dropout posterior and the prior determined on the $j$-th column of $\mathbf{W}$, respectively:

$$q_t(\mathbf{W}_{:j}) = \mathcal{N}(\Theta_{:j}, \mathrm{diag}(\Theta_{:j})\mathbf{U}\mathrm{diag}(\alpha)\mathbf{U}^T\mathrm{diag}(\Theta_{:j})) \quad \text{and} \quad p(\mathbf{W}_{:j}|\beta) = \mathcal{N}(0, \mathrm{diag}(\beta_{:j}^{-1})).$$

Let $\boldsymbol{\mu}_1 = \Theta_{:j}$, $\boldsymbol{\Sigma}_1 = \mathrm{diag}(\Theta_{:j})\mathbf{U}\mathrm{diag}(\alpha)\mathbf{U}^T\mathrm{diag}(\Theta_{:j})$ and $\boldsymbol{\mu}_2 = 0$, $\boldsymbol{\Sigma}_2 = \mathrm{diag}(\beta_{:j}^{-1})$. The KL divergence can then be calculated as follows:

$$\mathbb{D}_{KL}(q_t(\mathbf{W}_{:j})||p(\mathbf{W}_{:j})) = \frac{1}{2}\left[\log\frac{|\boldsymbol{\Sigma}_2|}{|\boldsymbol{\Sigma}_1|} - K + \mathrm{Trace}(\boldsymbol{\Sigma}_2^{-1}\boldsymbol{\Sigma}_1) + (\boldsymbol{\mu}_2 - \boldsymbol{\mu}_1)^T\boldsymbol{\Sigma}_2^{-1}(\boldsymbol{\mu}_2 - \boldsymbol{\mu}_1)\right].$$

Since $\mathbf{U}$ is a orthogonal matrix, we have:

$$\log\frac{|\boldsymbol{\Sigma}_2|}{|\boldsymbol{\Sigma}_1|} = -\sum_{i=1}^{K}\log\beta_{ij} - \sum_{i=1}^{K}\log\alpha_i\Theta_{ij}^2,$$

$$\mathrm{Trace}(\boldsymbol{\Sigma}_2^{-1}\boldsymbol{\Sigma}_1) = \mathrm{Trace}(\mathrm{diag}(\beta_{:j}\odot\Theta_{:j}^2)\mathbf{U}\mathrm{diag}(\alpha)\mathbf{U}^T) = \sum_{i=1}^{K}\beta_{ij}\Theta_{ij}^2\left(\sum_{j=1}^{K}\alpha_j U_{ij}^2\right),$$

$$(\boldsymbol{\mu}_2 - \boldsymbol{\mu}_1)^T\boldsymbol{\Sigma}_2^{-1}(\boldsymbol{\mu}_2 - \boldsymbol{\mu}_1) = \Theta_{:j}^T\mathrm{diag}(\beta_{:j})\Theta_{:j} = \sum_{i=1}^{K}\beta_{ij}\Theta_{ij}^2.$$

Given the above equations, the KL term on the $j$-th column of $\mathbf{W}$ can be rewritten by:

$$\mathbb{D}_{KL}(q_t(\mathbf{W}_{:j})||p(\mathbf{W}_{:j})) = \frac{1}{2}\sum_{i=1}^{K}\left[-\log\beta_{ij} - \log\alpha_i\Theta_{ij}^2 - 1 + \beta_{ij}\Theta_{ij}^2\left(1 + \sum_{j=1}^{K}\alpha_j U_{ij}^2\right)\right]. \tag{11}$$

We can find that the orthogonality of Householder transformations facilitates the tractable calculation for the KL term without complicated analyses. Next, we choose the prior hyper-parameter $\beta$ via the Empirical Bayes approach which is achieved by optimizing $\beta$ based upon the data. More specifically, taking the partial derivative of the RHS in equation (11) with respect to $\beta$, we get:

$$\frac{\partial\mathbb{D}_{KL}}{\partial\beta_{ij}} = \frac{1}{2}\left[-\frac{1}{\beta_{ij}} + \Theta_{ij}^2\left(1 + \sum_{j=1}^{K}\alpha_j U_{ij}^2\right)\right].$$

Letting this derivative to zero, we obtain the optimal value for $\beta$ in the analytical form: $\beta_{ij}^* = 1/\left(\Theta_{ij}^2(1 + \sum_{j=1}^{K}\alpha_j U_{ij}^2)\right)$. Substitute this value in the expression of the KL term, we get the form independent of the weight parameter $\Theta$ as follows:

$$\mathbb{D}_{KL}^{EB}(q_t(\mathbf{W}_{:j})||p(\mathbf{W}_{:j})) = \frac{1}{2}\sum_{i=1}^{K}\log\frac{1 + \sum_{j=1}^{K}\alpha_j U_{ij}^2}{\alpha_i}. \tag{12}$$

As a consequence, we obtain the conclusion in the main text about the KL condition in VSD.

**Discussion of the effect of the Empirical Bayes in our method**. The Empirical Bayes procedure presented above is equivalent to an iterative optimization algorithm for our objective, in which $\beta$ will be updated until convergence after every single update of other parameters. However, disappearing this precision parameter by directly substituting its Empirical Bayes values would help to clarify the KL-condition guarantee in VSD as analyzed. The data-dependent choice of this parameter is made explicitly through the dependence on the learned droprate $\alpha$, the matrix $U$, and the deterministic weight $\Theta$. In the literature, several studies have also offered interesting perspectives for the Empirical Bayes. Generally, this procedure usually suffers from the main criticism of using data twice that is

illegal in a strict Bayesian formalism. And particularly in mean-field BNNs, the Empirical Bayes was claimed to be able to yield slow convergence, introduce strange local minima and thus lead to miscalibrated predictive distributions [8]. However, there should be a more comprehensive study to investigate the effects of this procedure, especially in a complicated context of deep learning models, which are surrounded by other data-related techniques such as temperature scaling, data augmentation, and even Dropout - a data-dependent regularization. On the other hand, Empirical Bayes has been embraced and widely adopted in Bayesian machine learning, and especially in the seminal work on Bayesian neural nets. This technique has been employed as a principled approach to learning prior hyperparameters [87, 42], or embodying automatic-relevance determination [36].

## C    The derivation of the variational objective with hierarchical prior

In Dropout approximate inference, the KL condition restricts the scope of prior distribution family. Our works has overcome this limitation by proposing a unified framework using varational structured Dropout combined with hierarchical prior, in which we guarantees the KL condition without any simplifying assumptions about the prior family. Concretely, with our proposed prior hierarchy, we maximize a variational lower bound from the joint variational inference as follows:

$$\mathcal{L}(\alpha, \Theta, \psi, .) = \mathbb{E}_{q_\psi(\mathbf{z})q_t(\mathbf{W}|\mathbf{z})} \log p(\mathcal{D}|\mathbf{W})$$
$$- \mathbb{E}_{q_\psi(\mathbf{z})}(\mathbb{D}_{KL}(q_t^\star(\mathbf{W}|\mathbf{z}, \phi)||p(\mathbf{W}|\mathbf{z}, \beta)) - \mathbb{D}_{KL}(q_\psi(\mathbf{z})||p(\mathbf{z})). \quad (13)$$

For the latent variable $\mathbf{z}$, we choose the prior $p(\mathbf{z}|\eta)$ and the variational distribtuion $q(\mathbf{z})$ as (inverse) Gamma$(a, b)$ and log-Normal$(\gamma, \delta)$ distribution respectively. These distributions have positive support and can be reparametrized. The KL-divergence between them also has a closed-form expression, which is given by:

$$\mathbb{D}_{KL}(q_\psi(\mathbf{z})||p(\mathbf{z})) = a \log b - \log \Gamma(a) - a\gamma - \beta \exp(-\gamma + 0.5\delta) + 0.5(\log \delta + 1 + \log(2\pi)).$$

For the KL-divergence between the conditional posterior $q_t^\star(\mathbf{W}|\mathbf{z}, \phi)$ and the conditional prior $p(\mathbf{W}|\mathbf{z}, \beta)$, similarly, we need a form that does not depend on the deterministic weight $\Theta$. Since:

$$q_t(\mathbf{W}_{:j}|\mathbf{z}) = \mathcal{N}(\mathbf{z} \odot \Theta_{:j}, \mathbf{V}_j \mathbf{U}\text{diag}(\alpha)(\mathbf{V}_j\mathbf{U})^T) \quad \text{and} \quad p(\mathbf{W}_{:j}|\mathbf{z}, \beta_{:j}) = \mathcal{N}(0, \text{diag}(\mathbf{z} \odot \beta_{:j}^{-1}))$$

where $\mathbf{V}_j = \text{diag}(\mathbf{z} \odot \Theta_{:j})$, similar to the analysis in the previous section, we have:

$$\mathbb{D}_{KL}(q_t^\star(\mathbf{W}_{:j}|\mathbf{z}, \phi)||p(\mathbf{W}_{:j}|\mathbf{z}, \beta_{:j})) =$$
$$\frac{1}{2}\sum_{i=1}^{K}\left[ -\log z_i - 1 - \log \beta_{ij} - \log \alpha_i \Theta_{ij}^2 + \beta_{ij} z_i \Theta_{ij}^2 \left(1 + \sum_{j=1}^{K} \alpha_j U_{ij}^2\right)\right].$$

Because $\beta$ is referred to as the scaling factor, we can choose it by: $\beta_{ij}^* = 1/\left(\Theta_{ij}^2(1 + \sum_{j=1}^{K} \alpha_j U_{ij}^2)\right)$. The above KL then can be rewritten in the following form:

$$\mathbb{D}_{KL}^{EB}(q_t^\star(\mathbf{W}_{:j}|\mathbf{z}, \phi)||p(\mathbf{W}_{:j}|\mathbf{z}, \beta_{:j})) = \frac{1}{2}\sum_{i=1}^{K}\left[ z_i - \log z_i - 1 - \log \frac{1 + \sum_{j=1}^{K} \alpha_j U_{ij}^2}{\alpha_i}\right]. \quad (14)$$

As a result, this form satisfies the KL condition.

## D    Computational complexity and low-rank approximation

We describe here in detail the computational complexity of the different algorithms, in which the computation is considered when performing a forward pass through a single layer of the network. We also discuss the memory usage while constructing the dynamic computation graph. To ease the presentation, we briefly recall the abbreviations of methods from the main text, in particular: Bayes by Backprop (BBB) [8], Variational Matrix Gaussian (VMG) [47], Multiplicative Normalizing Flow (MNF) [48], low-rank approximations (SLANG, ELRG) [56, 80]; and the Bayesian Dropout methods including MC Dropout (MCD) [19, 20], Variational Dropout (VD) [39].

**Computational complexity.** Assume the weight matrix of the layer is of size $K \times L$, in which $K$

Table 6: Computational complexity per layer of MAP and different variational methods.

| Method | Time | Memory |
|---|---|---|
| MAP | $\mathcal{O}(KL\|\mathcal{B}\|)$ | $\mathcal{O}(L\|\mathcal{B}\|)$ |
| BBB | $\mathcal{O}(sKL\|\mathcal{B}\|)$ | $\mathcal{O}(sKL + L\|\mathcal{B}\|)$ |
| BBB-LTR | $\mathcal{O}(2KL\|\mathcal{B}\|)$ | $\mathcal{O}(2L\|\mathcal{B}\|)$ |
| VMG | $\mathcal{O}(m^3 + 2KL\|\mathcal{B}\|)$ | $\mathcal{O}(KL\|\mathcal{B}\|)$ |
| SLANG | $\mathcal{O}(r^2KL + rsKL\|\mathcal{B}\|)$ | $\mathcal{O}(rKL + sKL\|\mathcal{B}\|)$ |
| ELRG | $\mathcal{O}(r^3 + (r+2)KL\|\mathcal{B}\|)$ | $\mathcal{O}((r+2)L\|\mathcal{B}\|)$ |
| **VSD** | $\mathcal{O}(K^2 + KL\|\mathcal{B}\|)$ | $\mathcal{O}(K^2 + K\|\mathcal{B}\|)$ |
| **VSD-low rank** | $\mathcal{O}(rK + KL\|\mathcal{B}\|)$ | $\mathcal{O}(K^2 + K\|\mathcal{B}\|)$ |

Table 7: Computation time of variational methods compared to standard MAP (1x).

| Methods | Time/epoch (s) | | |
|---|---|---|---|
| | LeNet5 | AlexNet | ResNet18 |
| BBB-LTR | 1.53x | 1.75x | 3.28x |
| MNF | 2.86x | 3.40x | 4.88x |
| VD | 1.18x | 1.15x | 1.32x |
| VSD $T=1$ | 1.25x | 1.32x | 1.86x |
| VSD $T=2$ | 1.35x | 1.49x | 2.90x |
| time-scaling | 1.08 | 1.13 | 1.56 |

denotes the number of rows in fully connected layer or the number of channels in convolutional layer, respectively. First, MAP estimation performs a matrix multiplication with time cost $K \times L$ to forward each input $\mathbf{x}_i$ of size $K$ in data batch $\mathcal{B}$. MAP estimation needs to store the output of these calculations which gives a memory cost $L\|\mathcal{B}\|$. Next, BBB with naive reparameterization trick, in practice, needs to use $s \geq 2$ sampled weights of dimension $K \times L$ to reduce the variance of gradient estimator. This makes the computation hard to be performed in parallel, thus incurs multiple costs of both time and memory with $\mathcal{O}(sKL\|\mathcal{B}\|)$ and $\mathcal{O}(sKL + L\|\mathcal{B}\|)$ respectively. On the other hand, with the local reparameterization trick that translates uncertainty about global random weights into local noise in pre-activation unit, BBB can gain an alternative unbiased estimator with low variance while maintaining low complexity via sampling only a local noise. However, it requires two forward passes to obtain means and variances of the pre-activation. For VMF, SLANG, and ELRG, the detailed analysis can be found on the original papers, and note that SLANG is a method that fails to employ the local reparameterization trick, thereby leading to very high complexity on both time and memory.

VSD adopts the advantage of Dropout training (VD) via just sampling the low dimension noise instead of whole random weights. An additional benefit is that VSD only requires one forward pass in parallel compared with two steps of the local reparameterization trick. When using a fully connected layer (FC) size of $K \times K$ to parameterize the Householder vector, namely:

$$\mathbf{v}_t = \mathbf{FC}(\mathbf{v}_{t-1}), \qquad \mathbf{S}_t = \left(\mathbf{I} - 2\frac{\mathbf{v}_t \mathbf{v}_t^T}{\|\mathbf{v}_t\|_2^2}\right)\mathbf{S}_{t-1} = \mathbf{H}_t\mathbf{S}_{t-1},$$

for $t = 1, ..., T$, it will induce a complexity of $\mathcal{O}(K^2)$ to our method in terms of both time and memory cost. However, we also reduce the number of parameters of this FC by adding a low dimensional hidden layer. This simple solution results in lower computational time of $\mathcal{O}(rK + KL\|\mathcal{B}\|)$ without sacrificing much the performance (see Table 8 in Appendix D). In general, VSD has shown better computational efficiency than other structured approximation methods, even more practical than the mean-field BBNs.

Note that, taking the advantage of low dimensional space to enrich the quality of approximation is an appealing idea. The recent advances in variational inference have offered many modern techniques to exploit this idea, such as normalizing flow [70, 38, 5], auxiliary random variable, implicit distribution [69, 88], or mixture approximation [94, 2]. Nevertheless, the novelty depends on the sophistication when applied to specific models with certain constraints. More specifically, in the context of the problem we aim for, applying these above techniques to Bayesian Dropout frameworks requires dealing with some challenges including the difficulty of parallel backpropagation, high computational complexity, and more importantly, how to ensure the KL condition. The Householder parameterization helps our approach address these challenges in both theory and applicability. Moreover, we can extend our method to other parameterizations, for example Sylvester-based flows [5], as long as the orthogonality of matrix $\mathbf{U}$ is preserved.

**Practical runtime.** We show in Table 7 the empirical computation time of VSD and some other methods, in which BBB-LTR and MNF are *direct* approximatitons of BNNs, while VD and VSD represent Dropout inference frameworks. BBB-LTR and VD maintain a mean-field structure for the approximation, while MNF and VSD share the same intuition of enriching the variational approximate distribution via low dimensional space. However, we remark that there are some key considerations that distinguish VSD from MNF including: (1) MNF facilitates flexible approximation via normalizing flows (NFs) which is much more expensive compared with the orthogonal parameterizations of VSD, MNF even used two sequences of NFs to tighten the variational lower bound; (2) VSD exploits

Table 8: The performance of VSD when using low-rank approximation, where $r$ is the dimension of hidden unit, $T = 2$ is the number of Householder transformations. Random seed $= 1$. For all metrics, lower is better.

| $T = 2$ | MNIST | | | CIFAR10 | | | SVHN | | |
| | FC 750x3 | | | CNN 32x64x128 | | | | | |
| | NLL | err. rate | ECE | NLL | err. rate | ECE | NLL | err. rate | ECE |
| --- | --- | --- | --- | --- | --- | --- | --- | --- | --- |
| $r = 2$ | 0.049 | **1.12** | 0.007 | 0.7298 | 25.45 | **0.022** | **0.3007** | **8.36** | 0.008 |
| $r = 5$ | 0.046 | 1.15 | 0.006 | **0.7199** | **24.91** | 0.023 | 0.3024 | **8.36** | 0.009 |
| $r = 10$ | 0.049 | 1.15 | 0.007 | 0.7365 | 25.24 | 0.024 | 0.3048 | 8.47 | 0.009 |
| full rank | 0.045 | 1.13 | 0.006 | 0.7297 | 25.18 | 0.023 | 0.3021 | 8.41 | 0.008 |

Bayesian hierarchical modeling for the Dropout inference framework and then learns a joint posterior, while MNF adopts a implicit-marginal distribution for approximate weight posterior. The settings we use to implement MNF are given in the original paper [48]. Going back to Table 7, VSD with $T = 2$ exhibits extra computation time compared to $T = 1$, the increase on ResNet18 is more evident than on LeNet and AlexNet (see the time-scaling values). This is because the quantities $K$ of ResNet18 are larger than that of LeNet and AlexNet. However, on more modern architectures such as PreResNet110 which prefers to evolve in depth rather than width (namely using fewer channels), VSD with $T = 2$ should not endure much extra computation time and thus would make a good adaptation. Indeed, we verify this argument by measuring the computation time of VSD trained with PreResNet110 on CIFAR10 dataset, from which the figures obtained for $T = 1$ and $T = 2$ are 2.68x and 3.60x, then the corresponding time-scaling value is 1.34.

**Low-rank approximation for VSD.** We investigate a *low-rank* structure in the fully connected layer used to parameterize the Householder vectors in our method. Instead of using one layer with full size $K \times K$, we add a low dimensional hidden layer with ReLU activation. The size of this hidden layer is $r \in \{2, 5, 10\}$. This idea is quite natural because it reduces significantly the number of parameters in our method while ensuring flexible parametrization for the Householder vectors thanks to the nonlinearity in hidden activation.

We show the performance of VSD with low-rank approximation in Table 8, where we repeat the experiment on image classification in Section 4.1 of our main paper. We can see that although the rank $r$ is very small, the decrease in performance is negligible (still outperforms the baselines). This natural idea even improves the results on some settings such as the ECE metric in SVHN dataset.

# E The derivation of the induced regularization of VSD

In this appendix, we present the explicit regularization induced by the structured noise in our framework. Let $\mathcal{X} \subseteq \mathbb{R}^{d_{\text{in}}}$ and $\mathcal{Y} \subseteq \mathbb{R}^{d_{\text{out}}}$ denote the input and output spaces, respectively. We consider a deep linear neural net $f : \mathcal{X} \to \mathbb{R}^{d_{\text{out}}}$ with $L$ layers parameterized by $\{\Theta^{(i)}\}_{i=1}^{L}$, and then define a corresponding surrogate loss $\ell : \mathbb{R}^{d_{\text{out}}} \times \mathcal{Y} \to \mathbb{R}$. The goal of learning is to find a hypothesis $f$ that minimizes the population risk: $L_0 := \mathbb{E}_{(\mathbf{x}, \mathbf{y}) \sim \mathcal{X} \times \mathcal{Y}} \ell(f(\mathbf{x}), \mathbf{y})$. The expected term can be written when we instead optimize the empirical risk using data batch $\mathcal{B}$ as follows: $\widehat{L} := \mathbb{E}_{(\mathbf{x}, \mathbf{y}) \sim \mathcal{B}} \ell(f(\mathbf{x}), \mathbf{y})$. For convenience, we will ignore output $\mathbf{y}$ in the subsequent analyses.

Note that, while the overall function is linear, the representation in factored form makes the optimization landscape non-convex and hence, challenging to analyze. For simplicity, we start by considering Dropout applied to a single layer $i$ of the network and define some detailed notations as: $\mathbf{h}_i$ is the $i$-th hidden layer; $f_i(.)$ denotes the composition of the layers after $\mathbf{h}_i$, that means $f_i(\mathbf{h}_i(\mathbf{x})) = f(\mathbf{x})$; $\mathbf{J}_i(\mathbf{x})$ denotes the Jacobian of network output w.r.t $\mathbf{h}_i(\mathbf{x})$; $\mathbf{H}_i(\mathbf{x})$ and $\mathbf{H}_{\text{out}}(\mathbf{x})$ denotes the Hessian of the loss w.r.t $\mathbf{h}_i(\mathbf{x})$ and the network output, respectively. Then we have $\mathbf{J}_i = (\prod_{l=i}^{L} \Theta^{(l)})^T \triangleq \Theta^{[i:L]}$ the transposition of linear multiplication of weight matrices from $i$-th layer to the last one. Then, we define the loss function with multiplicative structured Dropout by:

$$\widehat{L}_{\text{drop}} := \mathbb{E}_{\mathbf{x} \sim \mathcal{B}} \mathbb{E}_{\xi^{(t)}} \ell(f(\mathbf{x} \odot \xi^{(t)})),$$

where $\xi^{(t)} = \{\xi^{(t,i)}\}_{i=1}^{L}$ with the noise variable $\xi^{(t,i)}$ is specified to the $i$-th layer respectively (the definition of $\xi^{(t)}$ here is a bit different from that in the main text, but it is clear from the context).

Then we define an explicit regularizer induced by Dropout as follows:

$$R_{VSD} := \widehat{L}_{\text{drop}} - \widehat{L} = \frac{1}{L} \sum_{i=1}^{L} \mathbb{E}_{\mathbf{x} \sim \mathcal{B}} \left[ \mathbb{E}_{\xi^{(t,i)}} \ell(f(\mathbf{x} \odot \xi^{(t,i)})) - \ell(f(\mathbf{x})) \right].$$

Let $\xi^{(t,i)} = 1 + \eta^{(t,i)}$, we then rewrite the loss after applying dropout on $\mathbf{h}_i$ by: $\ell(f(\mathbf{x} \odot \xi^{(t,i)})) = \ell(f_i(\mathbf{h}_i(\mathbf{x}) + \delta^{(t,i)}))$ with $\delta^{(t,i)} = \mathbf{h}_i(\mathbf{x}) \odot \eta^{(t,i)}$. To analyze the effect of this perturbation, we apply the Taylor expansion around $\delta^{(t,i)} = \vec{0}$ as follows:

$$\ell(f(\mathbf{x} \odot \xi^{(t,i)})) - \ell(f(\mathbf{x})) \approx D_{\mathbf{h}_i}(\ell \circ f_i)[\mathbf{h}_i(\mathbf{x})]\delta^{(t,i)} + \delta^{(t,i)T} D_{\mathbf{h}_i}^2(\ell \circ f_i)[\mathbf{h}_i(\mathbf{x})]\delta^{(t,i)},$$

where $\mathcal{D}_{(.)}$ denotes the derivative operator. Take the expectations on both sides of the above equation with the note that $\delta^{(t,i)}$ is a zero-mean random variable, we obtain an approximation of the Dropout explicit regularizer in VSD corresponding to $i$-th layer:

$$R_{VSD}^{(i)} := \mathbb{E}_{\mathbf{x} \sim \mathcal{B}} \mathbb{E}_{\delta^{(t,i)}} \left[ \delta^{(t,i)T} D_{\mathbf{h}_i}^2(\ell \circ f_i)[\mathbf{h}_i(\mathbf{x})]\delta^{(t,i)} \right]$$

$$= \mathbb{E}_{\mathbf{x} \sim \mathcal{B}} \left\langle D_{\mathbf{h}_i}^2(\ell \circ f_i)[\mathbf{h}_i(\mathbf{x})], \mathbb{E}_{\delta^{(t,i)}}[\delta^{(t,i)}\delta^{(t,i)T}] \right\rangle$$

$$= \mathbb{E}_{\mathbf{x} \sim \mathcal{B}} \left\langle \mathbf{H}_i(\mathbf{x}), \mathbb{E}_{\delta^{(t,i)}}[\delta^{(t,i)}\delta^{(t,i)T}] \right\rangle.$$

Because $\delta^{(t,i)} = \mathbf{h}_i(\mathbf{x}) \odot \eta^{(t,i)}$ and $\eta^{(t,i)} \sim \mathcal{N}(0, \mathbf{U}\text{diag}(\alpha)\mathbf{U}^T)$, we have:

$$\mathbb{E}_{\delta^{(t,i)}}[\delta^{(t,i)}\delta^{(t,i)T}] = \mathbb{V}_{\delta^{(t,i)}}[\delta^{(t,i)}] = \text{diag}(\mathbf{h}_i(\mathbf{x}))\mathbf{U}\text{diag}(\alpha)\mathbf{U}^T\text{diag}(\mathbf{h}_i(\mathbf{x})).$$

Meanwhile, for the term $\mathbf{H}_i(\mathbf{x}) = D_{\mathbf{h}_i}^2(\ell \circ f_i)[\mathbf{h}_i(\mathbf{x})]$, we can decompose into two components as:

$$D_{\mathbf{h}_i}^2(\ell \circ f_i)[\mathbf{h}_i(\mathbf{x})] = \mathbf{J}_i^T(\mathbf{x})\mathbf{H}_{\text{out}}(\mathbf{x})\mathbf{J}_i(\mathbf{x}) + \sum_j (D_f\ell[f(\mathbf{x})])_j D_{\mathbf{h}_i}^2(f_i)_j[\mathbf{h}_i(\mathbf{x})], \quad (15)$$

where $j$ indicates the $j$-th coordinate in the output. The first term in the RHS of equation (15) is positive-semidefinite (when $\ell$ is the mean-square error or the cross-entropy loss), but the second term is likely to be non positive-semidefinite since it involves the Hessian of a non-convex model. However, [73] suggests ignoring this quantity because it is less important empirically. Then, our regularizer can be approximated by:

$$R_{VSD}^{(i)} = \mathbb{E}_{\mathbf{x} \sim \mathcal{B}} \left\langle \mathbf{H}_i(\mathbf{x}), \text{diag}(\mathbf{h}_i(\mathbf{x}))\mathbf{U}\text{diag}(\alpha)\mathbf{U}^T\text{diag}(\mathbf{h}_i(\mathbf{x})) \right\rangle \quad (16)$$

$$\approx \mathbb{E}_{\mathbf{x} \sim \mathcal{B}} \left\langle \mathbf{J}_i^T(\mathbf{x})\mathbf{H}_{\text{out}}(\mathbf{x})\mathbf{J}_i(\mathbf{x}), \text{diag}(\mathbf{h}_i(\mathbf{x}))\mathbf{U}\text{diag}(\alpha)\mathbf{U}^T\text{diag}(\mathbf{h}_i(\mathbf{x})) \right\rangle, \quad (17)$$

which fulfills the criteria for a valid regulariser.

We next will derive some variants of our regularizer and from which provide several interpretations of the algorithmic aspects. To simplify the presentation, we denote the matrix $\Gamma := \text{diag}(\mathbf{h}_i)\mathbf{U}\text{diag}(\alpha^{1/2})$ and the matrix $\mathbf{Q} := \Gamma\Gamma^T = \text{diag}(\mathbf{h}_i(\mathbf{x}))\mathbf{U}\text{diag}(\alpha)\mathbf{U}^T\text{diag}(\mathbf{h}_i(\mathbf{x}))$.

**Interpretation 1:** *Dropout regularizer encourages the flatness of local minima.* In deep linear network, the second derivatives of the loss w.r.t the hidden unit and the weight parameter are tightly correlated to each other. Indeed, assume $\mathbf{Z}$ is a weight matrix following hidden layer $\mathbf{h}_i$ in the network architecture, namely $f(\mathbf{x}) = f_i(\mathbf{h}_i(\mathbf{x})) = f_{i+1}(\mathbf{Z}\mathbf{h}_i(\mathbf{x}))$, then we have:

$$D_{\mathbf{Z}}(\ell(f(\mathbf{x}))[\mathbf{Z}] = \mathbf{h}_i(\mathbf{x})D_{\mathbf{h}_{i+1}}(\ell \circ f_i)[\mathbf{h}_{i+1}(\mathbf{x})].$$

Thus, the loss derivatives w.r.t weight parameters can be expressed in terms of those w.r.t the hidden layers. For ReLU-like activations such as ELU, Softplus, our functions are at most linear, so this above equation still holds. Therefore, when the regularization of a deep linear network is measured by Hessian matrix $\mathbf{H}_i$, it will tend to penalize the magnitudes of the eigenvalues of the second derivative of the loss w.r.t the weight parameters, intuitively leading to small curvature of the corresponding local loss landscape. This helps the network converges to the flat minima which contributes to better generalization. We can also analyze this property directly from equation (17), specifically we penalize the Jacobian based on the norm of $\mathbf{H}_{\text{out}}$. This enables the learning algorithm to maintain a low empirical Lipschitz constant, and so facilitates the optimizer to exploit flat regions of the loss function.

**Interpretation 2:** *The structured Dropout regularizer adapts a Tikhonov-like regularization and reshapes the gradient of network weights.* From equation (16) and the relation between trace operator and inner product, we have:

$$R_{VSD}^{(i)} \approx \mathbb{E}_{\mathbf{x}\sim\mathcal{B}}\mathbf{Trace}\left(\mathrm{diag}(\mathbf{h}_i(\mathbf{x}))\mathbf{U}\mathrm{diag}(\alpha)\mathbf{U}^T\mathrm{diag}(\mathbf{h}_i(\mathbf{x}))\mathbf{H}_i(\mathbf{x})\right)$$
$$= \mathbb{E}_{\mathbf{x}\sim\mathcal{B}}\|\mathbf{H}_i^{1/2}(\mathbf{x})\mathrm{diag}(\mathbf{h}_i(\mathbf{x}))\mathbf{U}\mathrm{diag}(\alpha^{1/2})\|_F^2.$$

This form can be interpreted as Tikhonov-like regularization imposed on the square root of Hessian matrix $\mathbf{H}_i$, in which the Tikhonov matrix is $\Gamma = \mathrm{diag}(\mathbf{h}_i)\mathbf{U}\mathrm{diag}(\alpha^{1/2})$. Thanks to variational learning in our method, the matrix $\Gamma$ is automatically learned instead of being manually designed. This can bring beneficial properties for training objective by encoding a notion of the smoothness of the loss function [9]. In addition, when considering the case of regression problem, from equation (17) we have $\mathbf{H}_{\mathrm{out}} = 1$ and:

$$R_{VSD}^{(i)} \approx \mathbb{E}_{\mathbf{x}\sim\mathcal{B}}\mathbf{Trace}\left(\mathbf{Q}\mathbf{J}_i^T(\mathbf{x})\mathbf{J}_i(\mathbf{x})\right)$$
$$= \mathbb{E}_{\mathbf{x}\sim\mathcal{B}}\mathbf{Trace}\left(\mathbf{J}_i(\mathbf{x})\mathbf{Q}\mathbf{J}_i^T(\mathbf{x})\right)$$
$$= \mathbb{E}_{\mathbf{x}\sim\mathcal{B}}\left(\mathbf{J}_i(\mathbf{x})\mathbf{Q}\mathbf{J}_i^T(\mathbf{x})\right)$$
$$= \mathbb{E}_{\mathbf{x}\sim\mathcal{B}}\left(\Theta^{[i:L]}\mathbf{Q}\Theta^{[i:L].T}\right), \tag{18}$$

in which the penultimate equation is because the quantity in the trace operator is scalar. This form is a data-dependent regularization with adaptive structure determined by the matrix $\mathbf{Q}$. With vanilla Dropout, the matrix $\mathbf{Q} = \mathrm{diag}(\alpha\odot\mathbf{h}_i(\mathbf{x})^2)$ plays a role as a scaling factor which allows Dropout regularizer to capture highly discriminative characteristics in each data feature. This interpretation generalizes some prior works studying Dropout for simple linear models [81, 27, 59]. Moreover, in VSD, the trainable orthogonal matrix $\mathbf{U}$ offers a more distinctive regularization effect. Specifically, the regularizer $R_{VSD}$ in our method makes the training algorithm capable of adapting to non-isotropic the geometric shape of data distribution. In other words, it reshapes the gradient of network weights according to the geometry of the data based on both scale and direction information. This property also connects our method with subgradient methods for online convex optimization in [14, 25], from which our regularization is closely related to adaptive proximal functions in these online frameworks.

**Interpretation 3:** *VSD penalizes implicitly the spectral norm of weight matrices, which has connection to generalization.* Let $\Omega_i := \mathrm{diag}(\mathbf{h}_i(\mathbf{x}))\mathbf{J}_i^T(\mathbf{x})\mathbf{H}_{\mathrm{out}}(\mathbf{x})\mathbf{J}_i(\mathbf{x})\mathrm{diag}(\mathbf{h}_i\mathbf{x})$, then also from equation (17), we have:

$$R_{VSD}^{(i)} \approx \mathbb{E}_{\mathbf{x}\sim\mathcal{B}}\mathbf{Trace}\left(\mathrm{diag}(\mathbf{h}_i(\mathbf{x}))\mathbf{U}\mathrm{diag}(\alpha)\mathbf{U}^T\mathrm{diag}(\mathbf{h}_i(\mathbf{x}))\mathbf{J}_i^T(\mathbf{x})\mathbf{H}_{\mathrm{out}}(\mathbf{x})\mathbf{J}_i(\mathbf{x})\right)$$
$$= \mathbb{E}_{\mathbf{x}\sim\mathcal{B}}\mathbf{Trace}\left(\mathbf{H}_{\mathrm{out}}(\mathbf{x})^{1/2}\mathbf{J}_i(\mathbf{x})\mathrm{diag}(\mathbf{h}_i(\mathbf{x}))\mathbf{U}\mathrm{diag}(\alpha)\mathbf{U}^T\mathrm{diag}(\mathbf{h}_i(\mathbf{x}))\mathbf{J}_i^T(\mathbf{x})\mathbf{H}_{\mathrm{out}}(\mathbf{x})^{1/2}\right)$$
$$= \mathbb{E}_{\mathbf{x}\sim\mathcal{B}}\|\mathbf{H}_{\mathrm{out}}(\mathbf{x})^{1/2}\mathbf{J}_i(\mathbf{x})\mathrm{diag}(\mathbf{h}_i(\mathbf{x}))\mathbf{U}\mathrm{diag}(\alpha^{1/2})\|_F^2$$
$$= \mathbb{E}_{\mathbf{x}\sim\mathcal{B}}\sum_{k=1}^{K}\alpha_k\|\mathbf{H}_{\mathrm{out}}(\mathbf{x})^{1/2}\mathbf{J}_i(\mathbf{x})\mathrm{diag}(\mathbf{h}_i(\mathbf{x}))\mathbf{U}_{:k}\|_2^2$$
$$= \mathbb{E}_{\mathbf{x}\sim\mathcal{B}}\sum_{k=1}^{K}\alpha_k\mathbf{U}_{:k}^T\Omega_i\mathbf{U}_{:k}. \tag{19}$$

Since the trainable matrix $\mathbf{U}$ satisfies $\mathbf{U}_{:k}^T\mathbf{U}_{:k} = 1$, the above form suggests that our regularization encourages the model to converge to solution with smaller spectral norms of the matrix $\mathbf{H}_{\mathrm{out}}(\mathbf{x})^{1/2}\mathbf{J}_i(\mathbf{x})\mathrm{diag}(\mathbf{h}_i(\mathbf{x}))$ and thus of the network weights.

Our analysis here is well-motivated by the theoretical study about generalization bound of neural nets based on the spectral norm. Concretely, Neyshabur et al. [65] and Bartlett et al. [4] showed that the generalization error is upper bounded by $\mathcal{O}\left(\sqrt{\prod_{i=1}^{L}\|\Theta^{(i)}\|_2^2\sum_{i=1}^{L}\mathrm{srank}(\Theta^{(i)})}\right)$ that depends on two parameter-dependent quantities: a) the scale-dependent Lipschitz constant upper-bound $\prod_{i=1}^{L}\|\Theta^{(i)}\|_2^2$ (product of spectral norms) and b) the sum of scale-independent stable ranks $\sum_{i=1}^{L}\mathrm{srank}(\Theta^{(i)})$. Essentially, this upper bound implies that smaller spectral norm and stable rank can lead to better generalization. We empirically investigate this implication and show the results

Table 9: Comparisons of Spectral Norms (SN) and Stable Ranks (SR) from different methods. Lower is better.

| Methods | MLP | | LeNet | | | | AlexNet | |
| | MNIST | | SVHN | | CIFAR10 | | SVHN | |
| | SN | SR | SN | SR | SN | SR | SN | SR |
| --- | --- | --- | --- | --- | --- | --- | --- | --- |
| MAP | **1.47** | 6.33 | 3.95 | 6.64 | 3.15 | 5.45 | 1.62 | 6.78 |
| MCD | 1.93 | 4.36 | 3.30 | 3.99 | 3.35 | 5.19 | 1.83 | 5.42 |
| VD | 1.88 | 5.28 | 2.94 | 6.04 | 2.71 | 5.39 | 1.83 | 5.30 |
| VSD | 2.05 | **4.19** | **2.28** | **2.54** | **1.94** | **4.94** | **1.36** | **5.23** |

in Table 9, in which we measure both spectral norm and stable rank of the weight matrix in the last layer of MLP, LeNet, and AlexNet architecture trained on MNIST, CIFAR10 and SVHN respectively. It can be seen that our method leads to a consistent reduction in terms of both the spectral norm and stable rank when compared with weight decay (MAP) and vanilla Dropout methods (MCD, VD). This also is evidenced in [4] that weight decay does not significantly impact margins or generalization.

# F  A complementary Bayesian justification for Variational Dropout

In this section, we will bring a new Bayesian perspective for Variational Dropout methods [39, 36] and then derive a variational objective to implement them in our experiments. Note that, the analysis below does not directly address the issues of original VD, thus have not yet provided any standard Bayesian justification for this method. Our new interpretation, however, is consistent with some recent research in the community.

**Variational Gaussian Dropout as Subspace inference.** Reusing the analysis in Section 3.1 in the main text, we have:

$$\mathbf{W}^{(VD)} = \mathrm{diag}(\xi)\Theta = \Theta + \mathrm{diag}(\eta)\Theta = \Theta + \sum_{i=1}^{K} \eta_i(\mathrm{diag}(\mathbf{e}_i)\Theta) = \Theta + \sum_{i=1}^{K} \eta_i\Theta_{(i)}, \qquad (20)$$

where $\xi \sim \mathcal{N}(1, \mathrm{diag}(\alpha))$, $\xi = 1 + \eta$ and $\Theta_{(i)}$ is the matrix $\Theta$ with only the $i$-th row retained. It turns out this representation can be well-interpreted under the subspace inference frameworks proposed in [32, 13]. Specifically, at each iteration of the inference phase, the weight parameter can be treated as the shift matrix, $\{\Theta_{(i)}\}_{i=1}^{K}$ are basic vectors of the subspace $\mathcal{S}$ and the noise $\eta = \{\eta_i\}_{i=1}^{K}$ is the low dimensional subspace parameter. Because $\{\Theta_{(i)}\}_{i=1}^{K}$ is linearly independent, so we can consider $\mathcal{S}$ as a projected space of the full parameter one.

We then perform variational inference over the low dimensional parameter $\eta$, also over $\xi$, in which the true posterior and the approximate posterior are defined by $p(\xi|\mathcal{D})$ and $q_\alpha(\xi)$ respectively. The variational objective function is given by:

$$\mathbb{E}_{q_\alpha(\xi)} \log p(\mathcal{D}|\xi, \Theta) - \mathbb{D}_{KL}(q_\alpha(\xi)\|p(\xi)). \qquad (21)$$

This form suggests us a similar procedure using approximate inference on the variational noise $\xi$. However, our interpretation can show a distinctness in terms of model specification. Concretely, to employ variational inference on the noise *in a principle way*, we need to define a probabilistic model where the noise should play a role as a latent variable. While plausible, we are mildly cautious when introducing a new Bayesian model with an implicit treatment.

**Adjust the KL divergence term with temperature scaling.** Note that, the equation (21) clarifies the derivation of type-A version in Variational Dropout paper [39], in which the authors used $\mathbb{D}_{KL}(q_\alpha(\xi)\|p(\xi))$ instead of the undefined term $\mathbb{D}_{KL}(q_\phi(\mathbf{W})\|p(\mathbf{W}))$ for implementation, but it seems just a heuristic way. On the other hand, [31] claimed that optimizing the above objective function might lead to significant overfitting due to the lack of regularization of $\Theta$. This issue has actually been mentioned in the subspace inference framework [32] with a similar way, from which we can propose to leverage the temperature technique to prevent the posterior from concentrating around the maximum likelihood estimate. In particular, we use the tempered posterior:

$$p_T(\xi|\mathcal{D}) \propto p(\mathcal{D}|\xi)^{1/\tau} p(\xi), \qquad (22)$$

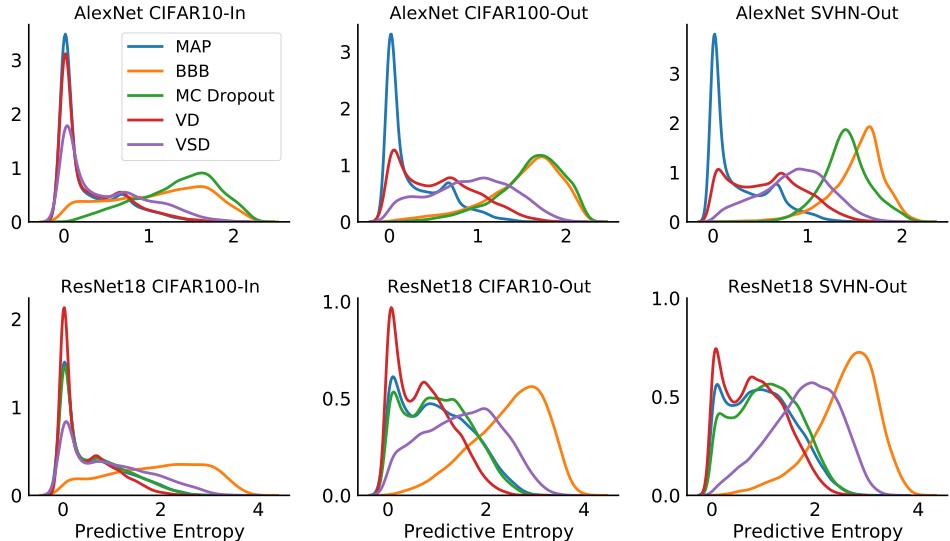

Figure 4: Histograms of predictive entropy for AlexNet (top) and ResNet18 (bottom) trained on CIFAR10 and CIFAR100 respectively.

with the temperature hyperparameter $\tau \gg 1$ chosen through cross-validation. It is also well-known that this technique is equivalent to the KL annealing (a detailed discussion is given in Appendix I.1), in which we optimize a fixed objective as:

$$\mathbb{E}_{q_\alpha(\xi)} \log p(\mathcal{D}|\xi, \Theta) - \tau * \mathbb{D}_{KL}(q_\alpha(\xi)\|p(\xi)). \tag{23}$$

We use this form for our implementation, the hyperparameter $\tau$ is tuned in the wide range instead of just picking values greater than 1. In particular for ARD-Variational Dropout method proposed in [36], the prior $p(\xi)$ is assigned by an isotropic Gaussian with the hyperparameter chosen via the Empirical Bayes, and then we obtain $\mathbb{D}_{KL}(q_\alpha(\xi)\|p(\xi)) = 0.5 \sum_{i=1}^{K} \log(1 + \alpha_i^{-1})$. It can be found that this term is a degenerate case of VSD when Householder transformation is deactivated, namely the orthogonal matrix $\mathbf{U}$ is just an identity matrix.

# G   Additional empirical results

## G.1   Predictive entropy performance on ResNet18 architecture

We recap here a consistent qualitative description for predictive entropy performance. Basically, an accurate and well-calibrated model is expected to represent entropy values being concentrated mostly around 0 (i.e. high confidence) when the test data comes from the same underlying distribution as the training data, and in the opposite case, the predictive entropies should be evenly distributed (i.e. higher uncertainty). In fact, the deep learning models do not achieve simultaneously on both expectations at the most ideal, but instead, accurate and well-calibrated ones tend to exhibit a moderate level of confidence on in-distribution data, and then provide a reasonable representation for uncertainty estimates on out-of-distribution data.

In the bottom row of Figure 4, we show the predictive entropy of methods when training ResNet18 architechture on CIFAR100 and testing out-of-distribution on CIFAR10 and SVHN. While most methods underestimate uncertainty in out-of-distribution data, our method - VSD, calibrates the prediction with moderate confidence on in-distribution data and provides proper uncertainty on out-of-distribution settings. We also observe that BBB fails to estimate the predictive uncertainty even on in-distribution data (same behavior as on AlexNet), and with a very low accuracy, this baseline has exhibited very poor results of the mean-field BNNs compared with the Bayesian Dropout methods in terms of predictive performance.

Table 10: Average test performance for UCI regression task. Results are reported with RMSE and Std. Errors.

| Dataset | BBB | VMG | MNF | SLANG | MCD | VD | D.E | VSD |
|---|---|---|---|---|---|---|---|---|
| Boston | 3.43 ± 0.20 | 2.70 ± 0.13 | 2.98 ± 0.06 | 3.21 ± 0.19 | 2.83 ± 0.17 | 2.98 ± 0.18 | 3.28 ± 0.22 | **2.64 ± 0.17** |
| Concrete | 6.16 ± 0.13 | 4.89 ± 0.12 | 6.57 ± 0.04 | 5.58 ± 0.19 | 4.93 ± 0.14 | 5.16 ± 0.13 | 6.03 ± 0.13 | **4.72 ± 0.11** |
| Energy | 0.97 ± 0.09 | 0.54 ± 0.02 | 2.38 ± 0.07 | 0.64 ± 0.03 | 1.08 ± 0.03 | 0.64 ± 0.02 | 2.09 ± 0.06 | **0.47 ± 0.01** |
| Kin8nm | 0.08 ± 0.00 | 0.08 ± 0.00 | 0.09 ± 0.00 | 0.08 ± 0.00 | 0.09 ± 0.00 | 0.08 ± 0.00 | 0.09 ± 0.00 | **0.08 ± 0.00** |
| Naval | 0.00 ± 0.00 | 0.00 ± 0.00 | 0.00 ± 0.00 | 0.00 ± 0.00 | 0.00 ± 0.00 | 0.00 ± 0.00 | 0.00 ± 0.00 | **0.00 ± 0.00** |
| Power Plant | 4.21 ± 0.03 | 4.04 ± 0.04 | 4.19 ± 0.01 | 4.16 ± 0.04 | 4.00 ± 0.04 | 3.99 ± 0.03 | 4.11 ± 0.04 | **3.92 ± 0.04** |
| Wine | 0.64 ± 0.01 | 0.63 ± 0.01 | **0.61 ± 0.00** | 0.65 ± 0.01 | 0.61 ± 0.01 | 0.62 ± 0.01 | 0.64 ± 0.00 | 0.63 ± 0.01 |
| Yacht | 1.13 ± 0.06 | 0.71 ± 0.05 | 2.13 ± 0.05 | 1.08 ± 0.06 | 0.72 ± 0.05 | 1.09 ± 0.09 | 1.58 ± 0.11 | **0.69 ± 0.06** |

Table 11: Average test performance for UCI regression task. Results are reported with test LL and Std. Errors.

| Dataset | BBB | VMG | MNF | SLANG | MCD | VD | D.E | VSD |
|---|---|---|---|---|---|---|---|---|
| Boston | -2.66 ± 0.06 | -2.46 ± 0.09 | -2.51 ± 0.06 | -2.58 ± 0.05 | -2.40 ± 0.04 | -2.39 ± 0.04 | -2.41 ± 0.06 | **-2.35 ± 0.05** |
| Concrete | -3.25 ± 0.02 | -3.01 ± 0.03 | -3.35 ± 0.04 | -3.13 ± 0.03 | **-2.97 ± 0.02** | -3.07 ± 0.03 | -3.06 ± 0.04 | **-2.97 ± 0.02** |
| Energy | -1.45 ± 0.10 | -1.06 ± 0.03 | -3.18 ± 0.07 | -1.12 ± 0.01 | -1.72 ± 0.01 | -1.30 ± 0.01 | -1.38 ± 0.05 | **-1.06 ± 0.01** |
| Kin8nm | 1.07 ± 0.00 | 1.10 ± 0.01 | 1.04 ± 0.00 | 1.06 ± 0.00 | 0.97 ± 0.00 | 1.14 ± 0.01 | 1.20 ± 0.00 | **1.17 ± 0.01** |
| Naval | 4.61 ± 0.01 | 2.46 ± 0.00 | 3.96 ± 0.01 | 4.76 ± 0.00 | 4.76 ± 0.01 | 4.81 ± 0.00 | 5.63 ± 0.00 | **4.83 ± 0.01** |
| Power Plant | -2.86 ± 0.01 | -2.82 ± 0.01 | -2.86 ± 0.01 | -2.84 ± 0.01 | **-2.79 ± 0.01** | -2.82 ± 0.01 | -2.79 ± 0.01 | **-2.79 ± 0.01** |
| Wine | -0.97 ± 0.01 | -0.95 ± 0.01 | -0.93 ± 0.00 | -0.97 ± 0.01 | **-0.92 ± 0.01** | -0.94 ± 0.01 | -0.94 ± 0.03 | -0.95 ± 0.01 |
| Yacht | -1.56 ± 0.02 | -1.30 ± 0.02 | -1.96 ± 0.05 | -1.88 ± 0.01 | -1.38 ± 0.01 | -1.42 ± 0.02 | -1.18 ± 0.05 | **-1.14 ± 0.02** |

## G.2 Regression with UCI datasets

We implement a standard experiment for Bayesian regression task on UCI dataset [3] proposed in [28]. We follow the original setup used in [20]. Detailed descriptions of the data and experimental setting can be found in Appendix I.3. We present the performance of methods based on standard metrics including root mean square error (RMSE) in Table 10 and predictive log-likelihood (LL) in Table 11. As shown in the tables, VSD performs better than baselines on most datasets in terms of both criteria (5/8 tasks on RMSE and 7/8 tasks on predictive LL). Especially, in comparison with other structured approximations on BNNs such as VMG, MNF and SLANG, our method presents much more convincing results, while MNF even shows no noticeable improvement compared to mean-filed approximation (BBB). VSD also achieves better results with a significant margin compared with VD and Deep Ensemble (D.E), specifically on *Boston, Concrete, Energy, Yacht* datasets. This demonstrates the effectiveness of learning a structured representation for multiplicative Gaussian noise instead of using a diagonal distribution as in VD.

For predictive log-likelihood measure, although VSD is comparable to MCD and VMG on some datasets such as *Concrete, Power Plant*, however our method overall shows better results consistently on almost all settings. For instance, MCD gets poor results on *Energy, Kin8nm* and *Yacht*, while VMG is worse than VSD in *Boston, Naval* and *Yacht* by large margins. In comparison to VD and MCD, our method improves considerably the performance on *Concrete, Energy* and *Yacht*.

## G.3 Uncertainty with toy regression

We provide an additional experiment to assess qualitatively the predictive uncertainty of methods using a synthetic regression dataset introduced in [28]. We generated 20 training inputs from $\mathcal{U}[-4, 4]$ and assigned the corresponding target as $y_n = x_n^3 + \epsilon_n$ where $\epsilon_n \sim \mathcal{N}(0, 9)$. We then fitted a neural network with a single hidden layer of 100 units. We also fixed the variance of likelihood regression to the true variance of noise $\epsilon$. We compare the performance of methods including BBB, MCD, VD and VSD. For the Dropout-based methods, we do not apply the dropout noise for the input layer since it is 1-dimensional. At the test time of all methods, we use 1000 MC samples to approximate the predictive distribution. The results are shown in Figure 5.

We would expect that in the area of observed data, the models should obtain the predictive means closer to the ground truth with high confidence, and at the same time, increase the predictive variance when moving away from the data. Thus we can see that our method, VSD, provides a more realistic predictive distribution than the remaining ones.

For MCD, we fixed the dropout rate $p$ at default 0.5, because tuning manually this value does not increase the variance of noise distribution Bernoulli$(1 - p)$, and this will lead to less variance in subsequent pre-activation unit by the Central Limit Theorem [82]. Therefore, with the shallow

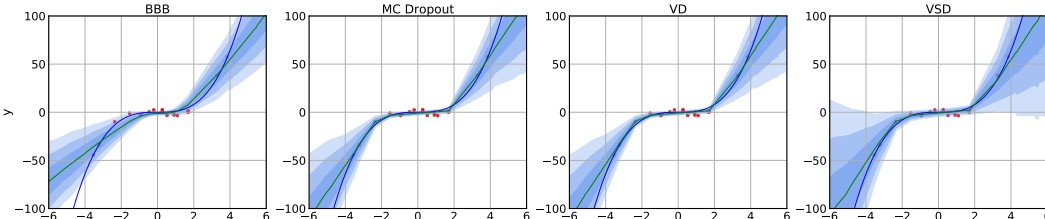

Figure 5: Predictive distributions for the toy dataset. The observations are shown as red dots. The blue line represents the true data generating function and the mean predictions are shown as the green line. Blue areas correspond to ±3 standard deviation around the mean.

architecture of one hidden unit in this experiment, any tuning of the droprate $p$ can result into a decrease in the predictive variance of model.

Whereas with VD, the droprate $\alpha$ needs to be restricted in range $(0, 1)$ during the training to avoid the poor local optima and to prevent the objective function from being degenerate [39, 57, 31]. As a consequence, this can reduce the ensemble diversity in the predictions of this method that leads to underestimate the uncertainty of predictive distribution.

On the contrary, the parameters in our method can be optimized without any limited assumptions. Moreover, with a structured representation for Gaussian perturbation, our method can capture rich statistical dependencies in the true posterior that facilitates fidelity posterior approximation and then provides proper estimates of the true model uncertainty.

In addition, we also conduct one more experiment to investigate the ability of VSD to estimate in-between uncertainty. This experiment is motivated by a recent work of Foong et al. [17], in which the authors proved that for shallow Bayesian neural nets, neither mean-field Gaussian nor Dropout posterior are capable of expressing meaningful in-between uncertainty in many situations. Due to the acquired distinctiveness in the structure of Dropout posterior distribution, we suggest that our method-VSD can theoretically overcome this limitation of MC Dropout and Variational Dropout. We validate empirically this statement by considering a regression problem with the dataset consisting of two well-separated clusters of covariates. We apply VSD to train a ReLU network with one hidden layer of size 50 and then present the predictive distributions in

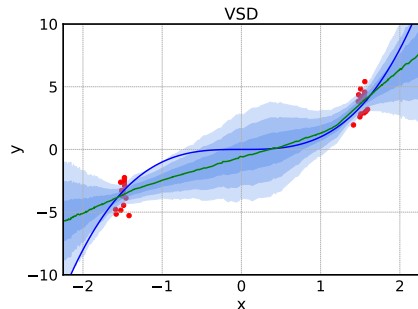

Figure 6: In-between uncertainty.

Figure 6. Contrary to the behavior of MC Dropout and VD reported in [17], VSD can represent a reasonable uncertainty between two data clusters. A more comprehensive analysis would be promising work for future research.

## H Further discussion of uncertainty measures

We present here several approaches used to measure or assess the quality of the predictive uncertainty of models. We also present some insights into what those metrics mean. For simplicity, we consider the multi-class classification problems on the supervised dataset $\mathcal{D} = \{\mathbf{x}_i, \mathbf{y}_i\}_{i=1}^N$. For Bayesian methods, we can compute the predictive probabilities for each sample $\{\mathbf{x}_i, \mathbf{y}_i\}_{i=1}$ that belongs to class $c$ as:

$$\hat{p}_{ic} := \int p(\mathbf{y}_i = c | \mathbf{x}_i, \mathbf{W}) p(\mathbf{W} | \mathcal{D}) d\mathbf{W}$$

$$\approx \int p(\mathbf{y}_i = c | \mathbf{x}_i, \mathbf{W}) q(\mathbf{W}) d\mathbf{W} \approx \frac{1}{S} \sum_{s=1}^S p(\mathbf{y}_i = c | \mathbf{x}_i, \mathbf{W}^{(s)}) \qquad (24)$$

with $\{\mathbf{W}^{(s)}\}_{s=1}^S$ are variational Monte Carlo samples. The following metrics are all based on this predictive probability.

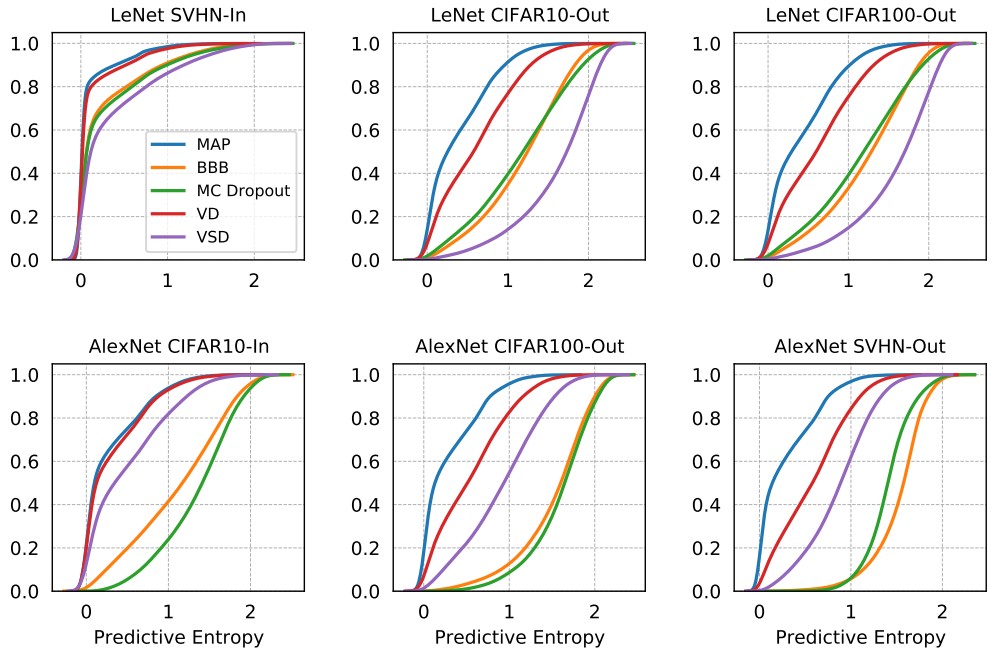

Figure 7: Empirical CDF of the predictive entropy forfor LeNet (top) and AlexNet (bottom) trained on SVHN and CIFAR10 respectively.

## H.1 Negative log-likelihood

The negative log-likelihood (NLL) is a standard measure of a probabilistic model's quality and also a common uncertainty metric which is defined by: $\frac{1}{N} \sum_{i=1}^{N} \sum_{c=1}^{K} -y_{ic} \log \hat{p}_{ic}$. If the predicted probabilities are overconfident, NLL will severely penalize incorrect predictions, causing this quantity to become large even if the test error rate is low. In contrast, an underconfident prediction contributes a substantial amount to the NLL regardless of whether the prediction is correct or not. Therefore, a model that achieves a good test NLL tends to make predictions with sufficiently high confidence on easy samples and hesitant predictions on hard, easy-to-fail samples.

These arguments can be evidenced by the results of experiments on modern convolutional networks (Table 4 and Table 5). Although MAP has the predictive accuracy being competitive with our method, it comes at a trade-off with the worst results on NLL. Thus the predictions of MAP are more likely to be overconfident. Corresponding experiments on out-of-distribution settings (Figure 3 bottom and Figure 4) confirmed this.

## H.2 Predictive entropy

Predictive entropy determined on a input sample $\{\mathbf{x}_i, \mathbf{y}_i\}_{i=1}$ is given by $\frac{1}{K} \sum_{c=1}^{K} -\hat{p}_{ic} \log \hat{p}_{ic}$. Underconfident models give noisy predictive predictions which result in high entropy on even in-distribution data. In contrast, overconfident models with spike predictive distributions tend to produce near zero predictive entropies.

In Figures 3 and Figure 4 we plot the histogram of predictive entropies to quantify uncertainty estimation of the methods on out-of-distribution settings. In some cases, when the histograms are difficult to distinguish from each other, we instead use the empirical CDF which may be more informative [48]. We redraw the Figure 3 in the main text by empirical CDF in Figure 7. The distance between lines gives more visual views on the performance differences between the methods.

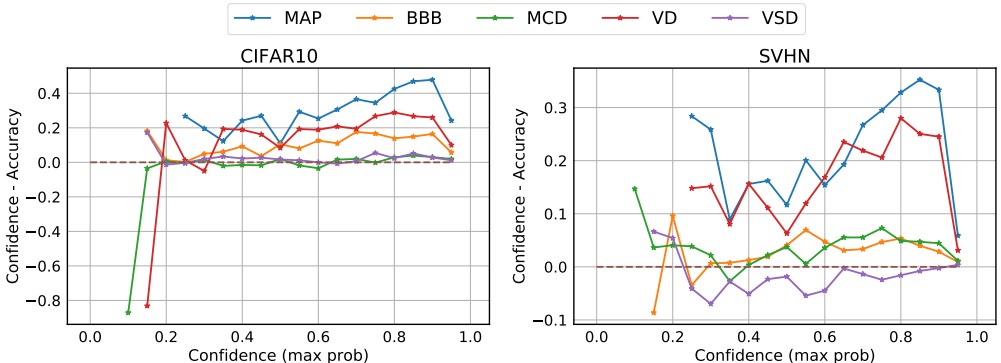

Figure 8: Reliability diagrams of LeNet-5 on CIFAR10 and SVHN dataset.

## H.3   Expected Calibration Error

Expected Calibration Error (ECE) [23] captures the discrepancy between model's predicted probability estimates and the actual accuracy. This quantity is computed by first binning the predicted probabilities into $M$ distinct bins and calculate the accuracy of each bin. Let $B_m$ be the set of indices of samples whose prediction confidence falls into the interval $I_m = (\frac{m-1}{M}, \frac{m}{M}]$. The accuracy of $B_m$ and the average confidence within bin $B_m$ are defined as follows:

$$\mathrm{acc}(B_m) := \frac{1}{|B_m|} \sum_{i \in B_m} \mathbf{1}[\hat{\mathbf{y}}_i = \mathbf{y}_i], \quad \mathrm{conf}(B_m) := \frac{1}{|B_m|} \sum_{i \in B_m} q(\mathbf{x}_i),$$

where $q(\mathbf{x}_i)$ is the confidence for sample $i$. In this work, we define the confidence score $q$ as the maximum predictive probability on each data of the classifier. ECE is then computed by taking a weighted average of the bins' accuracy/confidence difference:

$$\mathrm{ECE} := \sum_{m=1}^{M} \frac{|B_m|}{n} |\mathrm{acc}(B_m) - \mathrm{conf}(B_m)|. \tag{25}$$

**Reliability Diagrams** [23] in Figure 8 are visual representations of model calibration. These diagrams plot accuracy as a function of confidence. Any deviation from a perfect horizontal zero-line represents miscalibration. We can observe that MAP and VD exhibit overconfident prediction (low accuracy on many bins of high confidence). Meanwhile, VSD calibrates well the predictive probabilities resulting in the lowest ECE in both settings.

## H.4   Out-of-distribution detection metrics

We measure some metrics to evaluate the model's ability of distinguishing in distribution and out-of-distribution images.

An ideal classifier should have a low probability of false alarm, corresponding to a low False Positive Rate (FPR), while maintaining a high sensitivity, corresponding to a high True Positive Rate (TPR). **FPR at 95% TPR** is a suitable metric to measure the reality of this desire.

Receiver Operating Characteristic (ROC) curve plots the TPR against the FPR at all possible decision thresholds. The larger the Area Under the ROC curve (**AUROC**), the better the classifier's ability to separate the negative and positive samples. More intuitively, AUROC indicates the chance that the classifier produces a higher score on a random positive sample than a random negative one.

Besides the ROC curve, which represents the trade-off between the sensitivity and the probability of false alarm, Precision-Recall (PR) curves are often plotted to represent a trade-off between making accurate positive predictions and covering a majority of all positive results. We report the Area Under the Precision-Recall curve (AUPR) in two scenarios: (i) in-distribution samples are used as the positive samples (**AUPR-In**), (ii) out-of-distribution samples are used as the positive samples (**AUPR-Out**).

Table 12: Quality of out-of-distribution detection on image classification tasks. (Top) LeNet-5 train on SVHN, evaluate on CIFAR10, CIFAR100. (Middle) AlexNet train on CIFAR10, evaluate on CIFAR100, SVHN. (Bottom) ResNet-18 train on CIFAR, evaluate on CIFAR10, CIFAR100. ↑ (AUROC, AUPR IN, AUPR OUT) indicates larger value is better, and ↓ (FPR, Detection error) indicates lower value is better. VSD performs the best on almost all metrics and datasets.

| LeNet-5 (SVHN) | CIFAR10 | | | | | CIFAR100 | | | | |
|---|---|---|---|---|---|---|---|---|---|---|
| | FPR | Det. err. | AUROC | AUPR IN | AUPR OUT | FPR | Det. err. | AUROC | AUPR IN | AUPR OUT |
| MAP | 0.78 | 0.23 | 0.83 | 0.93 | 0.58 | 0.76 | 0.22 | 0.84 | 0.93 | 0.60 |
| BBB | 0.56 | 0.17 | 0.90 | 0.96 | 0.73 | 0.54 | 0.17 | 0.90 | 0.96 | 0.75 |
| MCD | 0.50 | 0.15 | 0.92 | **0.97** | 0.78 | 0.49 | 0.15 | **0.92** | **0.97** | 0.78 |
| VD | 0.62 | 0.17 | 0.89 | 0.96 | 0.71 | 0.64 | 0.17 | 0.89 | 0.96 | 0.71 |
| VSD | **0.45** | **0.14** | **0.93** | **0.97** | **0.81** | **0.47** | **0.14** | **0.92** | **0.97** | **0.79** |

| AlexNet (CIFAR10) | CIFAR100 | | | | | SVHN | | | | |
|---|---|---|---|---|---|---|---|---|---|---|
| | FPR | Det. err. | AUROC | AUPR IN | AUPR OUT | FPR | Det. err. | AUROC | AUPR IN | AUPR OUT |
| MAP | 0.88 | 0.35 | 0.70 | 0.73 | 0.65 | **0.89** | 0.33 | 0.71 | 0.59 | **0.83** |
| BBB | 0.93 | 0.46 | 0.55 | 0.54 | 0.54 | 0.99 | 0.45 | 0.53 | 0.33 | 0.70 |
| MCD | 0.91 | 0.41 | 0.63 | 0.63 | 0.60 | 0.97 | 0.39 | 0.59 | 0.47 | 0.74 |
| VD | 0.87 | 0.35 | 0.69 | 0.72 | 0.64 | 0.89 | 0.32 | 0.72 | 0.60 | **0.83** |
| VSD | **0.85** | **0.33** | **0.72** | **0.76** | **0.68** | 0.91 | **0.30** | **0.73** | **0.65** | **0.83** |

| ResNet-18 (CIFAR100) | CIFAR10 | | | | | SVHN | | | | |
|---|---|---|---|---|---|---|---|---|---|---|
| | FPR | Det. err. | AUROC | AUPR IN | AUPR OUT | FPR | Det. err. | AUROC | AUPR IN | AUPR OUT |
| MAP | 0.89 | **0.37** | 0.67 | 0.70 | 0.63 | 0.91 | 0.36 | 0.68 | 0.56 | 0.81 |
| BBB | 0.93 | 0.41 | 0.62 | 0.66 | 0.58 | 0.89 | 0.37 | 0.68 | 0.51 | 0.82 |
| MCD | 0.89 | **0.37** | 0.68 | 0.71 | 0.63 | 0.89 | 0.34 | 0.71 | 0.58 | 0.83 |
| VD | 0.90 | 0.38 | 0.66 | 0.70 | 0.62 | 0.87 | 0.34 | 0.70 | 0.58 | 0.83 |
| VSD | **0.87** | **0.37** | **0.69** | **0.72** | **0.65** | **0.83** | **0.31** | **0.76** | **0.65** | **0.86** |

Finally, we report the **detection error**, a measure of the minimum expected probability that the model incorrectly detects whether data samples come from in or out of training data distribution. This quantity is defined as $\min_\delta \{0.5 P_{in}(q(\mathbf{x}) \leq \delta) + 0.5 P_{out}(q(\mathbf{x}) > \delta)\}$ where $\delta$ is the decision threshold and $q(\mathbf{x})$ is the maximum value of softmax probability.

# I   Details for experimental settings

## I.1   Training techniques

**Initialization and Learning rate scheduling.** It is well known that good initialization and proper learning rate can improve both the speed and quality of convergence in the training process. In our experiments, we adopt `init.xavier_uniform` in PyTorch to initialize the weight parameters of all methods, of which we use 5 random seeds for different initializations and then report average results.

For positive-valued parameters such as dropout rate $\alpha$ or Gaussian variance $\sigma^2$, we optimize in the logarithmic form to avoid numerical issues and bad local optima. We use Adam optimizer with initial learning rate $lr \in \{0.001, 0.002\}$ on all of our experiments, and then we also apply `MultiStepLR` scheduler with multiplicative factor `gamma`=0.3 to adjust the learning rate after every 10 epochs.

**KL annealing.** This technique re-weights the expected log-likelihood and regularized term by a scaling factor $\lambda$ as follows:

$$\mathbb{E}_{q_\phi \mathbf{W}} \log p(\mathcal{D}|\mathbf{W}) - \lambda \mathbb{D}_{KL}(q_\phi(\mathbf{W})||p(\mathbf{W})). \tag{26}$$

The KL annealing in many contexts is remarkably effective to the problems using variational Bayesian inference. It intuitively can prevent underfitting/over-regularization issue, or mitigate KL-vanishing phenomenon. The KL annealing has an interpretation of the Bayesian principle. Indeed, re-weighting the KL term by a $\lambda$ is equivalent to tempering the posterior by a temperature factor $\tau = \lambda$, namely using $p_\tau(\mathbf{W}|\mathcal{D}) \propto p(\mathcal{D}|\mathbf{W})^{1/\tau} p(\mathbf{W})$ [84]. In another interpretation, one can use a different likelihood $p_\tau(\mathcal{D}|\mathbf{W}) = p(\mathcal{D}|\mathbf{W})^{1/\tau}$ instead of a tempered posterior $p_\tau(\mathbf{W}|\mathcal{D})$ above [86]. Even though the posteriors coincide, the predictive distribution differs for these two ways.

A temperature value $\tau < 1$, namely $\lambda < 1$, is corresponding to artificially sharpening the posterior distribution, which can be interpreted as overcounting the data $\mathcal{D}$ by a factor of $1/\tau$. So, the variational objective with KL annealing is a lower bound of $\tau$ times the model evidence defined on the overcounted data $\widehat{\mathcal{D}}$. Maximizing this objective is equivalent to minimizing the KL divergence between the approximate distribution $q_\phi(\mathbf{W})$ and the tempered posterior $p_\tau(\mathbf{W}|\mathcal{D})$.

We employ this technique for all methods in our paper and then tuning the weighting hyper-parameter $\lambda$ by cross-validation (of course with $\lambda = 1$, we have no modification). The empirical values of $\lambda$ is given in Appendix I.2.

## I.2   Hyperparameter tuning for all methods

We present here in detail the hyper-parameter setups of each method used in our experiments. These methods include MAP, Bayes by Backprop (BBB), MC Dropout (MCD), Variational Dropout (VD), and our method-Variational Structured Dropout (VSD); for the remaining methods such as Variational Matrix Gaussian (VMG), low-rank approximations (SLANG, and ELRG), we inherited the results reported in the original paper with the same experimental settings.

For **BBB** without the local reparameterization trick, we use two Monte Carlo samples for all of our experiments. This method needs to tune the hyper-parameters in the scale mixture Gaussian prior $p(\mathbf{W}) = \pi \mathcal{N}(0, \sigma_1^2) + (1 - \pi)\mathcal{N}(0, \sigma_2^2)$ with the search as follows: $-\log \sigma_1 \in \{0, 1, 2\}$, $-\log \sigma_2 \in \{6, 7, 8\}$ and mixture ratio $\pi \in \{0.25, 0.5, 0.75\}$.

For **MC Dropout** in image classification task, we tune the droprate $p \in \{0.2, 0.3, 0.4, 0.5, 0.6\}$ in Bernoulli distribution and the length-scale $l^2 \in \{0.0001, 0.001, 0.01, 0.1, 1, 10, 100\}$ in the isotropic Gaussian prior $p(\mathbf{W}) = \mathcal{N}(0, l^{-2}\mathbf{I}_K)$.

For **VD** and **VSD**, we find a good initialization for the Gaussian dropout rate $\alpha = p/(1-p)$. However, this droprate $\alpha$ in VD methods needs to be restricted in range $(0, 1)$ during the training to prevent the objective function from being degenerate which can lead to the poor local optima as mentioned in some related papers [39, 57, 31].

We also employ the KL annealing mentioned in the previous section I.1 for BBB, VD and VSD methods. This technique has actually been exploited in the original papers of BBB and VD to improve the predictive performance. The most appropriate values of annealing factor $\lambda$ we have found are consistent with the reports of previous works in the literature that have been synthesized in [84]. Specifically, we used $\lambda \in \{0.1, 0.2, 0.5\}$ for the classification tasks, and $\lambda$ from $10^{-5}$ to $10^{-2}$ for the regression tasks.

For **SWAG**, we train the models using SGD with momentum $\gamma = 0.9$. For the learning rate we adopt the same decaying schedule as in the paper: a constant learning rate of $5 \times 10^{-2}$, up to 50% of the total number of epochs, and then a linear decay down to a learning rate of $5 \times 10^{-4}$ until 90% of the total number of epochs, where once again the learning rate is kept constant until the end of training. We use rank $K = 20$ for estimation of Gaussian covariance matrix. At test time we use 30 weight samples for Bayesian model averaging. We also tune the weight decay in the range of $\{1e-3, 5e-3, 1e-4, 5e-4\}$.

## I.3   UCI regression settings

Following the original settings, we used a Bayesian neural network with one hidden layer of 50 units and ReLU activation functions. We also used the 20 splits of the data provided by [20][2] for training and testing. The models are trained to convergence using Adam optimizer [37] with the learning rate $lr = 0.001$, the batchsize $M = 128$ and 2000 epochs for all datasets. To make an ensemble prediction at the test time, we used 10000 Monte Carlo samples for the Bayesian Dropout methods as the suggestion in [20].

For VMG, SLANG and Deep Ensemble, we inherited the results reported in the original paper. The results of MNF is report in [78]. For a more fair comparison, we used the results of MC Dropout reported in [60] with the version using 4000 epochs for convergence training and tuning hyper-parameters by Bayesian Optimization.

---

[2] The splits are publicly available from `https://github.com/yaringal/DropoutUncertaintyExps`

Table 13: The performance of VSD and VOGN on CIFAR10. Results are averaged over 5 random seeds.

| CIFAR10 | AlexNet | | | | ResNet18 | | | |
|---------|---------|---------|---------|---------|----------|---------|---------|---------|
| | NLL | ACC | ECE | time | NLL | ACC | ECE | time |
| VOGN | 0.703 ± 0.006 | 75.48 ± 0.478 | **0.016 ± 0.001** | 3.25x | 0.477 ± 0.006 | 84.27 ± 0.195 | **0.040±0.002** | 4.44x |
| VSD | **0.656 ± 0.009** | **78.21 ± 0.153** | 0.046 ± 0.003 | **1.32x** | **0.464 ± 0.019** | **87.44 ± 0.497** | 0.061 ± 0.005 | **1.86x** |

In this experiment, we used the number of Householder transformations $T = 2$, and need to tune the precision $\tau$ of Gaussian likelihood $p(\mathbf{y}|\mathbf{W}, \mathbf{x}, \tau) = \mathcal{N}(\mathbf{y}|f(\mathbf{W}, \mathbf{x}), \tau)$. Similar to MC Dropout and SLANG, we used 40 iterations of Bayesian Optimization (BO) to tune this precision. For each iteration of BO, 5-fold cross-validation is used to evaluate the considered hyperparameter setting. This is repeated for each of the 20 train-test splits for each dataset. The final values of each dataset are reported with the mean and standard error from these 20 runs.

## I.4 Image classification settings

With the MNIST dataset, 60,000 training points were split into a training set of 50,000 and a validation set of 10,000. We then vectorized the images and trained using two fully connected Bayesian neural networks with the size of hidden layers of 400x2 and 750x3 respectively.

For the remaining two datasets CIFAR10 and SVHN, we used the same simple convolutional neural network consisting of two convolutional layers with 32 and 64 kernels, followed by a fully connected network with one hidden layer of size 128.

We trained the models with the default Adam optimizer using learning rate 0.001, batchsize 100, and the number of epochs 100. At the test time, we used 100 Monte Carlo samples for all methods. We also used the number of Householder transformations $T \in \{1, 2, 3\}$ for our method on all three datasets. With Bayes by Backprop (BBB), we did not employ the local reparameterization trick, instead we used two MC samples during the training follow the code published by the authors. We utilized the available results of the baselines in the same setting, including NLL and error rate of ELRG on MNIST and CIFAR10 dataset, error rate of VMG and SLANG on MNIST dataset.

Note that, in these experiments, we did not implement fully Bayesian inference for the convolutional layer, we instead just have done it on fully connected layers. This is because our initial intention was to conduct a fair experiment to compare with some other structured approximations including Variational Matrix Gaussian (VMG) and SLANG. These methods are nontrivial when applied to convolutional layers, even no available results for CNNs have been reported. Unfortunately, however, we had not successfully implemented these methods even on fully connected networks (only some of their results are inherited in the main text)). On the other hand, we have performed fully Bayesian approximations on the large-scale architecture AlexNet and ResNet18 as the following description.

## I.5 Scaling up modern CNNs settings

We follow the experimental setup for Bayesian deep convolutional networks proposed in [80]. We trained all algorithms for 300 epochs using a batch size of 200 and the ADAM optimizer with learning rate 0.001. We normalize datasets using empirical mean and standard deviation and then employ data augmentation for these experiments: random padding followed by flipping left/right (except SVHN). In the testing phase, we used 10 variational Monte Carlo samples for both AlexNet and ResNet18 architecture. We report average results over 5 random initializations. We refer to the code for MC Dropout and Bayes by Backprop on AlexNet and ResNet18 that is available at the repo `https://github.com/team-approx-bayes/dl-with-bayes` of VOGN paper [68]. We would also like to provide a few comparisons between VSD and VOGN as shown in Table 13, from which our method VSD has better performance than VOGN on almost all metrics, especially on the accuracy and computational time.

## I.6 Empirical classification results with standard deviations

For the convenience of presentation, we have not included in the main text the standard deviations of empirical results in Tables 3-4-5. We provide them here for clarification.

Table 3: Results for VSD and baselines on vectorized MNIST, CIFAR10 and SVHN. Results are averaged over 5 random seeds. For all metrics, lower is better.

| Method | MNIST FC 400x2 NLL | err. rate | ECE | CIFAR10 FC 750x3 NLL | err. rate | ECE | SVHN CNN 32x64x128 NLL | err. rate | ECE |
|---|---|---|---|---|---|---|---|---|---|
| MAP | 0.098 ± 0.009 | 1.32 ± 0.17 | 0.011 ± 0.002 | 2.847 ± 0.327 | 34.04 ± 1.547 | 0.272 ± 0.008 | 0.855 ± 0.049 | 12.26 ± 0.522 | 0.086 ± 0.009 |
| BBB | 0.109 ± 0.015 | 1.59 ± 0.24 | 0.011 ± 0.002 | 1.202 ± 0.114 | 30.11 ± 0.762 | 0.098 ± 0.006 | 0.545 ± 0.033 | 10.57 ± 0.236 | 0.017 ± 0.006 |
| MCD | 0.049 ± 0.005 | 1.26 ± 0.15 | 0.007 ± 0.001 | 0.794 ± 0.035 | 26.91 ± 0.241 | 0.024 ± 0.003 | 0.365 ± 0.018 | 9.23 ± 0.287 | 0.013 ± 0.006 |
| VD | 0.051 ± 0.005 | 1.21 ± 0.18 | 0.007 ± 0.001 | 1.176 ± 0.067 | 27.45 ± 0.222 | 0.156 ± 0.007 | 0.534 ± 0.042 | 9.47 ± 0.331 | 0.055 ± 0.007 |
| ELRG | 0.053 ± 0.006 | 1.54 ± 0.18 | - | 0.871 ± 0.011 | 29.43 ± 0.320 | - | - | - | - |
| **VSD** | **0.042 ± 0.004** | **1.08 ± 0.07** | **0.006 ± 0.000** | **0.730 ± 0.018** | **24.92 ± 0.355** | **0.020 ± 0.003** | **0.299 ± 0.009** | **8.39 ± 0.201** | **0.008 ± 0.001** |
| D.E | 0.057 ± 0.005 | 1.29 ± 0.11 | 0.009 ± 0.001 | 1.815 ± 0.218 | 26.44 ± 0.211 | 0.042 ± 0.003 | 0.783 ± 0.015 | 9.31 ± 0.298 | 0.070 ± 0.005 |
| SWAG | 0.044 ± 0.005 | 1.27 ± 0.13 | 0.008 ± 0.001 | 0.799 ± 0.103 | 26.94 ± 0.176 | **0.012 ± 0.002** | 0.312 ± 0.011 | 8.42 ± 0.191 | 0.021 ± 0.003 |

Table 4: Image classification using AlexNet architecture. Results are averaged over 5 random seeds.

| AlexNet | CIFAR10 NLL | ACC | ECE | CIFAR100 NLL | ACC | ECE | SVHN NLL | ACC | ECE | STL10 NLL | ACC | ECE |
|---|---|---|---|---|---|---|---|---|---|---|---|---|
| MAP | 1.038 ± 0.013 | 69.58 ± 0.57 | 0.121 ± 0.003 | 4.705 ± 0.075 | 40.23 ± 0.56 | 0.393 ± 0.015 | 0.418 ± 0.022 | 87.56 ± 0.58 | 0.033 ± 0.007 | 2.532 ± 0.278 | 65.70 ± 0.88 | 0.267 ± 0.060 |
| BBB | 0.994 ± 0.008 | 65.38 ± 0.32 | 0.062 ± 0.004 | 2.659 ± 0.051 | 32.41 ± 1.39 | **0.049 ± 0.010** | 0.476 ± 0.015 | 87.30 ± 0.56 | 0.094 ± 0.005 | 1.707 ± 0.085 | 65.46 ± 0.52 | 0.222 ± 0.008 |
| MCD | 0.717 ± 0.010 | 75.22 ± 0.33 | 0.023 ± 0.003 | 2.503 ± 0.025 | 42.91 ± 0.36 | 0.151 ± 0.021 | 0.401 ± 0.008 | 88.03 ± 0.13 | 0.023 ± 0.003 | 1.059 ± 0.013 | 63.65 ± 1.10 | **0.052 ± 0.021** |
| VD | 0.702 ± 0.014 | 77.28 ± 0.30 | **0.028 ± 0.002** | 2.582 ± 0.085 | 43.10 ± 0.64 | 0.106 ± 0.009 | 0.327 ± 0.015 | 90.76 ± 0.44 | 0.010 ± 0.004 | 2.130 ± 0.038 | 65.48 ± 0.98 | 0.195 ± 0.010 |
| ELRG | 0.723 ± 0.010 | 76.87 ± 0.42 | 0.065 ± 0.006 | 2.368 ± 0.043 | 42.90 ± 0.84 | 0.099 ± 0.022 | 0.312 ± 0.007 | 90.66 ± 0.31 | **0.006 ± 0.001** | 1.088 ± 0.046 | 59.99 ± 2.15 | **0.018 ± 0.008** |
| **VSD** | **0.656 ± 0.009** | **78.21 ± 0.15** | 0.046 ± 0.003 | **2.241 ± 0.026** | **46.85 ± 0.99** | 0.112 ± 0.010 | **0.290 ± 0.010** | **91.62 ± 0.33** | **0.008 ± 0.001** | **1.019 ± 0.039** | 67.98 ± 0.50 | 0.079 ± 0.010 |
| D.E | 0.872 ± 0.008 | 77.56 ± 0.17 | 0.115 ± 0.004 | 3.402 ± 0.067 | 46.42 ± 0.33 | 0.314 ± 0.020 | 0.319 ± 0.019 | 90.30 ± 0.47 | **0.008 ± 0.001** | 2.229 ± 0.155 | **68.51 ± 0.78** | 0.241 ± 0.057 |
| SWAG | **0.651 ± 0.014** | **78.14 ± 0.51** | 0.059 ± 0.003 | **1.958 ± 0.051** | **49.81 ± 0.62** | **0.028 ± 0.009** | 0.331 ± 0.019 | 90.04 ± 0.21 | 0.031 ± 0.005 | 1.522 ± 0.097 | **68.41 ± 0.68** | 0.161 ± 0.013 |

Table 5: Image classification using ResNet18 architecture. Results are averaged over 5 random seeds.

| ResNet18 | CIFAR10 NLL | ACC | ECE | CIFAR100 NLL | ACC | ECE | SVHN NLL | ACC | ECE | STL10 NLL | ACC | ECE |
|---|---|---|---|---|---|---|---|---|---|---|---|---|
| MAP | 0.644 ± 0.015 | 86.34 ± 0.36 | 0.093 ± 0.005 | 2.410 ± 0.012 | 55.38 ± 0.25 | 0.243 ± 0.003 | 0.232 ± 0.005 | 95.32 ± 0.16 | 0.028 ± 0.002 | 1.401 ± 0.014 | 71.26 ± 0.65 | 0.199 ± 0.022 |
| BBB | 0.697 ± 0.005 | 76.63 ± 0.32 | 0.071 ± 0.004 | 2.239 ± 0.005 | 41.07 ± 0.31 | 0.100 ± 0.001 | 0.218 ± 0.004 | 94.53 ± 0.24 | 0.047 ± 0.003 | 1.290 ± 0.143 | 71.55 ± 1.95 | 0.179 ± 0.019 |
| MCD | 0.534 ± 0.012 | 87.47 ± 0.24 | 0.084 ± 0.001 | 2.121 ± 0.020 | 59.28 ± 0.19 | 0.227 ± 0.001 | 0.207 ± 0.001 | 95.78 ± 0.19 | 0.026 ± 0.001 | 1.333 ± 0.024 | 72.28 ± 0.57 | 0.188 ± 0.027 |
| VD | 0.451 ± 0.028 | 87.68 ± 0.11 | **0.024 ± 0.008** | 2.888 ± 0.093 | 56.80 ± 0.77 | 0.284 ± 0.009 | 0.164 ± 0.006 | 96.11 ± 0.05 | 0.017 ± 0.006 | 1.084 ± 0.066 | 73.29 ± 0.47 | 0.084 ± 0.031 |
| ELRG | **0.382 ± 0.010** | 87.24 ± 0.37 | **0.018 ± 0.002** | 1.634 ± 0.019 | 58.14 ± 0.28 | **0.096 ± 0.005** | 0.145 ± 0.001 | 96.03 ± 0.05 | **0.003 ± 0.000** | 0.811 ± 0.018 | 73.66 ± 0.36 | **0.080 ± 0.012** |
| **VSD** | 0.464 ± 0.019 | 87.44 ± 0.50 | 0.061 ± 0.005 | **1.504 ± 0.011** | **60.15 ± 0.20** | 0.116 ± 0.002 | **0.140 ± 0.003** | **96.41 ± 0.03** | **0.003 ± 0.000** | **0.769 ± 0.013** | **74.50 ± 0.55** | **0.083 ± 0.008** |
| D.E | 0.488 ± 0.029 | **88.91 ± 0.67** | 0.069 ± 0.006 | 1.913 ± 0.015 | **60.16 ± 0.25** | 0.203 ± 0.004 | 0.171 ± 0.007 | 96.36 ± 0.11 | 0.020 ± 0.004 | 1.197 ± 0.017 | 73.16 ± 0.22 | 0.177 ± 0.025 |
| SWAG | **0.330 ± 0.025** | **88.77 ± 0.89** | 0.026 ± 0.007 | **1.417 ± 0.024** | **62.45 ± 0.49** | **0.028 ± 0.003** | **0.130 ± 0.005** | **96.72 ± 0.06** | 0.016 ± 0.003 | 0.843 ± 0.028 | 73.15 ± 0.44 | **0.069 ± 0.012** |

Table 14: The performance of VSD-Adam, VSD-SGD, and SWAG trained with AlexNet and ResNet18 on CIFAR10, CIFAR100 dataset. Results are averaged over 5 random seeds.

| AlexNet | CIFAR10 | | | CIFAR100 | | |
|---|---|---|---|---|---|---|
| | NLL | ACC | ECE | NLL | ACC | ECE |
| VSD with Adam | 0.656 ± 0.009 | 78.21 ± 0.153 | 0.046 ± 0.003 | 2.241 ± 0.026 | 46.85 ± 0.99 | 0.112 ± 0.010 |
| VSD with SGD | **0.579 ± 0.011** | **80.41 ± 0.275** | **0.010 ± 0.002** | **1.934 ± 0.018** | **51.68 ± 0.87** | 0.091 ± 0.009 |
| SWAG | 0.651 ± 0.014 | 78.14 ± 0.514 | 0.059 ± 0.003 | 1.958 ± 0.051 | 49.81 ± 0.62 | **0.028 ± 0.009** |

| ResNet18 | CIFAR10 | | | CIFAR100 | | |
|---|---|---|---|---|---|---|
| | NLL | ACC | ECE | NLL | ACC | ECE |
| VSD with Adam | 0.464 ± 0.019 | 87.44 ± 0.497 | 0.061 ± 0.005 | 1.504 ± 0.011 | 60.15 ± 0.20 | 0.116 ± 0.002 |
| VSD with SGD | 0.395 ± 0.022 | 87.17 ± 0.785 | **0.018 ± 0.004** | 1.440 ± 0.014 | 60.33 ± 0.45 | **0.015 ± 0.001** |
| SWAG | **0.330 ± 0.025** | **88.77 ± 0.889** | 0.026 ± 0.007 | **1.417 ± 0.024** | **62.45 ± 0.49** | 0.028 ± 0.003 |

## J  Further discussions and investigations

### J.1  Discussion of VSD and non-variational methods such as SWAG

In comparison with the non-variational baseline such as SWAG in Tables 3-4-5, VSD shows better performances on both predictive accuracy and uncertainty calibration on moderate deep models (FCs, LeNet). However, on more modern architectures (AlexNet, ResNet18), VSD performs worse than SWAG with noticeable gaps, especially on CIFAR10 and CIFAR100 dataset. Even on computational complexity, SWAG is also more appealing with no significant computational overhead compared to the conventional training schemes. Also, several recent reports have shown much better performances of SWAG and Deep ensemble than some variational inference methods (BBB, MC Dropout, etc.) in the field of deep learning uncertainty. Arguably, by considerable improvements of VSD compared to other variational methods, at least in the frame of reference for SWAG (and Deep ensemble), our method has made an important step to close the gap with non-variational baselines.

On the other hand, another important aspect worth noting that in our work, VSD is introduced as a general-purpose approach for Bayesian inference and in particular for BNNs. From that, we were not aiming for a state-ofthe-art in deep learning uncertainty (of SWAG baseline), but we instead proceed to demonstrate the effectiveness of VSD on many typical fronts: generalization, uncertainty calibration, robustness (OOD detection). We wish to emphasize that variational inference (VI) methods for BNNs always have a regime of their own for the meaningful problems they aim to address (compression or model selection for instance).

### J.2  Some additional investigations for VSD and improvements

VI represents a large Bayesian subfield on its own, although it unfortunately tends to lag behind non-VI methods when it comes to uncertainty in deep learning. Many questions for it remain controversial such as what is the best configuration for VI-BNN: weight parameterization, prior distribution, true and approximate posterior. And VSD obviously is also included in these problems. Inspired by the underlying principle of SWAG, which was developed on the previous work SWA - an optimization procedure guided by the Bayesian theoretical analysis of the stationary distribution of SGD iterations, we would investigate a similar perspective of optimization strategy for VSD. Indeed, optimizing a variational objective of BNN, in particular to VSD, has been a specialized problem. The current common algorithms, which adopt conventional gradients to give the direction of steepest descent of parameters in Euclidean space, might cause an unexpected difference in distribution space. This does not coincide with our original purpose of minimizing the distance between two high dimensional distributions in terms of KL divergence. In the literature of Bayesian deep learning, there are some relevant works using natural gradient [56, 68], which follows the direction of steepest descent in Riemannian space rather than Euclidean space, as a more appropriate solution. However, leveraging these algorithms is non-trivial that requires lots of complicated modifications in methodology and codebases.

We reckon that current optimization algorithms for VI in BNNs have not yet had an adequate theoretical investigation. A specific optimizer is non-ultimate and could have its own influence. Therefore, instead of using Adam optimizer for VSD as in our entire experiments, we here adopt

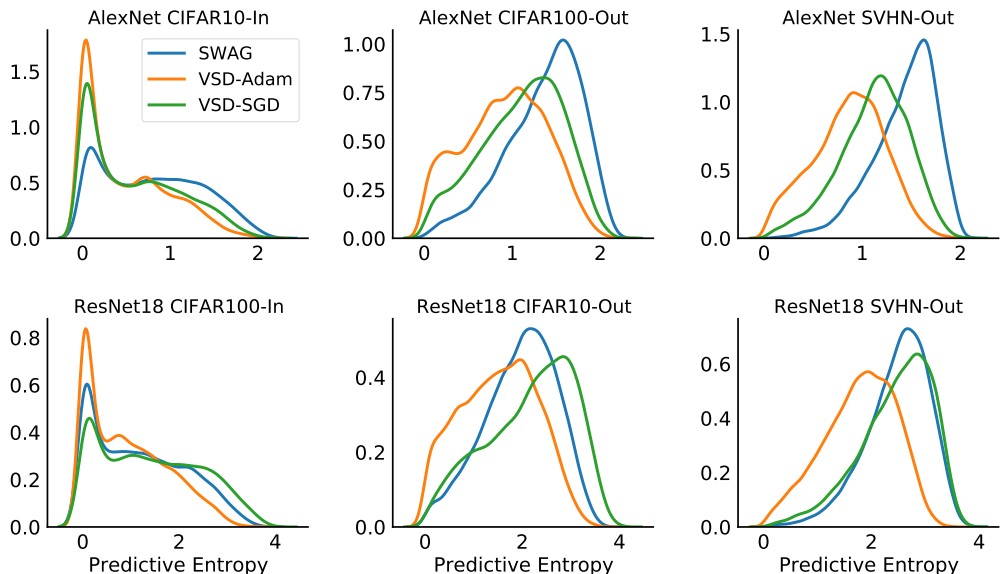

Figure 9: Histograms of predictive entropy for AlexNet (top) and ResNet18 (bottom) trained on CIFAR10 and CIFAR100 respectively.

SGD for VSD to elaborate some empirical observations and to compare with SWAG. We provide below some results as follows:

**Classification results.** We trained VSD by SGD-momentum in the experiment using AlexNet and ResNet18 with CIFAR10 and CIFAR100 dataset (SWAG outperforms VSD mainly on these settings in our paper). The results is given in Table 14, VSD with SGD gains better performances than SWAG on AlexNet, meanwhile on ResNet18 VSD with SGD shows slight improvements compared to using Adam. These observations consolidate our arguments of the influence of different optimizers, and specifically in this experiments SGD is more effective than Adam. Thus, understanding and devising efficient new optimization algorithms for VI in BNNs is necessary and promising.

**Predictive entropy performance.** We evaluated the predictive entropy performance of VSD-SGD compared to VSD-Adam and SWAG. We utilized the same settings in Figure 4 and show the new results in Figure 9. Interestingly, we find that VSD with SGD not only reduces the gap with SWAG on quantitative metrics, but also exhibits similar behavior on qualitative metrics. In the comparison between SWAG and VSD-Adam, we could observe a common trend, of which SWAG tends to maintain sufficient confidence on in-distribution data and provide higher uncertainties on out-of-distribution data (even on AlexNet, SWAG seems to be underconfident for in-distribution data). On the other hand, when using SGD optimizer, VSD archives a better balance (more well-calibrated) than VSD-Adam and SWAG on AlexNet, meanwhile on ResNet VSD-SGD tends to resemble the SWAG's behavior.

We once again argue that the optimization strategy could provide a compelling explanation for this phenomenon.

**1.** SWAG uses SGD with a cyclical or constant learning rate schedule to explore an optimal set of points corresponding to high-performing networks within a basin of attraction. SWAG then estimates the first two moments of the points in this optimal set to form an empirical mean and covariance of a Gaussian distribution. This means that Monte Carlo sampled models from the SWAG posterior have meaningfully different representations which provide complementary explanations for the data. Thus, when ensembling these models at the test time, we could obtain an average predictive probability that is not concentrated excessively on some certain classes (on in-distribution data, due to the high-performing essence, this average representation is still compelling for prediction). As a result, we get predictive entropy with larger values and thus explain the above behavior of SWAG.

**2.** For VSD-SGD, there is some evidence for its performance including: firstly, SGD minima is located in the same basin of attraction with the mean of SWAG posterior, especially SGD generally converges to a flat optima near the boundary of the manifold of the optimal set we mentioned above,

thus the local geometric information around SGD optima probably contains some similar properties as SWAG mean; and secondly, SGD with the implicit regularization has been proven to often find drastically different (and more desirable) solutions than Adam-like adaptive methods [85].

**Conclusion.** In the accuracy regime, many studies have pointed out the favor of SGD over adaptive optimizers [34, 93]. Our above observations further also advocate using SGD in particular for uncertainty calibration. This is consensual to the deep learning uncertainty community, in which SGD usually has been adopted as an appropriate baseline rather than Adam. Furthermore, we do believe that variational inference has been a compelling direction for learning BNNs, and exploring efficient optimization strategies to it is very promising for future works.

## K   Changes in the camera-ready version compared to the submitted version

- We added one more contribution in Section 1 to further highlight the primary purpose of our work.

- In Section 3.2, we provided more discussions about the plausibility of the new variational objective, as well as gave concrete intuitions on the choice of the number of transformation steps $T$. Some minor suggestions from reviewers were also added.

- In Section 3.3, we included the detailed description hyper-distribution $p(z)$ and $q(z)$.

- In Appendix A.2, we updated the role of hierarchical parameterization in terms of enforcing a stronger regularization.

- In Appendix B, we provided discussion about the effect of Empirical Bayes in our method.

- In Appendix D, we analyzed more thoroughly the effect of $T$ on VSD's practical runtime.

- In Appendix I.1, we provided more insights about the KL annealing/tempered posterior.

- In Appendix I.4, we made it clear why we only did Bayesian inference for fully connected layers of LeNet architecture.

- In Appendix I.5, we provided some additional comparisons between VSD and VOGN.

- In Appendix I.6, we added the standard deviations for empirical classification results in Tables 3-4-5.

- In Appendix J, we provided further discussions of VSD and SWAG, and also introduced some meaningful investigations.