# OpenReview forum: "Structured Dropout Variational Inference for Bayesian Neural Networks"
_NeurIPS.cc/2021/Conference — NeurIPS 2021 Poster_

### Official Review · Reviewer_6pWo · 2021-07-13

**Rating:** 5
**Confidence:** 4

**Summary:**

This paper presents a novel variational structured approximation of the posterior distribution of Bayesian neural networks by the Bayesian interpretation of Dropout regularization.   An orthogonal transformation is used to learn a structured representation of the variational noise and consequently induces statistical dependencies in the approximate posterior.

**Limitations And Societal Impact:**

1. There is a lack of comparison with non-variational baselines SWAG and SWAG in terms of predictive uncertainty in the out-of-distribution settings, as shown in Figure 3.

2. The results in Table 1-3 lacks the variances for the random runs.



**Main Review:**

Originality: The method is new and interesting.

Quality: Technically sound with extensive experiments.

Clarity: Clearly written.

Significance: Compared with previous variational-based methods, the proposed VSD achieves good results by incurring structure into the covariance, which is expected and reasonable.  The main concern is its importance of practical use. For example, in comparison with the standard non-variational baseline such as SWAG, which serves as a simple baseline for Bayesian uncertainty in deep learning,  the results in Table 2 and Table 3 show that in most cases the proposed VSD performs worse than SWAG. It then raises the long-standing question that why the Bayesian methods, though with high complexity,  do not even yield competitive results as compared with simpler methods. It would be appreciated if a comprehensive comparison of VSD and the baseline SWAG, which is simple modification of non-bayesian methods to obtain uncertainty estimates,  is further clarified.

**Time Spent Reviewing:**

3 hours

---

> ### Author Response · Authors · 2021-08-10
> **Response to Reviewer 6pWo**
>
> We would like to thank you for your comments and suggestions. Below, you can find our responses to your reviews:
>
> **Question 1.** "*The main concern is its importance of practical use. For example, in comparison with the standard non-variational baseline such as SWAG, which serves as a simple baseline for Bayesian uncertainty in deep learning, the results in Table 2 and Table 3 show that in most cases the proposed VSD performs worse than SWAG.
> It then raises the long-standing question of why Bayesian methods, though with high complexity, do not even yield competitive results as compared with simpler methods.*"
>
> **Response:** Thanks for your comment.
>
> Before answering your concerns, we would like to take an opportunity to further highlight our proposals in this work. First of all, we would appreciate it if the reviewer could refer to our first answer in the rebuttal to Reviewer 3, of which we have made our motivations and contributions more prominent. Arguably, besides the theoretical aspects presented in the paper, the substantial experimental improvements of our method over other variational methods have shown that VSD has made an important step to close the gap with non-variational baselines. Especially on moderate deep models, VSD exhibits better performances on both generalization and uncertainty quantification.
>
> Going back to your concerns. VI represents a large Bayesian subfield on its own, although it unfortunately tends to lag behind non-VI methods when it comes to uncertainty in deep learning. Many questions for it remain controversial such as what is the best configuration for VI-BNN: weight parameterization, prior distribution and approximate posterior. However, we instead would like to discuss the perspective of optimization strategy, which we hope that our responses satisfactorily address your concerns somewhat.
> Optimizing a variational objective of BNN has been a specialized problem. The current common algorithms, which adopt conventional gradients to give the direction of steepest descent of parameters in Euclidean space, might cause an unexpected difference in distribution space. This does not coincide with our original purpose of minimizing the distance between two high dimensional distributions in terms of KL divergence. In the literature of Bayesian deep learning, there are some great works using natural gradient [1, 2], which follows the direction of steepest descent in Riemannian space rather than Euclidean space, as a more appropriate solution. However, leveraging these algorithms is non-trivial that requires lots of complicated modifications in methodology and codebases. On the other hand, the SWAG baseline was developed on the previous work SWA - an optimization procedure inspired by the Bayesian theoretical analysis of the stationary distribution of SGD iterations. While SWAG is also a local approximation, we reckon that its remarkable performance might be highly dependent on the utilizing SWA solution as the first moment.
>
> Motivated by these above arguments, we suppose that adopting a different optimization algorithm in particular for VSD can provide some interesting findings. Instead of using Adam optimizer as in our entire paper, we train VSD by SGD+momentum on AlexNet and ResNet18 with CIFAR10 and CIFAR100 dataset (SWAG outperforms VSD mainly on these settings in our paper). We temporarily put the results of this additional experiment in the following anonymous link: https://imgur.com/a/MGM7vWR
>
> Given the results in tables in the link, VSD with SGD gains better performances than SWAG on AlexNet, meanwhile on ResNet18 VSD with SGD only shows slight improvements compared to using Adam. These observations consolidate our arguments of the influence and the non-ultimate of different optimizers, and thus devising more effective optimization algorithms for VI in BNNs is necessary and promising.
>
> [1] Osawa et al., "Practical Deep Learning with Bayesian Principles", NeurIPS 2019
>
> [2] Mishkin et al., "SLANG: Fast Structured Covariance Approximations for Bayesian Deep Learning with Natural Gradient", NeurIPS 2018.
>
> **Question 2.** "*It would be appreciated if a comprehensive comparison of VSD and the baseline SWAG, which is a simple modification of non-bayesian methods to obtain uncertainty estimates, is further clarified.*"
>
> **Response:** Thanks for your suggestion. We provide here some additional comparisons of VSD and SWAG.
>
> 1. The first one is a regression experiment on small UCI datasets. We still evaluate the performance on two criteria of RMSE and test log-likelihood. We temporarily put the results in the following anonymous link: https://imgur.com/a/ftUKPtu. The results show that VSD has a better performance than SWAG on both metrics. This is consistent with our other experiments on moderate architectures.
>
> 2. The second evaluation is on out-of-distribution detection metrics which has been given in Table 10 of Appendix H.4. We temporarily put the results of SWAG in the following anonymous link: https://imgur.com/a/jaynHH5. The results show that SWAG and VSD have comparable performances in almost all settings (on AlexNet with SVHN dataset, SWAG presents more noticeable results). We will add these results in the revision of the paper.
>
>
> **Question 3.** "*Lack of comparison with SWAG in terms of predictive uncertainty in the out-of-distribution settings, as shown in Figure 3. The results in Table 1-3 lack the variances for the random runs.*"
>
> **Response:** Thanks for your comment.
>
> *** Regarding the predictive entropy performance shown in Figure 3, we add more comparisons of VSD and SWAG baseline in the following anonymous link: https://imgur.com/a/tNq8nOu. The results show that: in the experimental scenario on LetNet and ResNet18 architecture, VSD and SWAG exhibit reasonable and comparable performance, while on AlexNet, SWAG tends to be underconfident even for in-distribution data. We will add these results in the revision of the paper.
>
> *** Regarding the standard deviations of different random seeds. For the convenience of presentation, we have not included the standard deviations of empirical results in the main text. We will add the detailed table in Appendix G in the revised version as suggested by the reviewer. Temporarily, we provide tables with standard deviations in the following anonymous link: https://imgur.com/a/odOuYdK

---

> > ### Comment · Reviewer_6pWo · 2021-08-21
> > **Thanks for the response and several additional questions.**
> >
> > Thanks for the authors' detailed response with experimental results, which are really appreciated. Regarding the responses, I have several questions to consult.
> >
> > 1. Improvement of VSD with SGD in Response 1:
> >
> > In the results of https://imgur.com/a/MGM7vWR, it is interesting that VSD with SGD outperforms Adam so that the gap between SWAG is reduced or even reversed. Then, how about the results of SWAG with SGD (as opposed Adam) with proper tuning? It would be fairer to use the same optimizer for comparison.
> >
> > 2. Additional comparisons of VSD and SWAG  in Response 2:
> >
> > Thanks for providing such detailed results.
> >
> > First, what are the optimizers used for VSD and SWAG in these two results? Are they based on the same optimizer, or are they the best results out of different optimizers?
> >
> > Second, what I mean by "a comprehensive comparison of VSD and the baseline SWAG"  is not restricted to the comparison of the performances but also other related issues, such as complexity, memory, ease to implementation, compatibility with the existing deep learning platforms, etc.  The reason is that, from current results, it seems that the performance of VSD is not very competitive compared to the simple baseline SWAG. In Table 4 and 5 in the appendix, there are some comparisons of complexity and running time with other algorithms but still a lack of comparison with SWAG.
> >
> > 3.  Out-of-distribution settings comparison  in Response 3:
> >
> > First, what are the optimizers used in https://imgur.com/a/tNq8nOu. Second, it is interesting that "on AlexNet, SWAG tends to be underconfident even for in-distribution data", could the authors provide some explanations? I noted that apparently, VSD tends to be much overconfident for the out-distribution data for AlexNet than SWAG, which is also the case for ResNet18. However, the trend is the opposite for LeNet, but the differences between SWAG and SVD are much smaller. Is there any explanation for such diverse results?

---

> > > ### Author Response · Authors · 2021-08-25
> > > **Response to Reviewer 6pWo, Part 2**
> > >
> > > We would like to thank you for specifically reviewing our responses and providing additional questions. Before answering your concerns in detail, we wish to highlight two central issues, which we think are crucial and will help the discussion become more consensual.
> > >
> > > **i. The scope of the research in our work and the roles of SWAG baseline in our empirical evaluation.**
> > >
> > > --- First of all, by the presentation in our paper which we think are consistent, we have introduced our method-VSD as a general-purpose approach for Bayesian inference and in particular for Bayesian neural networks (BNNs). More specifically, we aim to develop an efficient and scalable Bayesian inference framework on both criteria: expressive approximation and flexible modeling (via hierarchical parameterization). Then, **as an instance of variational Bayesian inference in deep neural nets**, we proceed to demonstrate the effectiveness of VSD on many typical fronts: generalization, uncertainty calibration, robustness (OOD detection). We wish to emphasize that VI methods for BNNs always have a regime of their own for the meaningful problems they aim to address. Indeed, in the analyses of prior hierarchy in Section 3.3 and Appendix C, we *intentionally mentioned* another aspect of great significance, which are features sparsity and model selection:
> > >
> > > - From the literature, we find that VSD embodies some compelling principles which are promising to acquire features sparsity or facilitate model selection. Basically, we can do structured compression by specifying a threshold for variational dropout rate (namely variance of the structured noise) to prune neurons/nodes or convolutional channels of the networks. With hierarchical parameterization, we can flexibly adjust the hyper prior, such as design a *per-layer* prior, and then perform a procedure of automatic depth determination [3] for the residual networks. For an example of practical use, these potentials could be useful for the problem of resource allocation, such as in Bayesian Continual Learning community.
> > >
> > > On the other hand, some approaches to deep learning uncertainty such as SWAG, Deep Ensemble were not devised (probably lacking theoretical principles) for the above purpose. This is a prominent viewpoint indicating that the primary purposes in our work could somewhat distinguish our method (and in general, VI methods for BNNs) from the SWAG proposal
> > >
> > > --- About the roles of SWAG (and also Deep ensemble) in our empirical evaluation. Because of the fact that recent reports have shown much better performances of SWAG and Deep ensemble than some variational inference methods (BBB, MC Dropout, etc.) in the field of uncertainty calibration, so it is timely and makes sense to include SWAG and Deep ensemble in our standard experimental comparisons. Most importantly, we are not aiming for a state-of-the-art in deep learning uncertainty, but we instead would like to point out the considerable improvements of VSD compared to other variational methods, at least in the frame of reference for SWAG and Deep ensemble. In other words, VSD has made an important step to close the gap with non-variational baselines.
> > >
> > > In summary, the above analyses are the main reasons for us to think that the comparisons with SWAG and Deep ensemble in Table 1,2,3,8 in the paper are fairly adequate for our assessments.
> > >
> > > [3] Nalisnick et al., "Dropout as a Structured Shrinkage Prior", ICML 2019
> > >
> > > **ii. The optimization algorithm used in SWAG.**
> > >
> > > --- As we already mentioned in the initial rebuttal, the SWAG method used informative iterations from SGD trajectory to approximate the weight posterior in the form of a Gaussian distribution. SWAG is motivated by theoretical analysis of the stationary distribution of SGD iterations, specifically in the low-dimensional subspace spanned by SGD iterates, the shape of posterior distribution is approximately Gaussian. In detail, SWAG first trains the network with conventional SGD (use momentum + a standard decaying learning rate schedule) until reaching the low-loss region around minima. After this pre-training, SWAG uses SGD with a cyclical or constant learning rate schedule to explore an **optimal set** of points corresponding to **high-performing** networks within a basin of attraction. SWAG then estimates the first two moments of the points in this optimal set to form an empirical mean (which is also SWA solution [6]) and covariance of a Gaussian distribution.
> > >
> > > --- So, in our entire experiments (in the main paper and rebuttal), we use SGD as a base optimizer for SWAG. Although it is possible to try using Adam optimizer for empirical examinations, this does not make much sense because it does not meet with the theoretical motivations and original methodology of SWAG. On the other hand, as we mentioned before, optimizing the variational objective in particular for VSD has been a very different matter, because a specific optimizer has not yet had an adequate theoretical investigation. Therefore, we think it's a fair comparison when instead of using Adam optimizer as in our current paper, we adopt SGD to elaborate some properties as in our initial responses to the reviewer's concerns. We think it would be useful to add some results of VSD with SGD in the Appendix as a further discussion.

---

> > > > ### Author Response · Authors · 2021-08-25
> > > > **Response to Reviewer 6pWo, Part 3**
> > > >
> > > > **Going back to your questions, we would like to answer you as follows:**
> > > >
> > > > **Question 1**. "*In the results of https://imgur.com/a/MGM7vWR, how about the results of SWAG with SGD (as opposed Adam) with proper tuning? It would be fairer to use the same optimizer for comparison.*"
> > > >
> > > > **Response:** In these results, as we mentioned in (ii)-Part-2, SWAG was implemented with the SGD optimizer.
> > > >
> > > > **Question 2.** "*In additional comparisons of VSD and SWAG in Response 2 of the author's initial rebuttal*"
> > > >
> > > > **Response:**
> > > >
> > > > *** Regarding the optimizers used for VSD and SWAG in those two results: In the regression experiment on small UCI datasets, SWAG was implemented with the SGD, meanwhile VSD used Adam (results inherited from Table 8 in Appendix G.2). In OOD metrics results, both SWAG and VSD were implemented with the SGD and showed comparable results.
> > > >
> > > > *** Regarding a comparison of complexity and running time with SWAG: SWAG is a practical method, easy to implement, and has virtually no significant computational overhead compared to the conventional training schemes. However, as we discussed in (i)-Part-2 about the scope of the research in our work and the roles of SWAG baseline in our comparison, it is *purely intentional* when we only compared the complexity and running time of VSD with various variational methods.
> > > >
> > > > **Question 3.** "*Out-of-distribution settings comparison in Response 3 of the author's initial rebuttal.*"
> > > >
> > > > **Response:**
> > > >
> > > > *** Regarding the optimizers used in https://imgur.com/a/tNq8nOu. SWAG was implemented with the SGD, meanwhile VSD used Adam (results inherited from Figure 4 in Appendix G.2)
> > > >
> > > > *** Regarding the empirical observation in predictive entropy performance in https://imgur.com/a/tNq8nOu. We would like to answer the reviewer through two main concerns: (1) assess the quality of uncertainty estimates via predictive entropy density; and (2) explain the behaviors of SWAG. Specifically as follows:
> > > >
> > > > **(1) assess the quality of uncertainty estimates via predictive entropy density**
> > > >
> > > > Basically, an accurate and well-calibrated model is expected to represent entropy values being concentrated mostly around 0 (i.e. high confidence) when the test data comes from the same underlying distribution as the training data, and in the opposite case, the predictive entropies should be evenly distributed (i.e. higher uncertainty). In fact, the deep learning models do not achieve simultaneously on both expectations at the most ideal, but instead, accurate and well-calibrated ones tend to exhibit a moderate level of confidence on in-distribution data, and then provide a **reasonable representation** for uncertainty estimates on out-of-distribution data. This can be seen as a trade-off (shown clearly in Figure 3-4 in the experiments in our paper) and thus a balanced state should be favored. In the comparison of VSD and SWAG on AlexNet, we reckon VSD is more well-balanced than SWAG, qualitatively: SWAG is quite underconfident on in-distribution data, but VSD is not much overconfident on out-of-distribution data, which is also the case for ResNet18.
> > > >
> > > > **(2) explain the behaviors of SWAG**
> > > >
> > > > Intuitively, we can find that VSD and SWAG show almost similar results on LeNet, meanwhile on AlexNet and ResNet18, these two methods exhibit a common trend, of which SWAG tends to maintain sufficient confidence on in-distribution data and provide higher uncertainties on out-of-distribution data.
> > > >
> > > > --- For the case of LeNet, because on SVHN dataset, VSD and SWAG both achieve comparable and noticeable performances on all three measures (NLL, ACC, ECE), leading to excellent uncertainty quantification. We try to conduct an alternative experiment by using CIFAR10 as in-distribution data. The result is given in the anonymous following link: https://imgur.com/a/8JlQRFx. We can observe a consistency between different architectures, but it is also worth noting that the difference between VSD and SWAG on LeNet is still smaller than on AlexNet and ResNet18. This can be partly explained through a trend in the literature: the smaller networks have better uncertainty calibration than the modern ones [4].
> > > >
> > > > --- The rest is why the SWAG method often represents flatter entropy densities than that of VSD (on both in-distribution and out-of-distribution data). We once again argue that the optimization strategy provides a compelling explanation for this phenomenon. Going back to (ii)-Part-2 about the optimization algorithms used in SWAG. The Gaussian posterior in SWAG is approximated in a low-dimensional subspace spanned by **high-performing but diverse** networks (proved through the line of works including [5-6]). This means that (Monte Carlo) sampled models from the SWAG posterior have meaningfully different representations which provide complementary explanations for the data. Thus, when ensembling these models at the test time, we obtain an average predictive probability that is not concentrated excessively on some certain classes (on in-distribution data, due to the high-performing essence, this average representation is still compelling for prediction). As a result, we get predictive entropy with larger values and thus explain the above concern about SWAG's behavior. Of course, there could be many other reasons for the behavior of SWAG, so to reinforce our argument, we have conducted more thorough investigations below.
> > > >
> > > > We on the other hand argue that leveraging SGD as a base optimizer can provide some insightful findings. There is some evidence for this including: firstly, SGD minima is located in the same basin of attraction with SWA solution (i.e. mean of SWAG posterior), especially SGD generally converges to a flat optima near the boundary of the manifold of the **optimal set** we mentioned in (ii)-Part-2 [6], thus the local geometric information around SGD optima probably contains some similar properties as SWA; and secondly, SGD with the implicit regularization has been proven to often find drastically different (and more desirable) solutions than Adam-like adaptive methods [7]. Promoted by these analyses, we implement two experiments as follows:
> > > >
> > > > - We evaluated the predictive entropy performance of VSD with SGD as in our previous settings. The result is given in this anonymous link: https://imgur.com/a/93MLypF (we adjusted the y-axis in ResNet18 results to make the figure clearer). Arguably, on AlexNet, VSD+SGD archives a better balance (more well-calibrated) than VSD+Adam and SWAG, meanwhile on ResNet VSD+SGD tends to resemble the behavior of SWAG.
> > > >
> > > > - We implemented Laplace approximations around Adam optima and SGD optima to form two Gaussian posteriors respectively. In which, we approximate the precision parameter by a diagonal Fisher information matrix. The result is given in the anonymous following link: https://imgur.com/a/vo4zS5L. Interestingly, SGD-Laplace consistently exhibits a trend of maintaining more modest levels of confidence than that of Adam-Laplace. Especially, on out-of-distribution data, while Adam-Laplace often has a tendency to be overconfident, SGD-Laplace still represents reasonable uncertainties with larger entropies. We do not claim the results of SGD-Laplace are always better than that of Adam-Laplace in general, however, this behavior of SGD is favorable for **robust** uncertainty calibration.
> > > >
> > > > In the accuracy regime, many studies have pointed out the favor of SGD over adaptive optimizers [8,9]. Our above observations further also advocate using SGD in particular for uncertainty calibration. This is consensual to the deep learning uncertainty community, in which SGD usually has been adopted as an appropriate baseline rather than Adam.
> > > >
> > > > [4] Guo et al., "On Calibration of Modern Neural Networks", ICML 2017.
> > > >
> > > > [5] Garipov et al., "Loss Surfaces, Mode Connectivity, and Fast Ensembling of DNNs", NeurIPS 2018.
> > > >
> > > > [6] Izmailov et al., "Averaging Weights Leads to Wider Optima and Better Generalization", UAI 2018.
> > > >
> > > > [7] Wilson et al., "The Marginal Value of Adaptive Gradient Methods in Machine Learning", NeurIPS 2017.
> > > >
> > > > [8] Keskar et al., "Improving Generalization Performance by Switching from Adam to SGD", 2017.
> > > >
> > > > [9] Zou et al., "Towards Theoretically Understanding Why SGD Generalizes Better Than ADAM in Deep Learning", NeurIPS 2020

---

> > > > > ### Comment · Reviewer_6pWo · 2021-08-27
> > > > > **Thank you for your so detailed responses.**
> > > > >
> > > > > Thank you very much for the detailed responses. I really appreciate the authors' effforts in clarifying my concerns.
> > > > >
> > > > > >Most importantly, we are not aiming for a state-of-the-art in deep learning uncertainty, but we instead would like to point out the considerable improvements of VSD compared to other variational methods, at least in the frame of reference for SWAG and Deep ensemble. In other words, VSD has made an important step to close the gap with non-variational baselines.
> > > > >
> > > > > I agree with this clarification and suggest to add a brief note in the manuscript to clarify this.
> > > > >
> > > > > >Regarding a comparison of complexity and running time with SWAG: SWAG is a practical method, easy to implement, and has virtually no significant computational overhead compared to the conventional training schemes. However, as we discussed in (i)-Part-2 about the scope of the research in our work and the roles of SWAG baseline in our comparison, it is purely intentional when we only compared the complexity and running time of VSD with various variational methods.
> > > > >
> > > > > I think it is kind of ok with such an  _intentional_ comparision for your special goal. However, it is important to point out this and discuss the limitations (e.g., compared with SWAG) of the proposed method in detail. Otherwise, it might be slightly misleading. Note that, as shown in the checklist, there is a lack of  descriptions of the limitations of the proposed method in the manuscipt. .
> > > > >
> > > > > >In the comparison of VSD and SWAG on AlexNet, we reckon VSD is more well-balanced than SWAG, qualitatively: SWAG is quite underconfident on in-distribution data, but VSD is not much overconfident on out-of-distribution data, which is also the case for ResNet18.
> > > > >
> > > > > Thank you very much for providing such a detailed analysis of the OOD results. I agree with most the above analysis and only disagree with this point a bit, especially in the case of ResNet 18. In fact, for ResNet 18, qualatively, the comparision of VSD vs  SWAG is quite similar to that of BBB (or MC dropout) vs. VSD in figure 4 of the manuscript. However, for the latter it is described as "VSD gains better results with entropy values being distributed over a larger support, meaning that the predictions of VSD are closer to uniform on unseen classes", as opposed to say that BBB is more balanced (I  noted that BBB is worse in other cases but here only focuses on the specific case and its description) when claiming the superiority of VSD vs  SWAG.  As no quantitive metric is given, it is better to have consistent qualitative descriptions.
> > > > >
> > > > > Anyway, I suggest to add results of SWAG for a full comparison explicitly in the manuscript.
> > > > >
> > > > > Thank you again for your thorough analysis and feedback.

---

> > > > > > ### Author Response · Authors · 2021-08-28
> > > > > > **Thanks for the recognition and a little explanation**
> > > > > >
> > > > > > We thank the reviewer very much for recognizing our efforts and providing useful comments.
> > > > > >
> > > > > > --- We would also like to explain a bit on your comment/concern in "*...for ResNet18, qualitatively, the comparison of VSD vs SWAG is quite similar to that of BBB (or MC dropout) vs VSD in Fig.4 of the manuscript...*". We guess you mean the result in Fig.3 in the main paper rather than in Fig.4 in Appendix, because the description the reviewer mentioned is for Fig.3, and furthermore in Fig.4, BBB and MCD both exhibit uncertainty estimates which are not really reasonable (dissimilar to the case of VSD vs. SWAG). In Fig.3 on LeNet, your comment is intuitively sensible. However, we believe there is a critical difference between the behaviors of *VSD vs. SWAG on ResNet18* and *BBB (or MCD) vs. VSD on LeNet* regarding **the range of density values represented on the y-axis**, from which predictive entropy of VSD-in-distribution on LeNet is still properly concentrated around zero with high density values, but SWAG on ResNet18 is not the case. Of course, we agree with the reviewer that since no quantitative metric is given, it should be cautious in making qualitative assessments.
> > > > > >
> > > > > > We will include your suggestions in the revised version of our paper: a brief clarification of our primary purpose, a consistent qualitative description for predictive entropy performance, and results of SWAG for a full comparison (additional results in the rebuttal, complexity with discussion).
> > > > > >
> > > > > > --- Finally, we would appreciate it if you could let us know whether you still have any concerns and whether you will increase your score. Thank you again.

---

> ### Author Response · Authors · 2021-08-30
> **Feedback before the discussion ends**
>
> Dear Reviewer 6pWo,
>
> We would like to thank you again for spending your time evaluating our paper and for posing several important questions during the rebuttal period.
>
> As the discussion period is expected to conclude early this week, we really appreciate it if you could let us know if you still have any other concerns with the paper and whether you will consider changing your rating based on our detailed responses to clarify your earlier concerns.
>
> Best,
>
> Authors

---

### Official Review · Reviewer_VNbM · 2021-07-15

**Rating:** 5
**Confidence:** 5

**Summary:**

The paper proposes a new extension to the well-known and widely used Monte-Carlo dropout scheme as variational approximation to the Bayesian posterior.


**Limitations And Societal Impact:**

Limitations and social impact are not discussed.

**Main Review:**

The paper is overall difficult to read and understand, especially the discussion on the different parameterization choices.
Inspired by the MC-dropout, the Authors end up with a very involved parameterization of the posterior and to achieve this they propose a lot of incremental improvements and changes, which are not well motivated, in my opinion.

Below some detailed comments to justify my overall score:

- From the presentation of the Householder parameterization in Sec. 3.1 (and around line 136), it seems that the $\Sigma$ is full-rank.
This is possible only when the number of transformations via $H$ (T) is equal to the dimensions of $\Sigma$ (K).
In other cases, this could be considered as low-rank parameterization of the covariance.
- The parameterization of the vectors $v_t$ as a flow (i.e. $v_t = f(v_{t-1})$) is not discussed in the paper, but only after line 528 in the Appendix.
This is an important modeling choice that should have been present in the main paper.
- The Authors propose to run an empirical Bayes procedure for choosing the prior parameters but the effect of this choice is not assessed. Minor: From the discussion around line 192 it might be confusing for a reader why $\beta$ does not appear in Eq. 5; maybe it would be better to first show the full KL (Eq. 11 in Appendix) and then show the optimized one. As an alternative, you could use a different symbol for the optimized KL (like $\tilde{\mathbb{D}}_{KL}$).
- Still regarding the empirical Bayes, given that the $\beta$ is a matrix with same dimensions of $\mathbf W$, would it be possible for the KL to collapse to 0? In the minimization of the variational objective, all it needs is for $\alpha_j \rightarrow 0$ and it's not clear how this could be prevented. Did you experienced this phenomenon in your experiments?
- Connected with the point above, I can see why the Authors needed a hierarchical parameterization of the prior to enforce a stronger regularization. Regarding the hierarchical prior, the main paper doesn't discuss the parameterization of $p(z)$ and $q(z)$ (some details are present in the Appendix but should be reported also here).
Minor: For Figure 2 with the double y-axis it's not possible to understand which axis is corresponding to which dataset.

Two important remarks/contradictions after checking the submitted code, which are not discussed anywhere:

- From the presentation of Sec. 4.2 I seems that the Authors used this approach also for convolutional layers. This indeed is not what is happening, or at least this is not what is implemented in the submitted code, where the weights of the convolutional layers are simply optimized. The extension from fully-connected layers to convolutional ones should be fairly simple.
- From the interpretation of line 233, I seems that the Authors will finally consider only the version with the hierarchical prior as their final proposal but, again, the code implements only the previous one (see for example classification/model.py:162, which implements only Eq. 5)




**Time Spent Reviewing:**

10

---

> ### Author Response · Authors · 2021-08-10
> **Response to Reviewer VNbM, Part 1**
>
> We really appreciate the reviewer for taking considerable time to evaluate our work. Due to limited space in the main paper, some of our slightly unreasonable arrangements led to confusion for the reviewer.
>
> **Question 1.** "*The paper proposes a new extension to the well-known and widely used Monte-Carlo dropout scheme as variational approximation to the Bayesian posterior. Inspired by the MC-dropout, the Authors end up with a very involved parameterization of the posterior and to achieve this they propose a lot of incremental improvements and changes, which are not well-motivated, in my opinion.*"
>
> **Response:** Thanks for your comments. We would like to take this opportunity to clarify more about the motivations and contributions in our work, which might not have been fully understood by the reviewer.
>
> Our proposal VSD is built upon the Variational Dropout (VD) method which adopted multiplicative Gaussian noise for the Dropout procedure, whereas the MC Dropout method the reviewer mentioned used Bernoulli noise. In the motivation part from Line 43, VSD has been promoted by two primary motivations:
>
> 1. acquiring complementary benefits of the flexible Bayesian inference and Dropout inductive bias,
>
> 2. overcoming significant challenges of VD including the expressiveness of approximate inference which leads to underestimating model uncertainties [1], and the singularity issue of Dropout posterior distribution which causes the ill-posed inference [2].
>
> The proposals in our work, involving orthogonal parametrization and prior hierarchy, have theoretically addressed the aforementioned drawbacks of the VD method. We have thoroughly elaborated these in the whole of Section 3. And more importantly, we have achieved an expressive and scalable Bayesian inference framework (detailed analyses given in Appendix A&D) with lots of intriguing inductive bias which contributes to better generalization (detailed analysis given in Section 3.5 and Appendix E), in particular for Bayesian neural nets.
>
> [1] Foong et al., "On the Expressiveness of Approximate Inference in Bayesian Neural Networks", NeurIPS 2020.
>
> [2] Hron et al., "Variational Bayesian dropout: pitfalls and fixes", ICML 2018.
>
> **Question 2.** "*From the presentation of the Householder parameterization in Sec. 3.1 (and around line 136), it seems that the $\Sigma$ is full-rank. This is possible only when the number of transformations via $H(T)$ is equal to the dimensions of  $\Sigma (K)$. In other cases, this could be considered as low-rank parameterization of the covariance.*"
>
> **Response:**  Thanks for your comment.
>
> By low-rank parameterization, we reckon the reviewer was referring to the role of the number of transformation steps $T$ in representing an arbitrary orthogonal matrix by the Householder transformations. If so, we would answer the reviewer as follows: as we mentioned in Line 132 to 136, by the basis-kernel representation theorem [3], any orthogonal matrix $P$ with the basis acting on the $T^*$-dimensional subspace, namely orthogonal matrices of degree $T^*$, can be expressed as a product of exactly $T^*$ Householder transformations. The degree $T^*$ of the orthogonal matrix $P$ is $\leq$ its order $K$. So, the number of transformation steps $T=T^* (\leq K)$ is sufficient to be capable of representing the orthogonal matrix $P$ of degree $T^*$.
>
> For deep learning architectures, the value of $K$, and so $T^*$, is relatively large (in our method, $K$ is the size of the hidden unit for fully connected layers, and the number of channels for convolutional layers), so it is quite challenging to adjust the value of $T$ in a principled way to meet the theory. We can employ an efficient parameterization introduced in [4], in which we only need the Householder vectors $\\{v_t, t \geq 1\\}$ with sizes much smaller than $K$, and this will facilitate tuning $T$ with a larger range in an applicable computation time. In our work, we instead only adopt a small $T \in \\{1,2,3\\}$  and to make expressive reflections, we use a fully connected layer between successive Householder vectors $\\{v_t, t \geq 0\\}$. Indeed, this neural parameterization has been successfully implemented in the context of deep latent-variable models [5] as we mentioned in Line 197.
>
> [3] Xiaobai Sun and Christian Bischof, "A Basis-Kernel Representation of Orthogonal Matrices", 1995.
>
> [4] Zhang et al., "Stabilizing Gradients for Deep Neural Networks via Efficient SVD Parameterization", ICML 2018
>
> [5] Berg et al., "Sylvester normalizing flows for variational inference", UAI 2018.
>
> **Question 3.** "*The parameterization of the vectors $v_t$ as a flow (i.e. $v_t = f(v_\{t-1\}) )$ is not discussed in the paper, but only after line 528 in the Appendix. This is an important modeling choice that should have been present in the main paper.*"
>
> **Response:** Thanks for your comment. In Line 196 to 204, we have already mentioned the leveraging of fully connected layers to parameterize successive Householder vectors $\\{v_t, t \geq 0\\}$. For more clarification, we will explicitly formulate this parameterization in the main text of the revised version as suggested by the reviewer.
>
> **Question 4.** "*The Authors propose to run an empirical Bayes procedure for choosing the prior parameters but the effect of this choice is not assessed. Minor: From the discussion around line 192 it might be confusing for a reader why $\beta$ does not appear in Eq. 5; maybe it would be better to first show the full KL (Eq. 11 in Appendix) and then show the optimized one. As an alternative, you could use a different symbol for the optimized KL (like $\hat {D}_\{KL\} $).*"
>
> **Response:** Thanks for your comment. We would like to answer your concern about the effect of empirical Bayes procedure via two fronts: (i) KL condition guarantee; (ii) inference efficiency in general.
>
> (i) If your concern regards the role of the empirical Bayes in our method, we have mentioned in Line 192 to 195 in the main text. Concretely, this procedure helps us to specify the precision parameter $\beta$ of the prior distribution such that the KL condition is guaranteed, specifically with the optimized value of $\beta$, the KL term has the form independent of the deterministic parameter $\Theta$. The KL condition is a prerequisite, but also a bottleneck restricting the flexibility of Dropout inference frameworks (such as MC Dropout, VD) in terms of the expressiveness of both prior distribution and approximate posterior. However, for the correlated structure (via orthogonal parametrization) and the prior hierarchy in our method, we can ensure this condition with the empirical Bayes procedure without any further simplified assumption.
>
> (ii) If your concern regards the effect in general of the empirical Bayes on our inference algorithm, not specifically on the KL condition, we would like to add some high-level conceptual discussions. The empirical Bayes, in other words, a principle of data-dependent prior, which suffers from the main criticism of using data twice that is illegal in a strict Bayesian formalism [6]. In the mean-field VI for BNNs, this procedure was claimed to be able to yield slow convergence, introduce strange local minima and thus lead to miscalibrated predictive distributions [7]. However, there should be a more comprehensive study to investigate the effects of this procedure, especially in a complicated context of deep learning models, which are surrounded by other data-related techniques such as temperature scaling, data augmentation, and even Dropout -- a data-dependent regularization. On the other hand, empirical Bayes has been embraced and widely adopted in Bayesian machine learning, and especially in the seminal work on Bayesian neural nets. This technique has been employed as a principled approach to learning prior hyperparameters [8, 9], or automatically embodying automatic-relevance determination [10]. It should also be added that for the hierarchical parameterization in our method, we have just applied partly empirical Bayes for the precision parameter $\beta$ of conditional prior $p(W|z)$, while the variance of hyperprior $p(z)$ is tuned via cross-validation. This will facilitate flexible Bayesian modeling in our method. Arguably, some explorations more specific to VSD are promising.
>
> [6] Darnieder, "Bayesian methods for data-dependent priors", 2011.
>
> [7] Blundell et al., "Weight Uncertainty in Neural Networks", ICML 2015
>
> [8] Krishnan et al., "Specifying Weight Priors in Bayesian Deep Neural Networks with Empirical Bayes", AAAI 2020
>
> [9] Wu et al., "Deterministic Variational Inference for Robust Bayesian Neural Networks", ICLR 2020.
>
> [10] Kharitonov et al., "Variational Dropout via Empirical Bayes", Bayesian Deep Learning workshop, NeurIPS 2018.
>
> *** Regarding the induced KL term after employing the empirical Bayes, we will use a different symbol as your suggestion to avoid confusing the reader.

---

> > ### Author Response · Authors · 2021-08-10
> > **Response to Reviewer VNbM, Part 2**
> >
> > **Question 5.** "*Still regarding the empirical Bayes, given that the $\beta$ is a matrix with the same dimensions of $W$, would it be possible for the KL to collapse to 0? In the minimization of the variational objective, all it needs is for $\alpha_i$ -> 0  and it's not clear how this could be prevented. Did you experience this phenomenon in your experiments?*"
> >
> > **Response:** Thanks for your questions.
> >
> > Before answering your question, let us recap how the empirical Bayes is applied in our method. The parametrization of approximate posterior and prior in VSD allows a tractable calculation of the KL term, and by the empirical Bayes, we obtain the optimal value for precision parameter $\beta$ in an analytical form. This value is substituted back into the KL expression, and then we get a form independent of the deterministic weight $\Theta$. This procedure is equivalent to an iterative optimization algorithm for the ELBO objective in equation (4), in which $\beta$ will be updated until convergence after every single update of other parameters. However, disappearing the precision $\beta$ by directly substituting its empirical Bayes values will help to clarify the KL-condition guarantee in VSD. The data-dependent choice of this parameter is made explicitly through the dependence on the learned $\alpha$, the matrix $U$, and the weight $\Theta$ in equation (12).
> >
> > Going back to your question, we suppose what the reviewer meant is that when using one prior parameter per weight element, namely allow too many degrees of freedom, running an empirical Bayes procedure to adjust the prior can lead to a degenerate objective (e.g. KL term collapses to 0), and generally push the weight parameters towards the maximum likelihood solution. However, this depends largely on the parameterization of both approximate posterior and prior distribution, and then which prior parameters are learned. Specifically in our method, we adopt a fully-structured posterior and a zero-mean Gaussian prior with diagonal covariance, the KL term has a closed-form as shown in equation (5). Our KL term does not vanish in empirical training, and mathematically, we find one special case where the KL term can collapse to 0 is when the orthogonal matrix U is degenerate to the identity matrix (meaning the Householder transformation is deactivated) and the droprates $\alpha$'s go to infinity rather than to 0 (which we think is a singularity rather than a proper-potential solution) as the reviewer's comment. However, this does not happen in our methodology. We show here the KDE of the alpha values in our experiments to support our above arguments: https://imgur.com/a/mp55Gv3, in which we give statistics on the droprate values trained with LeNet5 on CIFAR10 dataset.
> >
> > **Question 6.** "*Connected with the point above, I can see why the Authors needed a hierarchical parameterization of the prior to enforce a stronger regularization. Regarding the hierarchical prior, the main paper doesn't discuss the parameterization of p(z) and q(z) (some details are present in the Appendix but should be reported also here). Minor: For Figure 2 with the double y-axis it's not possible to understand which axis is corresponding to which dataset.*"
> >
> > **Response:**
> >
> > *** Thanks for your insightful observation. With a thorough presentation in Appendix A.2, we analyzed some significant roles of prior hierarchy in our method, including: bridging multiplicative Gaussian noise to Gaussian scale mixture prior, making variational Bayes robust with mixture posterior approximation, and imposing a perturbation with different levels of stochasticity. In addition to that, when doing a joint variational inference procedure, it also enforces a stronger regularization in the ELBO objective. Furthermore, in our prior hierarchy, the latent $z$ has the size of the number of rows and is shared across columns of the weight matrix $W$. This row-partitioning will discourage allowing too many degrees of freedom in the parameterization. Basically, such a technique applied to variational Bayesian inference will prevent the model from the overfitting issue.
> >
> > *** Regarding the parameterization of $p(z)$ and $q(z)$. Thanks for your suggestion. We have chosen the (inverse) Gamma and log-Normal distribution respectively. These distributions have positive support and can be reparametrized. The KL-divergence between them also has a closed-form due to the conjugacy. We will add this to the main text in the revised version for clarification.
> >
> > ***  For figure 2, the left and the right y-axis are corresponding to SVHN and CIFAR10 datasets, respectively. We will add a detailed description in the caption of this figure in the revised version.
> >
> > **Question 7.** "*Two important remarks/contradictions after checking the submitted code, which are not discussed anywhere:...*"
> >
> > **Response:** Thanks for your comments.
> >
> > First of all, we apologize for not providing the entire code. We also appreciate that the reviewer took the time to check our code carefully. We will now respond to your concerns as follows:
> >
> > *** Regarding the convolutional version. In the image classification task with fully connected networks (for MNIST) and a small convolutional network (for CIFAR10, SVHN) in Section 4.1, we did not implement fully Bayesian inference for the convolutional layer, we instead just have done it on fully connected layers. This is because we *tried to conduct* a fair experiment to compare with some other structured approximations including Variational Matrix Gaussian (VMG) and SLANG (we mentioned their results in Line 296, 297 in the main text). These methods are non-trivial when applied to convolutional layers, even no available results for CNNs have been reported. On the other hand, for AlexNet and ResNet18 experiments in Section 4.2, we have performed fully Bayesian approximations on the whole architecture.
> >
> > *** Regarding the joint inference version with hierarchical prior. We have proposed this version as a unified framework in our method. Some exploitation for specific problems such as model selection, sparsity employed this version, which we think will be promising.
> >
> > We would like to add our straightforward code on full-Bayesian AlexNet and the code of joint inference version in the following anonymous Github link: https://anonymous.4open.science/r/vsd_sourcecode-BC73/README.md

---

> > > ### Comment · Reviewer_VNbM · 2021-09-01
> > > **Post-discussion changes**
> > >
> > > First of all, many thanks for the response. We had a lengthy conversation with the other reviewers concerning this paper, which helped me in making a final decision.
> > >
> > > I acknowledge the novelty of this submission and that these contributions are interesting for the BDL community. As such, I will increase my score to 5, but unfortunately there are some concerns in the presentation that don't allow me to increase further.
> > >
> > > Whether this paper will be accepted or not, I encourange the Authors to include a detailed discussion on the role of the hierarichical prior and the differences between the two ELBOs. For example, you should empathize that this choices increases model flexibility but that also makes the ELBO more regularized ("When doing a joint variational inference procedure, it also enforces a stronger regularization in the ELBO objective" and "Basically, such a technique applied to variational Bayesian inference will prevent the model from the overfitting issue").
> > >
> > > My second concern is on the mismatch between what is discussed and what is implement. For example, you have an entire paragraph in S3.4 to discuss how VSD is implemented for convolutional layers; from the setup described in S4.2 it sounds like this is the default option, while it appears to be used only for the experiments in Table 2 and 3 (I couldn't find any discussion on this neither in the main paper nor in the supplement). Again, for a new version of the paper, please be more clear in the description of the experimental setup.

---

> > > > ### Author Response · Authors · 2021-09-01
> > > > **Thank you for your feedback**
> > > >
> > > > We would like to thank the reviewer for acknowledging the contributions of our paper and changing the score.
> > > >
> > > > Regarding your comment for the experimental descriptions, we will clarify in Sections 4.1 & 4.2 as in our response to your Question-7.
> > > >
> > > > Regarding the roles of hierarchical prior, we will include the regularization effect as your suggestion. However, by our response to your Question-6 (and also Question-4 & 5), we wish to remark that this effect is not a primary purpose of our original framework, and obviously, we have actually achieved noticeable improvements with the proposal of fully-structured posterior as shown in Fig 2.

---

> > ### Author Response · Authors · 2021-08-30
> > **Feedback before the discussion ends**
> >
> > Dear Reviewer VNbM,
> >
> > We would like to thank you again for spending your time evaluating our paper.
> >
> > As the discussion period is expected to conclude early this week, we look forward to hearing your feedback about whether we have addressed your concerns in the rebuttals. We would be happy to discuss if you still have any other concerns.
> >
> > Best,
> >
> > Authors

---

### Official Review · Reviewer_uDiH · 2021-07-16

**Rating:** 6
**Confidence:** 3

**Summary:**

The paper proposes Variational Structured Dropout, an extension to variational dropout that uses a full covariance matrix to model the posterior at the pre-activations. This covariance matrix, after spectral decomposition, is parameterized as the product of T Householder matrices.

The new parameterization of the posterior does not suffer from the singularity issue of Variational Dropout. When using a diagonal posterior approximation, the singularity issue is that the approximate posterior has zero measure, which results in the KL-term being undefined. VSD avoids this by using a full covariance matrix thus ensuring that the approximate posterior has non-zero measure.

To train the model using the new parameterization, the authors optimize a new variational lower bound. This lower bound is a lower bound to the ELBO.

Regarding the prior, the paper advocates for using a hierarchical prior distribution where the prior is a zero mean Gaussian distribution over the weights, with the hyperprior determining the standard deviation.

The experiments first examine the impact of T, the number of matrices used to parameterize the approximate posterior. Next, large scale experiments on deep neural networks demonstrate the performance of VSD against a number of baseline Bayesian inference methods measuring both predictive performance and predictive entropy (for OOD detection).


**Ethical Concerns:**

No ethical concerns.

**Limitations And Societal Impact:**

The limits and societal impacts are addressed adequately.

**Main Review:**

### Originality:

I find the idea of VSD quite interesting and it is, to my knowledge, novel. It is an excellent observation that one can address the singularity issues using a full covariance approximate posterior. Moreover, it can also be understood as a novel type of low rank parameterization of the approximate posterior.

Regarding the hierarchical prior, I have not seen a hierarchical prior over the weights in this form, although it is quite possible that similar ideas have been proposed before.

### Quality:

The work is high quality. It is well motivated, it addresses the concerns with variational dropout and it also improves performance.

I have a slight concern regarding the new variational lower bound $L_{MI}$ . it is unclear to me what the effect of discarding the mutual information is. I would expect that with $L_{MI}$, we get highly correlated weights posteriors. Why doesn’t this training objective result in the weight posterior having high correlations, thus having very low probability mass?

The experimental section of the paper could be improved. It is good to see that numerous benchmarks and baseline methods are used, but all the methods seem to be underperforming. For instance, the ResNet-18 results are lagging significantly behind the reference implementations available at https://github.com/google/uncertainty-baselines/tree/main/baselines/cifar . Apart from this, the experiments look at both predictive performance and predictive entropy (for OOD samples), and they show very promising results.

Regarding the derivations in the supplementary material: I read the results, but I did not carefully check them.

### Clarity:

The paper does an excellent job at presenting the method, explaining the motivation and describing related works. The paper flows very well, each section leading into the next.

### Significance:

The work has potential impact for both researchers and deep learning practitioners. For researchers, it is certainly interesting how the work addresses the theoretical concerns with VD. For practitioners, VSD could be an effective approach at capturing model uncertainty depending on the computational costs.


### Further questions:

* The paper mentions that using T > 2 has high computational cost. Is this because pre-activations are costly to compute, or that the method converges too slowly? What is the computational cost and complexity associated with the VSD parameterization?
* How many samples (or forward passes) are used for prediction?
* What hyperparameter values $\alpha$ and $\beta$ were used in the experiments?


### Minor:

* Line 235 convolutional instead of convolution?
* Many of the references, such as [26] and many others cite Arxiv versions of papers instead of the versions that are published at ML conferences. Also names such as Bernoulli are sometimes not capitalized correctly.



**Time Spent Reviewing:**

4

---

> ### Author Response · Authors · 2021-08-10
> **Response to Reviewer uDiH**
>
> We would like to thank you for your comments and suggestions. Below, you can find our responses to your reviews:
>
> **Question 1.** "*I have a slight concern regarding the new variational lower bound $L_\{MI\}$. It is unclear to me what the effect of discarding the mutual information is. I would expect that with $L_\{MI\}$, we get highly correlated weights posteriors. Why doesn’t this training objective result in the weight posterior having high correlations, thus having very low probability mass?*"
>
> **Response:** Thank you for your comment. Let us take this opportunity to clarify your confusing point about the mutual information term. When adding a mutual information term to the original ELBO as an additional regularization, we maximize an alternative variational objective $L_{MI}$ which is not necessarily a valid lower bound of the model evidence. Our technique also means that we have discarded the **negative** mutual information term involved in the **negative** KL of the original ELBO. This makes sense because as we argued in Line 175 to 182, the presence of this **negative** mutual information term in the original ELBO can break the strong correlations between the columns of $W$ in the approximate posterior $q_t(W)$.
>
> We would also like to add that, although we may not optimize a valid lower bound of the model evidence, our information-theoretic solution has fixed appropriately the "broken" ELBO as mentioned above. Indeed, our idea was further motivated by the similar technique that has been extensively adopted in deep latent variable models, in which a mutual information maximization is also added to the variational lower bound to mitigate the degenerate issue of amortized inference in these models [1].
>
> [1] Zhao et al., "InfoVAE: Balancing Learning and Inference in Variational Autoencoders", AAAI 2019.
>
> **Question 2.** "*All the methods seem to be underperforming. For instance, the ResNet-18 results are lagging significantly behind the reference implementations available at https://github.com/google/uncertainty-baselines/tree/main/baselines/cifar.*"
>
> **Response:** Thank you very much for providing the useful reference. Firstly, we want to clarify that our experiments in Tables 2 & 3 adopted the settings implemented in previous works, specifically in [2] and [3] as mentioned in Appendix I.5. Our reported results are consistent with the results of those works. For the link the reviewer referred to, we found [4] (cSGHMC method) reports some results of ResNet-18 trained on CIFAR10 and CIFAR100 datasets. Following the code provided in this paper [4], we noticed that the authors used a different setting. Concretely, in the first convolutional layer, they used a smaller kernel size (3 versus 7 in our setting) and a smaller stride ( 1 versus 2 in our setting). In addition,  we also used one more Max-Pooling layer (kernel_size=3, stride=2, padding=1) before forwarding through blocks. As a result, the feature maps in [4] is 16 times larger than ours in each convolutional layer after the first one, and thus leads to greater capacity and improving performance.
> We conducted a quick experiment with the setting of [4], our method achieves competitive results (5.26±0.03 %error on CIFAR10, 22.58±0.05 %error on CIFAR100) as in the following anonymous link: https://imgur.com/a/QHUPlZb, especially VSD is comparable to typical MCMC methods such as SGDL, SGHMC.
>
> [2] Tomczak et al., "Efficient Low Rank Gaussian Variational Inference for Neural Networks", NeurIPS 2020.
>
> [3] Osawa et al., "Practical Deep Learning with Bayesian Principles", NeurIPS 2019.
>
> [4] Zhang et al., "Cyclical Stochastic Gradient MCMC for Bayesian Deep Learning", ICLR 2020.
>
>
> **Question 3.** "*Further questions: ...*"
>
> **Response:** Thanks for your questions.
>
> *** Regarding the computational complexity and memory cost. We first would like to refer to Section 3.4 and Tables 4 & 5 in Appendix D. We have provided a thorough analysis of the complexity of our method compared to other variational methods in both algorithmic and experimental aspects. Concretely, VSD implements a structured approximation with practical computation time even more effective than mean-field VI. This is because the structured parameterization in our method has successfully adopted the advantages of Dropout training: just sampling the low dimension noise instead of whole random weights, and allows one forward pass in parallel compared to two steps of the local reparameterization trick in mean-field VI.
>
> Our additional complexity depends on two quantities:
>
> - $T$: the number of transformation steps. To make an expressive orthogonal parametrization with small $T$, we used a fully connected layer between successive Householder vectors as mentioned in Line 296 to 201, so VSD will incur additional time when computing the product of Householder matrices with larger $T$. On the other hand, VSD can be trained more effectively with a parallel algorithm introduced in [5].
>
> - $K$: the number of rows in fully connected layers, or the number of channels in convolutional layers. The size of Householder matrices and the FC layer between Householder vectors all are $K \times K$, so a larger $K$ will add more complexity.
>
> In the following anonymous link: https://imgur.com/a/ANywrcS, we provide a practical computation time Table of VSD trained with LeNet5, AlexNet, ResNet18, and PreResNet110 on CIFAR10 dataset. In the table, VSD with $T=2$ exhibits extra computation time compared to $T=1$, the increase on ResNet18 is more evident than on LeNet and AlexNet. However, on more modern architectures such as PreResNet 110 which prefers to evolve in depth rather than width (namely using fewer channels), VSD with $T=2$ does not endure much extra computation time and thus makes a good adaptation. This can be observed directly in the Table in the link via the time-scale value of VSD when using $T=2$ compared to $T=1$.
>
> [5] Mathiasen et al., "Faster Orthogonal Parameterization with Householder Matrices", 2020.
>
> *** Regarding the number of Monte Carlo samples for prediction. We have already mentioned in Appendix I. For the UCI regression task, following the original settings of previous methods, we used 10000 MC samples for Bayesian Dropout methods (MC Dropout, VD, VSD). For the classification task, we used 50 MC samples and 10 MC samples for the experiments in Table 1 and Table 2 & 3, respectively.
>
> *** Regarding the hyperparameter values $\alpha$ and $\beta$ were used in the experiments. The $\alpha$ is the variance of Gaussian noise and also the dropout rate in VSD. This quantity, instead of being adjusted manually, is treated as a variational parameter in our method and thus is learned simultaneously with other parameters via maximizing the variational objective function. On the other hand, the precision $\beta$ is specified through an empirical Bayes procedure, as a result, we obtain an optimal value for this hyperparameter in an analytical form. The optimal $\beta$ is then substituted back into the KL term, and thereby we get a form independent of the deterministic weight $\Theta$. This process is equivalent to an iterative optimization algorithm for the ELBO objective in Equation (4), in which $\beta$ will be updated until convergence after every single update of other parameters. However, disappearing the precision $\beta$ by directly substituting its empirical Bayes values will help to clarify the KL-condition guarantee in VSD. The data-dependent choice of this parameter is made explicitly through the dependence on the learned $\alpha$, the matrix $U$, and the weight $\Theta$ in Equation (12). A more detailed analysis has been given in Appendix B.
>
> **Question 4.** "*Minor: Line 235 convolutional instead of convolution? Many of the references, such as [26] and many others cite Arxiv versions of papers instead of the versions that are published at ML conferences. Also, names such as Bernoulli are sometimes not capitalized correctly.*"
>
> **Response:** Thank you very much for pointing these out. We will correct them all in the revised version.

---

> > ### Comment · Reviewer_uDiH · 2021-08-17
> > **Thank you for the reply**
> >
> > Thank you for the reply. I appreciate the detailed responses to my concerns.

---

### Official Review · Reviewer_idcU · 2021-07-19

**Rating:** 7
**Confidence:** 4

**Summary:**

This paper introduces Variational Structured Dropout (VSD), a scalable method building on previous Variational Dropout methods, and with a structured representation of the variational noise. The paper derives the method, introduces a hierarchical prior, and provides results on many benchmarks while comparing to many baselines. VSD performs reasonably well in all benchmarks on uncertainty metrics and accuracy. VSD also performs well on out-of-distribution uncertainty experiments.

**Limitations And Societal Impact:**

The authors could add a line or two on potential societal impacts, but this is not necessary.

**Main Review:**

I think this is a good paper and a nice method. There is good motivation, good explanations and a thorough experiments section, where VSD is seen to perform well. Overall I like the paper, and so I focus the rest of the review on a few questions that, if answered, could increase my score:

1. In Equation 4, the authors add a mutual information term to the variational lower bound. Is this new objective still a valid lower bound of the model evidence? It apears to me like it might not be due to the mutual information term. If indeed it is no longer a valid lower bound, I think the authors should make it clearer in the text.
- Also, in Appendix I.1, the authors describe the KL annealing that they use, where they weigh the KL term by $\lambda$. For $\lambda \neq 1$, the objective is no longer a valid lower bound. I think it is fine to do this (as it has been found to help in the past), but I think this additional hyperparameter should be explicitly mentioned in the main text somewhere.
2. The results are all means over 5 random seeds. I would like to see standard deviations too, either in the main text or in additional tables in the Appendix.

Additional, more minor points:

3. I think it would be nice to have some further intuition on what T=1 vs T=2 means.
4. The authors could also compare to natural-gradient VI methods like VOGN (Osawa et al., 2019: "Practical Deep Learning with Bayesian Principles").


---------- After author rebuttal -----------
Thanks to the authors for responding to my questions and concerns. They answered many of my questions. However, given very valid concerns from other reviewers (particularly the discussion with Reviewer VNbM), I have decided to keep my score and not increase it. I do this with the expectation that the authors will fix/clarify many of Reviewer VNbM's concerns, such as:
- Being explicit in the text about the potential limitation that the new ELBO has a relatively poor regularisation, which may be the reason that the hyperprior is necessary and works well.
- More importantly, being explicit in the text regarding *exactly* which implementation is used for which experiment, with key differences in the main paper, and smaller settings in an Appendix. Reviewer VNbM brought up a very important point regarding implementations and this should really have been in the submitted draft of the paper itself.

**Time Spent Reviewing:**

3

---

> ### Author Response · Authors · 2021-08-10
> **Response to Reviewer idcU**
>
> We would like to thank the reviewer for spending time reading our paper and providing positive feedbacks. Below, you can find our response with your questions:
>
> **Question 1a.** "*In Equation 4, the authors add a mutual information term to the variational lower bound. Is this new objective still a valid lower bound of the model evidence? It appears to me like it might not be due to the mutual information term. If indeed it is no longer a valid lower bound, I think the authors should make it clearer in the text.*"
>
> **Response:** Thanks for your insightful comment. Our surrogate variational objective obtained when adding a mutual information term to the original ELBO is not necessarily a valid lower bound of the model evidence. However, this information-theoretic solution helps to fix the "broken" ELBO as analyzed in Equation (3) in section 3.2. Indeed, our idea was further motivated by the similar technique that has been extensively adopted in deep latent variable models, in which a mutual information maximization is also added to the variational lower bound to mitigate the degenerate issue of amortized inference in these models [1]
>
> [1] Zhao et al., "InfoVAE: Balancing Learning and Inference in Variational Autoencoders", AAAI 2019.
>
> **Question 1b.** "*Also, in Appendix I.1, the authors describe the KL annealing that they use, where they weigh the KL term by $\lambda$. For $\lambda \ne 1$, the objective is no longer a valid lower bound. I think it is fine to do this (as it has been found to help in the past), but I think this additional hyperparameter should be explicitly mentioned in the main text somewhere.*"
>
> **Response:** Thanks for your suggestion. KL annealing has an interpretation of the Bayesian principle. Reweighting the KL term by a $\lambda$ is equivalent to tempering the posterior by a temperature factor $\tau = \lambda$ [2], namely using $p_{\tau}(W|D) \propto p(D|W)^{1/\tau}p(W)$ (see Appendix F). In another interpretation, one can use a different likelihood $p_{\tau}(D|W) = p(D|W)^{1/\tau}$ instead of a tempered posterior $p_{\tau}(W|D)$ above. Even though the posteriors coincide, the predictive distribution differs for these two ways [3].
>
> A temperature value $\tau < 1$, namely $\lambda < 1$ in our experiments, is then corresponding to artificially sharpening the posterior distribution, which can be interpreted as overcounting the data $D$ by a factor of $1/\tau$. So, the variational objective with KL annealing is a lower bound of $\tau$ times the model evidence defined on the overcounted data $\widehat\{D\}$. Maximizing this objective is equivalent to minimizing the KL divergence between the approximate distribution $q(W)$ and the tempered posterior $p_{\tau}(W|D)$.
>
> Regarding our choice to the $\lambda$, this hyperparameter is tuned by cross-validation. The most appropriate values we have found are consistent with the reports of previous works in the literature that have been synthesized in [2]. Specifically, we used $\lambda \in \\{0.1, 0.2, 0.5\\}$ for the classification tasks, and $\lambda$ from $10^{-5}$ to $10^{-2}$ for the regression tasks. We will include these settings in the main text in the revision of our manuscript.
>
> [2] Wenzel et al., "How Good is the Bayes Posterior in Deep Neural Networks Really?", ICML 2020.
>
> [3] Wilson and Izmailov, "Bayesian Deep Learning and a Probabilistic Perspective of Generalization", NeurIPS 2020.
>
>
> **Question 2.** *"The results are all means over 5 random seeds. I would like to see standard deviations too, either in the main text or in additional tables in the Appendix."*
>
> **Response**: Thanks for your suggestion. For the convenience of presentation, we have not included the standard deviations of empirical results in the main text. We will add the detailed table in Appendix G in the revised version as suggested by the reviewer. Temporarily, we provide tables with standard deviations in the following anonymous link: https://imgur.com/a/odOuYdK
>
> **Question 3.** *"I think it would be nice to have some further intuition on what $T=1$ vs $T=2$ means.*"
>
> **Response:** Thanks for your question. To provide some insights about the number of transformation steps $T$ in our method, we first would like to recap its role in representing an arbitrary orthogonal matrix by the Householder transformations. As we mentioned in Line 132 to 136, by the basis-kernel representation theorem [3], any orthogonal matrix $P$ with the basis acting on the $T^*$-dimensional subspace, namely orthogonal matrices of degree $T^*$, can be expressed as a product of exactly $T^*$ Householder transformations. The degree $T^*$ of the orthogonal matrix $P$ is less than equal to its order $K$. So, the number of transformation steps $T=T^*$ is sufficient to be capable of representing the orthogonal matrix $P$ of degree $T^*$.
>
> - Why $T \in \\{1,2\\}$ is proposed in our method: For deep learning architectures, the value of $K$, and so $T^*$, is relatively large (in our method, $K$ is the size of the hidden unit for fully connected layers, and is the number of channels for convolutional layers), so it is quite challenging to adjust the value of $T$ in a principled way to meet the theory. We can employ an efficient parameterization introduced in [5], in which we only need the Householder vectors $\\{v_t\\}$ with sizes much smaller than $K$, and this will facilitate tuning $T$ with a larger range in an applicable computation time. In our work, we instead only adopt a small $T \in \\{1,2,3\\}$  and to make expressive reflections, we use a fully connected layer between successive Householder vectors $\\{v_t\\}$. Indeed, this neural parameterization has been widely leveraged in the context of deep latent-variable models [6] as we mentioned in Line 197.
>
> [4] Xiaobai Sun and Christian Bischof, "A Basis-Kernel Representation of Orthogonal Matrices", 1995.
>
> [5] Zhang et al., "Stabilizing Gradients for Deep Neural Networks via Efficient SVD Parameterization", ICML 2018.
>
> [6] Berg et al., "Sylvester normalizing flows for variational inference", UAI 2018.
>
>
> **Question 4.** *"The authors could also compare to natural-gradient VI methods like VOGN (Osawa et al., 2019: "Practical Deep Learning with Bayesian Principles").*"
>
> **Response:** Thank you for your suggestion. We already compared our method to a natural gradient VI approach named SLANG [7]. Unfortunately, we have not successfully implemented this baseline to consistently reproduce reasonable results. Some inherited results of this method are shown at Line 297 (MNIST classification task) in the main text and in Table 8 (UCI regression task) of Appendix G.2.
> For the VOGN method, we provide some additional comparisons with VSD at the following anonymous link: https://imgur.com/a/dhGzLhs
>
> As shown in the table, our method VSD has better performance than VOGN on almost all metrics, especially on the accuracy and computational time (1x corresponding to standard MAP training).
>
> [7] Mishkin et al., "SLANG: Fast Structured Covariance Approximations for Bayesian Deep Learning with Natural Gradient", NeurIPS 2018.

---

> ### Author Response · Authors · 2021-08-30
> **Feedback before the discussion ends**
>
> Dear Reviewer idcU,
>
> We would like to thank you again for spending your time evaluating our paper.
>
> As the discussion period is expected to conclude early this week, we look forward to hearing your feedback about whether we have addressed your concerns in the rebuttals. We would be happy to discuss if you still have any other concerns.
>
> Best,
>
> Authors

---

### Decision · Program_Chairs · 2021-09-27

**Decision:**

Accept (Poster)

**Comment:**

This paper proposes structural variational dropout for BNNs.

Reviewers think the proposed methodology is novel, and the paper's theoretical contribution is significant, in the sense that it addresses a well-known theoretical issue of variational dropout & MC-dropout. Given that MC-dropout is quite often used in practice, such theoretical contribution is useful.

Still the paper has the problem that, some reviewers are concerned with the experiments using implementations that are different from the version used in theoretical analysis. So in revision, either this difference needs to be explained (if it is a confusion), or, the practical approach needs to be justified better to so that they do not have the infinite KL issues of MC-dropout.